# Generalization of Spectral Graph Neural Networks

## Abstract

Spectral graph neural networks (GNNs) have achieved remarkable success across various applications, yet their generalization properties remain poorly understood. This paper bridges this gap by analyzing the impact of graph homophily and architectural choices on the generalization of spectral GNNs. We derive a general form of uniform transductive stability for spectral GNNs and provide an explicit stability analysis for graphs with two node classes, providing a comprehensive framework to understand their generalization. Based on this stability analysis, we establish a generalization error bound, demonstrating that better stability leads to improved generalization. Our theoretical findings reveal that spectral GNNs generalize well on graphs with strong homophily or heterophily but struggle on graphs with weaker structural properties. We also identify conditions under which increasing the polynomial order in spectral GNN architectures may degrade generalization. Empirical results on synthetic and real-world benchmark datasets align closely with our theoretical findings.

## 1 Introduction

Generalization is a fundamental challenge in machine learning, crucial for understanding both the theoretical limits and practical performance of algorithms. Researchers have developed various measures to study generalization, including Vapnik–Chervonenkis (VC) dimension (Cherkassky et al., 1999), PAC-Bayes bound (McAllester, 1998), Rademacher complexity (Bartlett & Mendelson, 2002), and the stability of learning algorithms (Bousquet & Elisseeff, 2002). These measures provide insights into a model's ability to generalize beyond its training data. Graph neural networks (GNNs) have achieved remarkable success across various practical applications (Zhou et al., 2020), yet their generalization capabilities remain poorly understood. Unlike traditional machine learning models that operate on independent and identically distributed (i.i.d.) data, GNNs work on interdependent data where the graph topology and node/edge features are inherently linked. This interconnected structure makes it difficult to evaluate how well a GNN trained on one graph or a set of graphs can generalize to unseen graphs.

Research on GNN generalization primarily focuses on two tasks: graph classification and node classification. In graph classification, where graphs are typically i.i.d., generalization has been studied through connections with WL algorithms (Morris et al., 2023; D'Inverno et al., 2024; Franks et al., 2024) and data-dependent PAC-Bayes bounds (Liao et al., 2021; Ju et al., 2023). In node classification, which involves transductive learning where node features are known during training, approaches like Transductive Rademacher complexity and uniform transductive stability are more common. These methods explore the impact of factors such as graph matrix representations and GNN depth on generalization (Oono & Suzuki, 2020; El-Yaniv & Pechyony, 2007; Tang & Liu, 2023b; Zhou & Wang, 2021; Cong et al., 2021).

In this work, we focus on the generalization of GNNs for node classification. Unlike graph classification, node classification performance is influenced by the distribution of node classes, which is closely tied to graph homophily. In homophilic graphs, connected nodes tend to belong to the same class, whereas in heterophilic graphs, connected nodes are often from different classes. Empirical evidence shows that the edge homophilic ratio (Zhu et al., 2020) significantly affects GNN performance. For example, models like GCN and GAT excel on homophilic graphs but underperform on heterophilic graphs (Kipf & Welling, 2017; Velickovic et al., 2018). This motivates us to explore the relationship between graph homophily and the generalization of GNNs in node classification tasks,

raising the question: *how does GNN generalization depend on node class distributions?* Notably, this question has remained largely unexplored in previous research.

We examine the generalization of GNNs through a spectral perspective, as spectral GNNs have demonstrated strong performance on both homophilic and heterophilic graphs. Spectral GNNs operate in the spectral domain, applying frequency-domain convolutions to extract structural information (Balcilar et al., 2021). Formally, a spectral GNN is defined as:

$$\Psi(M, X) = \sigma(g_\Theta(M) f_W(X)), \tag{1}$$

where $M \in \mathbb{R}^{n \times n}$ is a graph matrix (e.g., Laplacian or adjacency matrix), $g_\Theta(M) = \sum_{k=0}^{K} \theta_k T_k(M)$ represents graph convolution using $\Theta = \{\theta_k\}_{k=0}^{K}$ and the $k$-th polynomial basis $T_k(\cdot)$, $f_W(X)$ is a feature transformation function parameterized by $W$, and $\sigma$ is a non-linear function such as softmax.

One notable observation about spectral GNNs is that, although the frequency response function can theoretically be approximated by a sum of polynomial basis with negligible error as the order $K$ increases (Powell, 1981), empirical results show that higher polynomial orders do not consistently lead to better performance. This discrepancy raises the question: *is the gap between theoretical study and empirical performance, particularly with respect to polynomial order $K$, related to the generalization of spectral GNNs?* To date, no work has provided a clear explanation for this phenomenon.

We address the aforementioned questions by focusing on the generalization of spectral GNNs. In transductive learning, the relationships between labeled and unlabeled nodes are critical for generalization. To measure this, we employ uniform transductive stability (El-Yaniv & Pechyony, 2006), which evaluates the stability of spectral GNNs under perturbations to individual training samples.

To study the interplay between graph structure and GNN performance, we use the contextual stochastic block model (cSBM) (Baranwal et al., 2023; Deshpande et al., 2018), a widely used generative model that captures both homophilic and heterophilic graph structures in a controlled and analytically tractable manner. Previous studies have demonstrated that cSBM models real-world datasets such as Citeseer, Cora, and Polblogs, which are frequently used in GNN research (Deshpande et al., 2018; LEI, 2016; Dreveton et al., 2023; Kipf & Welling, 2017; Zhang et al., 2021). Specifically, cSBM generates graphs with well-defined block structures, where nodes within the same block are more likely to be connected (homophilic graphs) or nodes between blocks have a higher connection probability (heterophilic graphs). Leveraging cSBM allows us to systematically vary graph homophily and examine its impact on GNN generalization properties.

**Contributions.** Our main contributions are summarized as follows:

- We analyze the $\gamma$-uniform transductive stability of spectral GNNs by decomposing it into two factors: the Lipschitz continuity and smoothness of the loss function and the spectral GNN, and the gradient norm bound (Theorem 6). This enables us to study the effects of node class distribution and spectral GNN architecture on training gradients through *an explicit gradient norm bound* (Theorem 8).

- We establish the generalization error bound of spectral GNNs based on their stability, where good stability indicates strong generalization capability (Theorem 9). To further explicitly analyze the effects of graph homophily and polynomial order on generalization, we derive an explicit form of the gradient norm bound for two node classes (Theorem 13).

- We prove that spectral GNNs generalize well on graphs that are strong homophilic or heterophilic, but perform poorly on graphs that are moderately homophilic or heterophilic. We identify conditions under which increasing the polynomial order in spectral GNN architectures may degrade generalization, providing insights into architectural design (Theorems 14 and 15; Proposition 16).

To validate our theoretical findings, we conduct experiments on nine synthetic datasets and nine real-world benchmark datasets for node classification. The experimental results align closely with our theoretical analysis.

## 2 RELATED WORKS

**Previous work.** We review prior studies on GNN generalization, typically categorized into graph classification and node classification, with a primary focus on the latter.

| Generalization Bound Methods | | Analysis | | Key factors in Bounds | | | | |
|---|---|---|---|---|---|---|---|---|
| | | Lipschitz | Gradient | Graph Matrix | Tsize | Depth | Hom | Poly |
| VC bound | (Scarselli et al., 2018) | – | n/a | $n$ | $\checkmark$ | – | – | – |
| RC bound (*transductive*) | (Esser et al., 2021) | $\checkmark$ | n/a | $\|M\|_\infty, \|MX\|_{2\to\infty}$ | $\checkmark$ | $\checkmark$ | – | – |
| | (Tang & Liu, 2023b) | $\checkmark$ | n/a | $\|M\|_\infty$ | $\checkmark$ | $\checkmark$ | – | – |
| | (Oono & Suzuki, 2020) | $\checkmark$ | n/a | $\|MX\|_F$ | $\checkmark$ | $\checkmark$ | – | – |
| US bound (*Inductive*) | (Verma & Zhang, 2019) | – | $\checkmark$ | $\|M\|_2$ | $\checkmark$ | – | – | – |
| | (Zhou & Wang, 2021) | – | $\checkmark$ | $\|M\|_2, \|MX\|_2$ | $\checkmark$ | $\checkmark$ | – | – |
| US bound (*transductive*) | (Cong et al., 2021) | $\checkmark$ | $\checkmark$ | $d_{max}$ | $\checkmark$ | $\checkmark$ | – | – |
| | Our work | $\checkmark$ | $\checkmark$ | $M_{ij}$ | $\checkmark$ | – | $\checkmark$ | $\checkmark$ |

Table 1: Comparison of generalization bounds for GNNs on node classification. Here, $\|\cdot\|_2, \|\cdot\|_F, \|\cdot\|_\infty, \|\cdot\|_{2\to\infty}$ denote the spectral norm, Frobenius norm, infinity norm and maximum column $\ell_2$-norm. $n$ is graph node number. $d_{\max}$ denotes the maximum node degree. The other factors Tsize, Depth, Hom, and Poly refer to training sample size, model depth, graph homophily, and polynomial order, respectively. $\checkmark$ indicates "discussed", while – indicates "not discussed".

Graph classification is typically considered as an inductive learning task. One prominent method is the Vapnik–Chervonenkis (VC) bound, which relates the VC dimension of a GNN to the number of colors generated by the 1-WL algorithm, reflecting the number of graphs the 1-WL algorithm can distinguish (Morris et al., 2023). The PAC-Bayes bound is another approach. Liao et al. (2021) connects generalization bound to factors like maximum node degree and GNN depth, while Ju et al. (2023) refines this by tying the bound to the largest singular value of the graph matrix. Behboodi et al. (2022) further extends the PAC-Bayes framework to equivariant networks, highlighting the influence of group properties on generalization. Rademacher complexity and uniform convergence have also been explored. Garg et al. (2020) shows that a GNN's computational tree complexity bounds its overall complexity, and Maskey et al. (2022) shows that generalization bound increases with model complexity but decreases with higher average node degrees.

Generalization analysis for node classification is more challenging than for graph classification due to its transductive nature (Tang & Liu, 2023b). Recent studies focus on how graph matrices and GNN depth influence generalization. Rademacher complexity (RC) is commonly used for node classification tasks due to its strong theoretical foundation in transductive learning (El-Yaniv & Pechyony, 2007). It has been shown that the transductive RC of a GNN is proportional to the infinity norm of its graph matrix, with generalization bounds provided for several classic GNNs (Tang & Liu, 2023b). Esser et al. (2021) uses a planted model to illustrate the relationship between GNN generalization and graph matrix compatibility. Oono & Suzuki (2020) shows that gradient boosting reduces RC in multi-scale GNNs. Uniform stability (US) offers another key approach for analyzing generalization. Verma & Zhang (2019) relates the generalization error bound of single-layer GCNs to the largest absolute eigenvalue of the graph matrix, while Cong et al. (2021) demonstrates that increasing GNN depth improves stability and lowers generalization error bounds. Other works deviate from transductive assumptions, addressing non-i.i.d. settings. For example, Ma et al. (2021) derives PAC-Bayesian bounds by assuming i.i.d. node classes given fixed node features.

**Our work.** In this work, we focus on node classification tasks, investigating how node class distribution and the architecture of spectral GNNs influence generalization. While prior studies have examined factors like graph size, training set size, graph matrix norms, and node features, they have largely overlooked the role of graph homophily in generalization and the impact of increasing the polynomial order of spectral GNNs. To our knowledge, this is the first study to analyze the effects of graph homophily and polynomial order on the generalization of spectral GNNs.

Table 1 compares our work with other methods for node classification, highlighting several key aspects: (1) **Analysis settings** (inductive or transductive): The VC bound (Scarselli et al., 2018) is data-independent and thus agnostic to inductive or transductive settings. While Verma & Zhang (2019); Zhou & Wang (2021) derive bounds for GNNs in inductive settings, others (Esser et al., 2021; Tang & Liu, 2023b; Oono & Suzuki, 2020; Cong et al., 2021) and our work address the more complex transductive setting. (2) **Analysis frameworks**: Rademacher complexity estimates a model's capacity to fit noise based on graph structure and node features but does not account for node labels. As a result, methods such as (Esser et al., 2021; Tang & Liu, 2023b; Oono & Suzuki, 2020) are unable to analyze the effect of graph homophily, which depends on both graph structure and node

labels. In contrast, uniform stability enables analysis of the relationship between generalization and graph homophily using gradient-based analysis. It is worth noting that while (Cong et al., 2021) employs uniform transductive stability, their analysis focuses solely on how GNN depth impacts Lipschitz continuity and gradient. (3) **Key factors in bounds**: Training sample size is a critical factor in all bounds except the VC bound (Scarselli et al., 2018). Model depth (number of GNN layers) is addressed in (Esser et al., 2021; Tang & Liu, 2023b; Zhou & Wang, 2021; Cong et al., 2021). Our work examines spectral GNNs, where the architecture comprises only one layer of $K$-order polynomials. Unlike prior studies that focus on various graph matrix norms, our analysis takes a finer-grained approach by considering the expectation of individual graph matrix elements. Moreover, we explore the effects of graph homophily and the polynomial order on the generalization of spectral GNNs, aspects not previously investigated.

## 3 PROBLEM SETUP

An undirected, unweighted graph is denoted as $G = (V, E)$, where $V$ is the set of nodes, $E$ is the set of edges, and $|V| = n$. In this work, we model graphs with controlled properties using the generalized multi-class contextual stochastic block model (cSBM) (Baranwal et al., 2023). A graph $G \sim cSBM(n, f, \Pi, Q)$ consists of $n$ nodes, partitioned into $C$ distinct classes. Here, $f$ is the dimension of node features, $\Pi = \{\Pi_i\}_{i \in [C]}$ is a set of $C$ continuous distributions, where $\pi_i \in \mathbb{R}^f$ and $\Sigma_i \in \mathbb{R}^{f \times f}$ are the mean and variance of $\Pi_i$ respectively, and $Q \in \mathbb{R}^{C \times C}$ is a symmetric edge-probability matrix. Each node $v_i$ is assigned a class $y_i$ sampled uniformly from a set of $C$ classes, and its feature vector $x_i \in \mathbb{R}^f$ is drawn from the distribution $\Pi_{y_i}$. This generates the node feature matrix $X \in \mathbb{R}^{n \times f}$. For the adjacency matrix $A \in \mathbb{R}^{n \times n}$, an edge between node $v_i$ and $v_j$ (i.e., $A_{ij} = 1$) is generated according to the edge-probability matrix $Q$ based on the following probability:

$$P(A_{ij} = 1 \mid y_i, y_j) = Q_{y_i y_j},$$

where $Q_{y_i y_j}$ gives the edge formation probability between class $y_i$ and class $y_j$.

For a spectral GNN $\Psi$, following (Wang & Zhang, 2022; Lu et al., 2024), we first consider $f_W(X) = XW$, and then discuss the effect of non-linear feature transformation. Here $W \in \mathbb{R}^{f \times C}$. Let $S_m = (X, \{y_i\}_{i=1}^m)$ be a training set containing $m$ labelled nodes, randomly sampled form the graph $G$, and $\mathcal{D}_u = (X, \{y_i\}_{i=m+1}^n)$ be the testing set containing the other nodes in the graph $G$. We define a loss function $\ell(y_i, \hat{y}_i|_{\Theta, W})$ to measure the discrepancy between the truth class $y_i$ and the prediction $\hat{y}_i$ when a spectral GNN is parameterized by $\Theta, W$. The empirical loss $\mathcal{L}_{S_m}(\Theta, W)$ and the expected loss $\mathcal{L}_{\mathcal{D}_u}(\Theta, W)$ are defined as:

$$\mathcal{L}_{S_m}(\Theta, W) = \frac{1}{m} \sum_{i=1}^m \ell(y_i, \hat{y}_i|_{\Theta, W}), \quad \mathcal{L}_{\mathcal{D}_u}(\Theta, W) = \frac{1}{n - m} \sum_{i=m+1}^n \ell(y_i, \hat{y}_i|_{\Theta, W}).$$

Following (El-Yaniv & Pechyony, 2006), testing datasets are randomly sampled from real data and we treat the loss on these testing datasets as the expected loss. Given that the optimal parameters $\Theta^*, W^*$ minimize the empirical loss $\mathcal{L}_{S_m}(\Theta^*, W^*)$, our goal is to bound the generalization error:

$$\mathcal{L}_{\mathcal{D}_U}(\Theta^*, W^*) - \mathcal{L}_{S_m}(\Theta^*, W^*).$$

A small generalization error bound indicates that spectral GNNs can perform well on testing data.

### 3.1 ASSUMPTIONS

We first introduce assumptions used in the generalization analysis of spectral GNNs.

**Assumption 1** (Lipschitz Continuity and Smoothness). The loss function $\ell$ and the spectral GNN $\Psi$ are both Lipschitz continuous and smooth.

Assumption 1 is commonly used in the analysis of neural networks trained with gradient descent (Ghadimi & Lan, 2013). They are necessary conditions ensuring that the neural network training converges (Arfken et al., 2011; Liao et al., 2021). We use $\text{Lip}(\bullet)$ and $\text{Smt}(\bullet)$ to denote the *Lipschitz* constant and smoothness of a function, respectively, and $\| \bullet \|_F$ denotes the Frobenius norm.

**Assumption 2** (Bounded Gradients). The gradients of both the spectral GNN and the loss function $\ell(y_i, \hat{y}_i)$ w.r.t. any parameters $\Theta, W$, and for any node $v_i$ with class $y_i$ and prediction $\hat{y}_i$, are bounded:

$$\|\nabla_\Theta \hat{y}_i\|_F^2 + \|\nabla_W \hat{y}_i\|_F^2 \leq \beta_2^2; \quad \|\nabla_{\hat{y}_i} \ell(y_i, \hat{y}_i)\|_F \leq \beta_1. \tag{2}$$

Here, $\nabla_\Theta \hat{y}_i$ and $\nabla_W \hat{y}_i$ represent the gradients of $\hat{y}_i$ with respect to the parameters $\Theta$ and $W$, respectively. $\nabla_{\hat{y}_i} \ell(y_i, \hat{y}_i)$ refers to the gradient of the loss function with respect to $\hat{y}_i$.

*Remark.* The loss surface of neural networks often contains many local minima (Dinh et al., 2017). The bounded gradient assumption ensures convergence during optimization (Li & Liu, 2021).

Unlike margin loss, the cross-entropy loss function is unbounded. For theoretical analysis, it is typically assumed that the sample loss is bounded.

**Assumption 3** (Bounded Sample Loss). For any sample $(x_i, y_i)$, the maximum loss between the ground truth class $y_i$ and the prediction $\hat{y}_i$ satisfies $\ell(\hat{y}_i, y_i) \leq B_\ell$.

Following the previous work (Zhang et al., 2019; Kuzborskij & Lampert, 2018), we assume that parameter norms are bounded during training.

**Assumption 4** (Bounded Parameters). Each parameter $\theta_k \in \Theta$ and the parameter matrix $W$ in a spectral GNN are bounded, i.e., $|\theta_k| \leq B_\Theta$ and $\|W\|_F \leq B_W$.

## 4 GENERAL RESULTS

In this section, we examine the generalization of spectral GNNs through *uniform transductive stability* (El-Yaniv & Pechyony, 2006). We define $\gamma$-uniform transductive stability for spectral GNNs, analyze the key factors influencing $\gamma$ by deriving the gradient norm bound, and use these insights to establish the generalization error bound.

**Definition 5** ($\gamma$-Uniform Transductive Stability). Let $S_m^{ij} = (X, \{y_t\}_{t=1, t\neq i}^m \cup \{y_j\})$ be a perturbed dataset obtained by replacing the $i$-th sample in $S_m$ with the $j$-th sample from $\mathcal{D}_u$. Let $\{\Theta^*, W^*\}$ and $\{\Theta', W'\}$ be the optimal parameters of a spectral GNN $\Psi$ trained on $S_m$ and $S_m^{ij}$, respectively. Denote the predictions for node $v_i$ by $\hat{y}_i|_{\Theta^*, W^*}$ and $\hat{y}_i|_{\Theta', W'}$. The spectral GNN $\Psi$ is $\gamma$-*uniform transductive stable* if for any $i \in [n]$:

$$\max_{1 \leq i \leq n} |\ell(y_i, \hat{y}_i|_{\Theta^*, W^*}) - \ell(y_i, \hat{y}_i|_{\Theta', W'})| \leq \gamma.$$

A larger $\gamma$ indicates worse stability of spectral GNNs. Below, we decompose $\gamma$ into two terms: $r$ and $\beta$. Here, $r$ accounts for the Lipschitz continuity and smoothness of the loss function and spectral GNNs, while $\beta$ bounds the gradient norm during training.

**Theorem 6** (Stability and Gradient Norm). *Let $\Psi$ be a spectral GNN trained using gradient descent for $T$ iterations with a learning rate $\eta$ on a training dataset $S_m$, and evaluated on a testing set $\mathcal{D}_u$. Under Assumption 1, for all iterations $t \in [1, T]$ and any sample $(x_i, y_i)$ in $S_m$ or $\mathcal{D}_u$, if the gradient norm satisfies $\|\nabla \ell(y_i, \hat{y}_i|_{\Theta^t, W^t})\|_F \leq \beta$, where $\{\Theta^t, W^t\}$ are the parameters at the $t$-th iteration, then $\Psi$ satisfies $\gamma$-uniform transductive stability with:*

$$\gamma = r\beta, \quad r = \frac{2\eta\alpha_1}{m} \sum_{t=1}^T (1 + \eta\alpha_2)^{t-1},$$

*where $\alpha_1 = Lip(\ell) \cdot Lip(\Psi)$ and $\alpha_2 = Smt(\Psi)\beta_1 + Smt(\ell)Lip(\Psi)\beta_2$.*

*Remark.* $\alpha_1$ and $\alpha_2$ represent the Lipschitz constant and smoothness of the loss function $\ell$ and the spectral GNN $\Psi$, respectively. They determine how parameter updates $\Theta^t$ and $W^t$ during training affect the loss of a sample $(x_i, y_i)$ through the term $r$. As described in Eq. (2), $\beta_1$ and $\beta_2$ are the bounds for the loss and its gradient, respectively. The proof is provided in Appendix A.2.

Unlike previous work (Cong et al., 2021), which assumes a fixed gradient norm $\beta$, we explicitly derive $\beta$ to analyze how graph homophily and the polynomial order of spectral GNNs influence the gradient norm and, in turn, stability. To begin, we introduce the concept of a walk on a graph and its expectation, both critical for analyzing the stability of spectral GNNs. A $k$-*length walk* on a graph $G$ is defined as a sequence of $k$ edges $\{e_1, e_2, \ldots, e_k\}$, where $e_i \in E$, and the endpoint of $e_i$ is the starting point of $e_{i+1}$ for $i \in \{1, \ldots, k-1\}$. The expectation of $k$-length walks is defined as follows.

**Definition 7** ($\mathbb{E}[A_{ij}^k]$). For a graph $G \sim cSBM(n, f, \Pi, Q)$ with adjacency matrix $A$, the expectation of the element $A_{ij}^k$ in the $k$-th power of $A$ is:

$$\mathbb{E}[A_{ij}^k] = \sum_{p \in P_{ij}^k} \prod_{(v,v') \in p} Q_{yy'},$$

where $P_{ij}^k$ is the set of all $k$-length walks between $v_i$ and $v_j$, and $Q_{yy'}$ is the transition probability between the classes $y$ of $v$ and $y'$ of $v'$.

*Remark.* Intuitively, $A_{ij}^k$ represents the number of $k$-length walks between nodes $v_i$ and $v_j$. The first moment $\mathbb{E}[A_{ij}^k]$ gives the expected number of such walks in the random graph generated by the cSBM. Since $Q_{yy'}$ represents the probability of an edge between nodes of classes $y$ and $y'$, a larger $Q_{yy'}$ increases the likelihood of edges in walks involving transitions between these classes, resulting in higher expected counts of such walks.

The following theorem reveals how the gradient norm bound $\beta$ depends on the expectation of $k$-length walks and their node class distributions within the graph.

**Theorem 8.** *Consider a spectral GNN $\Psi$ with polynomial order $K$ trained using full-batch gradient descent for $T$ iterations with a learning rate $\eta$ on a training dataset $S_m$ sampled from a graph $G \sim cSBM(n, f, \Pi, Q)$ with average node degree $d \ll n$. When $n \to \infty$ and $K \ll n$, under Assumptions 1, 2, and 4, for any node $v_i$, $i \in [n]$, and for a constant $\epsilon \in (0, 1)$, with probability at least $1 - \epsilon$, $\Psi$ satisfies $\gamma$-uniform transductive stability, where $\gamma = r\beta$ and*

$$\beta = \frac{1}{\epsilon} \left[ O\left(\mathbb{E}\left[\|\hat{y}_i - y_i\|_F^2\right]\right) + O\left(\|\pi_{y_i}^\top \pi_{y_i} + \Sigma_{y_i}\|_F\right) \right.$$

$$\left. + O\left(\sum_{k=1}^K \sum_{j=1}^n \mathbb{E}[A_{ij}^k] \left\|\sum_{t=1}^n \mathbb{E}[A_{it}^k]\pi_{y_j}^\top \pi_{y_t} + \mathbb{E}[A_{ij}^k]\Sigma_{y_j}\right\|_F\right) \right].$$

*Proof sketch.* The main idea is to first derive the gradient of a sample loss and the expected gradient norm $\mathbb{E}[\|\nabla \ell(y_i, \hat{y}_i|_{\Theta^t, W^t})\|_F]$ for node $(x_i, y_i)$ with fixed parameters $(n, f, \Pi, Q)$. Then the gradient norm bound $\beta$ is derived using Markov's Inequality (Evans & Rosenthal, 2004). When parameters $\Theta, W$ are bounded (Assumption 4), the main factors in $\mathbb{E}[\|\nabla \ell(y_i, \hat{y}_i|_{\Theta^t, W^t})\|_F]$ are $\|\hat{y}_i - y_i\|_F^2$ and moments of $A^k X$. When graph structure and node features are independent given node labels, we have $\mathbb{E}[A^k X] = \mathbb{E}[A^k] \mathbb{E}[X]$. While $A^k$ depends on the graph structure, governed by the parameter $Q$, node feature $X$ is controlled by parameter $\Pi$, shown as the mean $\pi_{y_i}$ and variance $\Sigma_{y_i}$ of nodes belong to class $y_i$. The full proof is provided in Appendix B.2. $\square$

*Remark.* Theorem 8 shows that the gradient norm bound $\beta$ is primarily influenced by two factors:

(1) **Expected prediction error** $\mathbb{E}[\|\hat{y}_i - y_i\|_F^2]$ quantifies the difference between the truth node class $y_i$ and the predicted label $\hat{y}_i$ for a node $v_i$ by a spectral GNN. A well-performing spectral GNN is characterized by a low expected prediction error.

(2) **Expectation of $k$-length walks** $\mathbb{E}[A_{ij}^k]$ measures the interaction between nodes $v_i$ and $v_j$ through $k$-length walks. The existence of these walks depends on the labels of all nodes along the walk, with edge probabilities in $k$-length walks determined by $Q$ in cSBM.

When a $\gamma$-transductive learning algorithm is trained on two nearly identical training sets, differing by just one sample, the expected generalization error equals the expected increase in sample loss (El-Yaniv & Pechyony, 2006). Based on this known result, Theorem 6, and $\beta$ obtained from Theorem 8, we have the following generalization error bound for spectral GNNs.

**Theorem 9** (Generalization Error Bound). *Let $H_2(n) \triangleq \sum_{i=1}^n \frac{1}{i^2}$ and $\Omega(m, n - m) \triangleq (n - m)^2 (H_2(n) - H_2(n - m))$. For $\epsilon \in (0, 1)$, if a spectral GNN is $\gamma$-uniform transductive stability with probability $1 - \epsilon$, then under Assumption 3, for $\delta \in (0, 1)$, with probability at least $(1 - \delta)(1 - \epsilon)$, the generalization error $\mathcal{L}_{\mathcal{D}_u}(\Theta, W) - \mathcal{L}_{S_m}(\Theta, W)$ is upper-bounded by:*

$$\gamma + \left(2\gamma + \left(\frac{1}{n - m} + \frac{1}{m}\right)(B_\ell - \gamma)\right)\sqrt{2\Omega(m, n - m)\log\frac{1}{\delta}}. \qquad (3)$$

*Remark.* The generalization error bound of a spectral GNN is closely related to its uniform transductive stability $\gamma$, the number of training samples $m$, and the total number of nodes $n$ in the graph. A smaller $\gamma$ indicates stronger stability, which in turn suggests better generalization performance. Factors such as graph homophily and the polynomial order of a spectral GNN affect $\gamma$, thereby impacting the generalization error. From Theorem 6, we observe that $\gamma = O(1/m)$. When $n$ is sufficiently large, the term $1/(n - m)$ becomes negligible, and $\Omega(m, n - m)$ increases as $O(m^{1/2})$.

The following lemma shows that increasing the number of training samples $m$ improves the generalization. The proof is provided in Appendix C.2.

**Lemma 10.** *Consider a spectral GNN trained with $m$ samples as $n \to \infty$. As the sample size $m$ increases, the generalization error bound decreases at the rate $O(1/m) + O\big(\sqrt{2\log(1/\delta)/m}\big)$.*

Thus far, we have considered only linear feature transformation functions $f_W$ in spectral GNNs. We now establish a result on how a non-linear $f_W$ influences uniform transductive stability.

**Proposition 11.** *For a spectral GNN $\Psi_{\tilde{\sigma}}$ with a non-linear feature transformation function $f_W(X) = \tilde{\sigma}(XW)$, assume the gradient norm bound $\beta$ in Theorem 9 is the same for $\Psi$ and $\Psi_{\tilde{\sigma}}$. If $Lip(\tilde{\sigma}) \leq 1$ and $Smt(\tilde{\sigma}) \leq 1$, then $\gamma_{\tilde{\sigma}} \leq \gamma$, where $\gamma_{\tilde{\sigma}}$ is the stability of $\Psi_{\tilde{\sigma}}$.*

*Remark.* The $\gamma_{\tilde{\sigma}}$-uniform transductive stability of spectral GNNs $\Psi_{\tilde{\sigma}}$ with a non-linear transformation is bounded by the stability of its linear counterpart $\Psi$, provided that the activation function satisfies $Lip(\tilde{\sigma}) \leq 1$ and $Smt(\tilde{\sigma}) \leq 1$. These conditions ensure that the non-linear transformation does not excessively amplify inputs or gradients, thus maintaining stability. Common activation functions like ReLU, Sigmoid, and Tanh satisfy these criteria, indicating that they may contribute to stabilizing the training of spectral GNNs by preventing large output fluctuations in response to small input changes. The proof is provided in Appendix C.2.

## 5 FURTHER ANALYSIS

In this section, we analyze the impact of node class distribution and spectral GNN architecture on the generalization error bound. To derive an explicit form for property analysis, we consider $cSBM(n, f, \mu, u, \lambda, d)$, a well-studied specialization of the general multi-class cSBM (Deshpande et al., 2018), widely used in prior studies on graph analysis (Esser et al., 2021; Ma et al., 2022; Baranwal et al., 2021; Baranwal et al.). Specifically, for a node $v_i$ with label $y_i \in \{\pm 1\}$, its feature is sampled from a Gaussian distribution:

$$x_i \sim \mathcal{N}(y_i \sqrt{\mu/n} u, I_f/f).$$

Two nodes of the same class are connected with probability $c_{\text{in}} = d + \lambda\sqrt{d}$, while nodes of different classes are connected with probability $c_{\text{out}} = d - \lambda\sqrt{d}$. In this simplified 2-class cSBM, the distribution $\Pi$ reduces to $\{\Pi_{\pm}\}$, and the edge-probability matrix simplifies to a $2 \times 2$ matrix with diagonal elements $c_{\text{in}}/n$ and off-diagonal elements $c_{\text{out}}/n$.

By adjusting the parameter $\lambda$ in cSBM, we can generate graphs with varying node class distributions. One way to quantify the node class distribution is the edge homophilic ratio (Zhu et al., 2020),

$$H_{edge} = \frac{|\{e_{ij} \mid v_i, v_j \in V, e_{ij} \in E, y_i = y_j\}|}{|E|}.$$

The relationship between the parameters $d, \lambda$ in cSBM and the edge homophilic ratio is as follows.

**Proposition 12.** *For a graph $G \sim cSBM(n, \mu, u, \lambda, d)$, the expected edge homophily ratio is:*

$$\mathbb{E}[H_{edge}] = \frac{d + \lambda\sqrt{d}}{2d}; \quad \mathbb{E}[H_{edge}] = \frac{c_{in}}{c_{in} + c_{out}}. \tag{4}$$

When $\lambda > 0$, the graph tends to be homophilic as $\mathbb{E}[H_{edge}] > 0.5$. Conversely, when $\lambda < 0$, the graph tends to be heterophilic. The proof is provided in Appendix E.1.

### 5.1 UNIFORM TRANSDUCTIVE STABILITY

We now establish stability for graphs with two node classes and Gaussian-distributed node features.

**Theorem 13.** *Consider a spectral GNN $\Psi$ parameterized by $\Theta, W$ trained using full-batch gradient descent for $T$ iterations with a learning rate $\eta$ on a training dataset containing $m$ samples drawn from nodes on a graph $G \sim cSBM(n, f, \mu, u, \lambda, d)$. When $n \to \infty$, $k \ll n$, and $d \ll n$, under Assumptions 1, 2, and 4, for any node $v_i$ on the graph, with probability at least $1 - \epsilon$ for a constant $\epsilon \in (0, 1)$, $\Psi$ satisfies $\gamma$-uniform transductive stability, where $\gamma = r\beta$ and*

$$\beta = \frac{1}{\epsilon}\left[O\left(\mathbb{E}\left[\|\hat{y}_i - y_i\|_F^2\right]\right) + O\left(\sum_{k=2}^{K}\left(\mathbb{E}\left[\left(A_{ij}^k \mid y_i = y_j\right)^2\right] + \mathbb{E}\left[\left(A_{ij}^k \mid y_i \neq y_j\right)^2\right]\right)\right)\right].$$

*Proof sketch.* The proof follows the same structure as Theorem 8. The gradient norm bound $\beta$ can be explicitly expressed as the expected prediction error $\mathbb{E}\left[\|\hat{y}_i - y_i\|_F^2\right]$ and the homophily-aware walk variance $\zeta_k = \mathbb{E}\left[(A_{ij}^k \mid y_i = y_j)^2\right] + \mathbb{E}\left[(A_{ij}^k \mid y_i \neq y_j)^2\right]$. The connection between $\zeta_k$ and $H_{edge}$ can be analyzed in a tractable manner. The full proof is provided in Appendix D.3. $\qquad\square$

*Remark.* The theorem derives the explicit form of $k$-length walks $A_{ij}^k$. Notably, $(A_{ij}^k|y_i = y_j)$ and $(A_{ij}^k|y_i \neq y_j)$ follow distinct distributions based on whether nodes $v_i$ and $v_j$ share the same label. When $k = 1$, $A_{ij}^k$ follows a Bernoulli distribution. For $n \to \infty$ with $d \ll n$ and $2 \leq k \ll n$, $A_{ij}^k$ follows a Poisson distribution. The term $\zeta_k = \mathbb{E}[(A_{ij}^k|y_i = y_j)^2] + \mathbb{E}[(A_{ij}^k|y_i \neq y_j)^2]$ captures the *homophily-aware walk variance*, reflecting the variance in $k$-length walks between same-class or different-class nodes. This depends on the edge probabilities $c_{in}$ and $c_{out}$: (1) $c_{in} = c_{out}$: the graph is essentially an Erdős-Rényi graph, lacking clusters or multipartite structure, leading to higher variance in $k$-length walks. (2) $c_{in} > c_{out}$: the graph is homophilic with cluster patterns, and walks are concentrated within clusters, reducing variance. (3) $c_{in} < c_{out}$: the graph is heterophilic with multipartite patterns, and walks are concentrated along edges connecting different classes, affecting the variance. In general, the absence of clear cluster or multipartite structures increases randomness in $k$-length walks, resulting in higher $\zeta_k$.

## 5.2 MAIN FACTORS IN STABILITY

We first analyze how exactly the expected prediction error $\mathbb{E}[\|\hat{y}_i - y_i\|_F^2]$ and the homophily-aware walk variance $\zeta_k$ vary with the parameters $\lambda$ and $K$, and then examine the combined effects of $\lambda$ and $K$ on the stability and generalization of spectral GNNs.

**Theorem 14** ($\mathbb{E}\left[\|\hat{y}_i - y_i\|_F^2\right]$ and $\lambda, K$)**.** *Given a graph $G \sim cSBM(n, \mu, u, \lambda, d)$ and a spectral GNN of order $K$, $\mathbb{E}[\|\hat{y}_i - y_i\|_F^2]$ for any node $v_i$ satisfies the following: it increases with $\lambda \in [-\sqrt{d}, 0]$, decreases with $\lambda \in [0, \sqrt{d}]$, and reaches its maximum at $\lambda = 0$; it increases with $K$ if $\sum_{k=2}^{K} \theta_k \frac{(k-1)!}{2^{k-1}}$ grows more slowly than $\sum_{k=2}^{K} \theta_k^2 \frac{(k-1)!}{2^k}$ as $K$ increases.*

*Remark.* When $\lambda = 0$, the graph is neither homophilic nor heterophilic, resulting in the maximum expected error. When $\lambda = \pm\sqrt{d}$, the expected error is minimized. This implies that spectral GNNs perform well on strong homophilic or heterophilic graphs but poorly on graphs that are neither. The relationship between the expected norm $\mathbb{E}[\|\hat{y}_i - y_i\|_F^2]$ and the order $K$ is nonetheless intricate, depending on $\Theta = \{\theta_k\}_{k=0}^K$. The proof is provided in Appendix E.2.

We observe that $\zeta_k$ exhibits the same trend as $\mathbb{E}[\|\hat{y}_i - y_i\|_F^2]$ with respect to changes in $\lambda$; however, their behavior diverges with respect to $K$, as characterized in the following theorem.

**Theorem 15** ($\zeta_k$ and $\lambda, K$)**.** *Given a graph $G \sim cSBM(n, \mu, u, \lambda, d)$ and a spectral GNN of order $K$, $\zeta_k$ has the following properties: (1) it increases with $\lambda \in [-\sqrt{d}, 0]$, decreases with $\lambda \in [0, \sqrt{d}]$, and achieves its maximum value at $\lambda = 0$; (2) it increases with $k$ as $k$ grows, for $k \in [0, K]$.*

*Remark.* When $d$ is fixed, $\lambda \to \sqrt{d}$, nodes with the same class form clusters, and when $\lambda \to -\sqrt{d}$, they form a bipartite structure. In both cases, the graph structure exhibits clear patterns, leading to a small variance $\mathbb{V}\left[A_{ij}^k\right] = \mathbb{E}\left[(A_{ij}^k)^2\right] - (\mathbb{E}\left[A_{ij}^k\right])^2$ and, consequently, a small $\zeta_k$. When $\lambda \to 0$, the graph lacks simple patterns, resulting in a large variance and a correspondingly large $\zeta_k$. When $k \in [0, K]$ increases, more walks between two nodes exist and thus the variance $\mathbb{V}\left[A_{ij}^k\right]$ increases. Larger variance corresponds to a larger $\zeta_k$. The proof is provided in Appendix E.3.

Based on Theorems 14 and 15, the following proposition summarizes how $\lambda$ and $K$ influence the $\gamma$-uniform transductive stability of spectral GNNs. The proof is provided in Appendix E.4.

**Proposition 16.** *For a fixed $K$, $\gamma$-uniform transductive stability and generalization error bound strictly increase as $\lambda$ moves from $-\sqrt{d}$ to 0, and decreases as $\lambda$ moves from 0 to $\sqrt{d}$. For a fixed $\lambda$, if $\sum_{k=2}^{K} \theta_k \frac{(k-1)!}{2^{k-1}}$ grows more slowly than $\sum_{k=2}^{K} \theta_k^2 \frac{(k-1)!}{2^k}$ as $K$ increases, then $\gamma$-uniform transductive stability and generalization error bound increase with $K$.*

## 5.3 PRACTICAL IMPLICATIONS

We discuss two practical implications of our theoretical findings.

**Rewiring graphs**: Our analysis establishes a strong connection between graph homophily and the generalization error bound, offering practical insights for rewiring graphs to enhance the performance

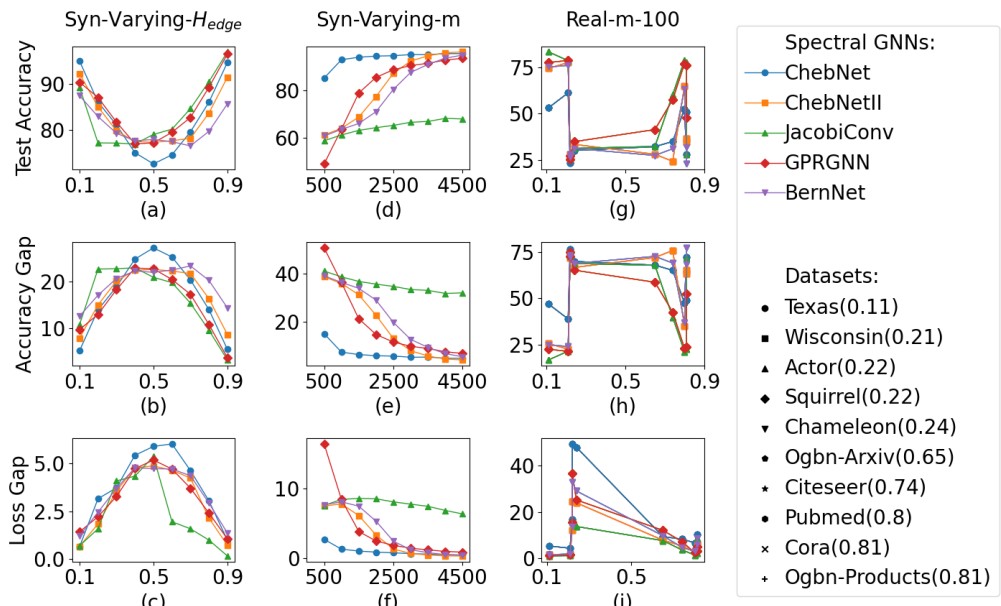

Figure 1: Testing accuracy, accuracy gap, and loss gap of five spectral GNNs on synthetic and real-world datasets: **(a)-(c)** *Syn-Varying-$H_{edge}$*: synthetic datasets with varying $H_{edge} \in [0.1, 0.9]$ (step 0.1) and $m = 3,000$; **(d)-(f)** *Syn-Varying-m*: synthetic datasets with varying training sample sizes $m \in [500, 4500]$ (step 500) of fixed $H_{edge}$; **(g)-(i)** *Real-m-100*: real-world datasets with varying $H_{edge}$ (shown on the right) and $m = 100$.

of spectral GNNs. Graphs with strong homophilic or heterophilic structures exhibit low variance in $k$-length walks, which reveals clearer structural patterns. This reduction in variance decreases the gradient norm bound $\beta$ (Theorems 8 and 13), leading to improved $\gamma$-uniform transductive stability (Theorem 6). Enhanced stability, in turn, reduces the generalization error bound (Theorem 9).

**Constrained graph convolution**: Our theoretical results indicate that constraining the graph convolution parameters $0 \le \theta_k \le 1$ prevents the generalization error bound from increasing with the polynomial order $K$. This is because the constraint ensures that the condition in Proposition 16, where $\sum_{k=2}^K \theta_k \frac{(k-1)!}{2^{k-1}}$ increases slower than $\sum_{k=2}^K \theta_k^2 \frac{(k-1)!}{2^k}$, is violated, as $\theta_k \ge \theta_k^2$. Previous work (He et al., 2021) reports that constraining $\theta_k$ to non-negative values with Bernstein polynomial basis leads to valid polynomial filters. Our analysis further suggests adding the constraint $\theta_k \le 1$ to maintain stable generalization error as $K$ increases.

## 6 EXPERIMENTS

**Synthetic and real-wrold datasets.** We use the following datasets: (1) *Synthetic datasets* consist of nine graphs generated using $G \sim cSBM(n, f, \mu, u, \lambda, d)$ following (Chien et al., 2021). Each graph contains 5,000 nodes with 2,000-dimensional features and an average degree of 5. The edge homophily ratios ($H_{edge}$) range from 0.1 to 0.9 in steps of 0.1. (2) *Real-world datasets* consist of ten benchmark node classification datasets (Texas, Wisconsin, Actor, Chameleon, Squirrel, Citeseer, Pubmed, Cora, Ogbn-Arxiv, Ogbn-Products) with $H_{edge}$ varying between 0.11 and 0.81. Following previous work (He et al., 2021; 2022a; Chien et al., 2021), we randomly split each dataset into 60% for training, 20% for validation, and 20% for testing.

**Spectral GNNs.** We select five widely recognized spectral GNNs for our experiments: ChebNet (Defferrard et al., 2016), GPRGNN (Chien et al., 2021), BernNet (He et al., 2021), JacobiConv (Wang & Zhang, 2022), and ChebNetII (He et al., 2022a). For consistency with Eq. (1), we use a single-layer ChebNet rather than the typical two-layer version.

Further details about the dataset statistics, spectral GNNs, hyper-parameter settings used in our experiments and additional experimental results are provided in Appendix F. Below, we discuss the effects of node class distribution and polynomial order on the accuracy and loss gaps of spectral GNNs.

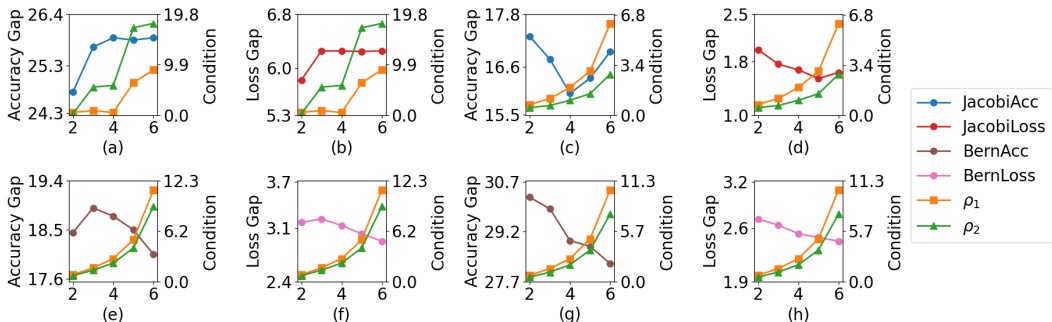

Figure 2: Accuracy gap and loss gap of JacobiConv and BernNet when the polynomial order $K$ increases, where $\rho_1 = \sum_{k=2}^{K} \theta_k \frac{(k-1)!}{2^{k-1}}$ and $\rho_2 = \sum_{k=2}^{K} \theta_k^2 \frac{(k-1)!}{2^k}$: (a-b),(e-f) show the results on a synthetic dataset of $H_{edge} = 0.2$; (c-d),(g-h) show the results on Chameleon dataset of $H_{edge} = 0.24$.

Figure 1(a)-(c) illustrates that as $H_{edge}$ of synthetic datasets varies from 0.1 to 0.9, accuracy and loss gaps increase monotonically for $H_{edge} \in [0.1, 0.5]$ and decrease for $H_{edge} \in [0.5, 0.9]$, reaching their maximum at $H_{edge} = 0.5$ across all spectral GNNs. This aligns with our theoretical analysis in Proposition 16, which states that the generalization error bound increases as $H_{edge}$ moves from 0 to 0.5 and decreases as $H_{edge}$ moves from 0.5 to 1. In Figure 1(d)-(f), when the training sample number $m$ increases from 500 to 4,500 on synthetic datasets, both accuracy and loss gaps decrease consistently. This is consistent with Lemma 10, which shows that the generalization error bound decreases with increasing $m$. Figure 1(g)-(i) shows that when training all real-world datasets with the same sample size $m = 100$, spectral GNNs exhibit a similar trend to synthetic datasets. Loss gaps are small at the extreme ends of the $H_{edge}$ range but increase as $H_{edge}$ transitions from Texas (0.11) to Ogbn-Products (0.81). These results align with Proposition 16.

Figure 2 shows that when the slope of $\rho_1$ is smaller than that of $\rho_2$, accuracy and loss gaps increase. Conversely, gaps decrease when $\rho_1$ grows faster than $\rho_2$, aligning with Proposition 16. Intuitively, this condition can be understood as follows: (1). *Non-negative $\theta_k$*: Spectral GNNs constrained to $0 \le \theta_k \le 1$ demonstrate strong generalization. In this case, $\theta_k \ge \theta_k^2$, ensuring $\rho_1$ grows faster than $\rho_2$, violating the condition of Proposition 16, and preventing the generalization error bound from increasing with $K$. For instance, BernNet enforces non-negative $\theta_k$, and as shown in Figure 2(e-h), its accuracy and loss gaps remain stable with increasing $K$. (2). *Unrestricted $\theta_k$*: Spectral GNNs allowing both positive and negative $\theta_k$ may exhibit poor generalization. If $\theta_k < 0$, $\rho_1 \le \rho_2$. When $\theta_{k_1} \le 0$ and $\theta_{k_2} \ge 0$, $\rho_1$ typically grows slower than $\rho_2$, satisfying the condition of Proposition 16 and leading to increasing generalization error bounds with $K$. For example, JacobiConv does not restrict the sign of $\theta_k$, and its accuracy and loss gaps increase with $K$ in Figure 2(a-b).

# 7 CONCLUSION, LIMITATIONS, AND FUTURE WORK

This work investigates how node class distribution and architectural choices impact the generalization of spectral GNNs. Our findings show that spectral GNNs generalize well on graphs with strong homophilic or heterophilic structures, where node class distributions exhibit clear patterns, and the generalization error of spectral GNNs increases with polynomial order under certain conditions.

We derive the uniform transductive stability of spectral GNNs on graphs generated by the general multi-class cSBM, providing insights into the relationships between graph homophily, polynomial order, and generalization error bounds. However, this analysis is limited to a specialized cSBM, leaving room for further exploration of more diverse graph generation models. Another limitation lies in architectural choices. While these choices, such as the selection of the graph matrix (e.g., Laplacian vs. adjacency matrix) and polynomial basis (e.g., Chebyshev vs. Bernstein), are critical to generalization performance, we do not explore their specific impacts on generalization bounds. Future work could investigate how these design decisions influence the theoretical and practical performance of spectral GNNs. Finally, our theoretical analysis assumes training with gradient descent, whereas Adam is the optimizer most commonly used in practice. This discrepancy between theoretical assumptions and practical applications highlights an important direction for future research to bridge the gap and improve the relevance of theoretical findings to real-world scenarios.

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

# CONTENTS (APPENDIX)

## A  STABILITY AND GRADIENT

### A.1  LEMMAS FOR THEOREM 6

We start by establishing the maximum variation in the sample loss and the maximum change in the gradient of the loss function with respect to the parameters $\{\Theta, W\}$ of spectral GNNs, as defined in Eq. (1). These two properties play a crucial role in the subsequent analysis.

Based on Assumption 1, we derive the following lemmas.

**Lemma 17** (Bound of Loss function to Parameters). *Under Assumption 1, given a loss function $\ell$ and a spectral GNN, for parameters $\bar{\Theta}, \bar{W}, \Theta', W'$ and any node $v_i$ with truth class $y_i$ we have*

$$\|\ell(y_i, \hat{y}_i|_{\Theta=\bar{\Theta}, W=\bar{W}}) - \ell(y_i, \hat{y}_i|_{\Theta', W'})\|_F \leq \alpha_1 \sqrt{\|\bar{\Theta} - \Theta'\|_F^2 + \|\bar{W} - W'\|_F^2}$$

*where $\alpha_1 = Lip(\ell)Lip(\Psi)$.*

*Proof.* Under Assumption 1, we have:

$$\|\ell(y_i, \hat{y}_i|_{\tau=\bar{\tau}}) - \ell(y_i, \hat{y}_i|_{\tau=\tau'})\| \leq Lip(\ell)\|\hat{y}_i|_{\tau=\bar{\tau}} - \hat{y}_i|_{\tau=\tau'}\|_F;$$

$$\|Lip(\ell)\|\hat{y}_i|_{\tau=\bar{\tau}} - \hat{y}_i|_{\tau=\tau'}\|_F \leq Lip(\Psi)\|\bar{\tau} - \tau'\|_F.$$

By combining the two inequalities above, we arrive at:

$$\|\ell(y_i, \hat{y}_i|_{\tau=\bar{\tau}}) - \ell(y_i, \hat{y}_i|_{\tau=\tau'})\| \leq Lip(\ell)Lip(\Psi)\|\bar{\tau} - \tau'\|_F.$$

$\square$

**Lemma 18** (Bound of Gradient to Parameters). *Under Assumption 1, Assumption 2, for parameters $\bar{\Theta}, \bar{W}, \Theta', W'$ of a spectral GNN, the following holds for any node $v_i$ with truth class $y_i$*

$$\|\nabla \ell(y_i, \hat{y}_i|_{\Theta=\bar{\Theta}, W=\bar{W}}) - \nabla \ell(y_i, \hat{y}_i|_{\Theta', W'})\|_F \leq \alpha_2 \sqrt{\|\bar{\Theta} - \Theta'\|_F^2 + \|\bar{W} - W'\|_F^2}$$

*where $\alpha_2 = (Smt(\Psi)\beta_1 + Smt(\ell)Lip(\Psi)\beta_2)$.*

*Proof.* Since we have

$$\nabla \ell(y_i, \hat{y}_i|_{\tau=\bar{\tau}}) = \nabla_{\hat{y}_i} \ell(y, \hat{y}_i)|_{\tau=\bar{\tau}} \cdot \nabla \hat{y}_i|_{\tau=\bar{\tau}};$$
$$\nabla \ell(y_i, \hat{y}_i|_{\tau=\tau'}) = \nabla_{\hat{y}_i} \ell(y, \hat{y}_i)|_{\tau=\tau'} \cdot \nabla \hat{y}_i|_{\tau=\tau'},$$

this leads to

$$\nabla \ell(y_i, \hat{y}_i|_{\tau=\bar{\tau}}) - \nabla \ell(y_i, \hat{y}_i|_{\tau=\tau'}) = \nabla_{\hat{y}_i} \ell(y, \hat{y}_i)|_{\tau=\bar{\tau}}(\nabla \hat{y}_i|_{\tau=\bar{\tau}} - \nabla \hat{y}_i|_{\tau=\tau'})$$
$$+ (\nabla_{\hat{y}_i} \ell(y, \hat{y}_i)|_{\tau=\bar{\tau}} - \nabla_{\hat{y}_i} \ell(y, \hat{y}_i)|_{\tau=\tau'}) \nabla \hat{y}_i|_{\tau=\tau'}.$$

Hence, we obtain the following

$$\|\nabla \ell(y_i, \hat{y}_i|_{\tau=\bar{\tau}}) - \nabla \ell(y_i, \hat{y}_i|_{\tau=\tau'})\|_F \leq \|\nabla_{\hat{y}_i} \ell(y, \hat{y}_i)|_{\tau=\bar{\tau}}\|_F \cdot \|\nabla \hat{y}_i|_{\tau=\bar{\tau}} - \nabla \hat{y}_i|_{\tau=\tau'}\|_F$$
$$+ \|\nabla_{\hat{y}_i} \ell(y, \hat{y}_i)|_{\tau=\bar{\tau}} - \nabla_{\hat{y}_i} \ell(y, \hat{y}_i)|_{\tau=\tau'}\|_F \cdot \|\nabla \hat{y}_i|_{\tau=\tau'}\|_F. \tag{5}$$

Under Assumption 1 and Assumption 2, we have:

$$\|\nabla \hat{y}_i|_{\tau=\bar{\tau}} - \nabla \hat{y}_i|_{\tau=\tau'}\|_F \leq Smt(\Psi)\|\bar{\tau} - \tau'\|_F$$
$$\|\nabla_{\hat{y}_i} \ell(y, \hat{y}_i)|_{\tau=\bar{\tau}}\|_F \leq \beta_1. \tag{6}$$

Under Assumption 1, we have:

$$\|\nabla_{\hat{y}_i} \ell(y, \hat{y}_i)|_{\tau=\bar{\tau}} - \nabla_{\hat{y}_i} \ell(y, \hat{y}_i)|_{\tau=\tau'}\|_F \leq Smt(\ell)\|\hat{y}_i|_{\tau=\bar{\tau}} - \hat{y}_i|_{\tau=\tau'}\|_F$$
$$\leq Smt(\ell)Lip(\Psi)\|\bar{\tau} - \tau'\|_F. \tag{7}$$

Under Assumption 2, we have:

$$\|\nabla \hat{y}_i|_{\tau=\tau'}\|_F \leq \beta_2. \tag{8}$$

Substitute Eq. (6), Eq. (7), and Eq. (8) into Eq. (5), we have

$$\|\nabla \ell(y_i, \hat{y}_i|_{\tau=\bar{\tau}}) - \nabla \ell(y_i, \hat{y}_i|_{\tau=\tau'})\|_F \leq Smt(\Psi)\|\bar{\tau} - \tau'\|_F \cdot \beta_1 + Smt(\ell)Lip(\Psi)\|\bar{\tau} - \tau'\|_F \cdot \beta_2$$
$$= (Smt(\Psi)\beta_1 + Smt(\ell)Lip(\Psi)\beta_2) \|\bar{\tau} - \tau'\|_F.$$

$\square$

## A.2 PROOF OF THEOREM 6

**Theorem 6** (Stability and Gradient Norm). *Let $\Psi$ be a spectral GNN trained using gradient descent for $T$ iterations with a learning rate $\eta$ on a training dataset $S_m$, and evaluated on a testing set $\mathcal{D}_u$. Under Assumption 1, for all iterations $t \in [1, T]$ and any sample $(x_i, y_i)$ in $S_m$ or $\mathcal{D}_u$, if the gradient norm satisfies $\|\nabla \ell(y_i, \hat{y}_i|_{\Theta^t, W^t})\|_F \leq \beta$, where $\{\Theta^t, W^t\}$ are the parameters at the $t$-th iteration, then $\Psi$ satisfies $\gamma$-uniform transductive stability with:*

$$\gamma = r\beta, \quad r = \frac{2\eta \alpha_1}{m} \sum_{t=1}^{T} (1 + \eta \alpha_2)^{t-1},$$

*where $\alpha_1 = Lip(\ell) \cdot Lip(\Psi)$ and $\alpha_2 = Smt(\Psi)\beta_1 + Smt(\ell)Lip(\Psi)\beta_2$.*

*Proof.* We define $\tau = [\Theta; W]$ as the concatenation of the parameters $\Theta$ and $W$. From Lemma 17 and Lemma 18, we derive:

$$\|\ell(y_i, \hat{y}_i|_{\tau}) - \ell(y_i, \hat{y}_i|_{\tau'})\|_F \leq \alpha_1 \|\tau - \tau'\|_F;$$
$$\|\nabla \ell(y_i, \hat{y}_i|_{\tau}) - \nabla \ell(y_i, \hat{y}_i|_{\tau'})\|_F \leq \alpha_2 \|\tau - \tau'\|_F,$$

where $\alpha_1 = Lip(\ell)Lip(\Psi)$ and $\alpha_2 = (Smt(\Psi)\beta_1 + Smt(\ell)Lip(\Psi)\beta_2)$. The updating rule for gradient descent is given by:

$$\tau^{t+1} = \tau^t - \eta\nabla\mathcal{L}_{S_m}(\tau^t);$$
$$\tau_{ij}^{t+1} = \tau_{ij}^t - \eta\nabla\mathcal{L}_{S_m^{ij}}(\tau_{ij}^t),$$

where

$$\mathcal{L}_{S_m}(\tau^t) = \frac{1}{m}\sum_{r=1}^{m}\ell(y_r, \hat{y}_r|_{\tau^t}) \text{ and } \mathcal{L}_{S_m^{ij}}(\tau_{ij}^t) = \frac{1}{m}\sum_{r=1}^{m}\ell(y_r, \hat{y}_r|_{\tau_{ij}^t}).$$

represent the empirical loss on the training dataset $S_m$ and $S_m^{ij}$, respectively. The difference between the empirical losses is given by:

$$\mathcal{L}_{S_m^{ij}}(\tau_{ij}^t) - \mathcal{L}_{S_m}(\tau^t) = \frac{1}{m}\left[\sum_{r=1,r\neq i,j}^{m}\left(\ell(y_r, \hat{y}_r|_{\tau_{ij}^t}) - \ell(y_r, \hat{y}_r|_{\tau^t})\right) + \ell(y_j, \hat{y}_j|_{\tau_{it}^t}) - \ell(y_i, \hat{y}_i|_{\tau^t})\right].$$

We derive the parameter difference:

$$\left\|\tau_{ij}^{t+1} - \tau^{t+1}\right\|_F = \left\|\tau_{ij}^t - \eta\nabla\mathcal{L}_{S_m^{ij}}(\tau_{ij}^t) - \tau^t + \eta\nabla\mathcal{L}_{S_m}(\tau^t)\right\|_F$$

$$\leq \|\tau_{ij}^t - \tau^t\|_F + \eta\|\nabla(\mathcal{L}_{S_m}(\tau^t) - \mathcal{L}_{S_m^{ij}}(\tau_{ij}^t))\|_F$$

$$= \|\tau_{ij}^t - \tau^t\|_F + \frac{\eta}{m}\left\|\nabla\left[\sum_{\substack{r=1\\r\neq i,j}}^{m}\left(\ell(y_r, \hat{y}_r|_{\tau_{ij}^t}) - \ell(y_r, \hat{y}_r|_{\tau^t})\right) + \ell(y_j, \hat{y}_j|_{\tau_{ij}^t}) - \ell(y_i, \hat{y}_i|_{\tau^t})\right]\right\|_F$$

$$\leq \|\tau_{ij}^t - \tau^t\|_F + \frac{\eta}{m}\left\|\sum_{\substack{r=1\\r\neq i,j}}^{m}\alpha_2\|\tau_{ij}^t - \tau^t\|_F + \nabla\left[\ell(y_j, \hat{y}_j|_{\tau_{ij}^t}) - \ell(y_i, \hat{y}_i|_{\tau^t})\right]\right\|_F \quad (Assumption\ 1)$$

$$\leq \|\tau_{ij}^t - \tau^t\|_F + \frac{\eta}{m}(m-1)\alpha_2\|\tau_{ij}^t - \tau^t\|_F + \frac{\eta}{m}\left\|\nabla\left[\ell(y_j, \hat{y}_j|_{\tau_{ij}^t}) - \ell(y_i, \hat{y}_i|_{\tau^t})\right]\right\|_F$$

$$\leq \|\tau_{ij}^t - \tau^t\|_F + \frac{\eta}{m}(m-1)\alpha_2\|\tau_{ij}^t - \tau^t\|_F + \frac{2\eta\beta}{m} \quad (Theorem\ 13)$$

$$= \left(1 + \frac{m-1}{m}\eta\alpha_2\right)\|\tau_{ij}^t - \tau^t\|_F + \frac{2\eta\beta}{m}$$

$$\leq (1 + \eta\alpha_2)\|\tau_{ij}^t - \tau^t\|_F + \frac{2\eta\beta}{m}.$$

After $T$ iterations, we obtain

$$\left\|\tau_{ij}^T - \tau^T\right\|_F \leq (1 + \eta\alpha_2)\left\|\tau_{ij}^{T-1} - \tau^{T-1}\right\|_F + \frac{2\eta\beta}{m}$$

$$\leq (1 + \eta\alpha_2)[(1 + \eta\alpha_2)\left\|\tau_{ij}^{T-2} - \tau^{T-2}\right\|_F + \frac{2\eta\beta}{m}]$$

$$\leq (1 + \eta\alpha_2)^T\left\|\tau_{ij}^0 - \tau^0\right\|_F + \sum_{t=1}^{T}(1 + \eta\alpha_2)^{t-1}\frac{2\eta\beta}{m}$$

$$= \sum_{t=1}^{T}(1 + \eta\alpha_2)^{t-1}\frac{2\eta\beta}{m}.$$

Since the loss function $\ell$ is $\alpha_1$-Lipschitz continuous, for any sample $(x_i, y_i)$ with parameters $\tau^T = [\Theta^T; W^T]$ and $\tau_{ij}^T = [\Theta_{ij}^T; W_{ij}^T]$, we have:

$$\left|\ell(\hat{y}_i, y_i; \tau^T) - \ell(\hat{y}_i, y_i; \tau_{ij}^T)\right| \leq \alpha_1\left|\tau^T - \tau_{ij}^T\right|$$

$$\leq \alpha_1\sum_{t=1}^{T}(1 + \eta\alpha_2)^{t-1}\frac{2\eta\beta}{m}.$$

The proof is completed. $\qquad\square$

# B  STABILITY ON GENERAL MULTI-CLASS cSBM

We derive the uniform transductive stability of spectral GNNs defined in Eq. (1) on graphs generated by $G \sim cSBM(n, f, \Pi, Q)$. Then we discuss how the non-linear feature transformation function affect the stability.

We first give a brief introduction to inequalities and lemmas used in this proof.

## B.1  LEMMAS FOR THEOREM 8

**Lemma 19** (Jensen's Inequality). *Let $X$ be an arbitrary random variable, and let $f : \mathbb{R}^1 \to \mathbb{R}^1$ be a convex function such that $\mathbb{E}[f(X)]$ is finite. Then $f(\mathbb{E}[f(X)]) \leq \mathbb{E}[f(X)]$.*

**Lemma 20** (Markov's Inequality). *If $X$ is a non-negative random variable, then for all $a > 0$,*

$$P(X \geq a) \leq \frac{\mathbb{E}[X]}{a}.$$

*That is, the probability that $X$ exceeds any given value $a$ is no more than the expectation of $X$ divided by $a$.*

*Remark.* Lemma 19, Lemma 20 are important inequalities about a variable and its expectation. Details can be found in (Evans & Rosenthal, 2004).

**Lemma 21** (Cauchy-Schwarz Inequality (Arfken et al., 2011)).

$$(\sum_{k=1}^{n} a_k b_k)^2 \leq (\sum_{k=1}^{n} a_k^2)(\sum_{k=1}^{n} b_k^2).$$

The square of the $\ell_2$-norm of the product of two vectors is less than or equal to the product of the squares of the $\ell_2$-norms of the individual vectors.

**Lemma 22** (Trace and Frobenius Norm). *For any matrix $A \in \mathbb{R}^{n \times n}$, the relation between its trance and its Frobenius norm is*

$$Tr(A) \leq \sqrt{n} \cdot \|A\|_F.$$

*Proof.* The trace of $A$ is defined as:

$$\mathrm{Tr}(A) = \sum_{i=1}^{n} a_{ii}.$$

Applying the absolute value, we have:

$$\mathrm{Tr}(A) \leq \sum_{i=1}^{n} |a_{ii}|.$$

Using the Cauchy-Schwarz inequality (Lemma 21), this becomes:

$$\sum_{i=1}^{n} |a_{ii}| \leq \sqrt{n} \cdot \sqrt{\sum_{i=1}^{n} |a_{ii}|^2}.$$

Since $|a_{ii}|^2 = a_{ii}^2$, we can write:

$$\sqrt{\sum_{i=1}^{n} |a_{ii}|^2} = \sqrt{\sum_{i=1}^{n} a_{ii}^2}.$$

Thus:

$$\mathrm{Tr}(A) \leq \sqrt{n} \cdot \sqrt{\sum_{i=1}^{n} a_{ii}^2} = \sqrt{n} \cdot \|A\|_F.$$

$\square$

**Lemma 23** (Partial Derivatives). *For spectral graph neural networks defined as $\hat{Y} = \text{softmax}\left(\sum_{k=0}^{K}\theta_k\tilde{A}^kXW\right)$, with node feature matrix $X \in \mathbb{R}^{n\times f}$ and ground truth node label matrix $Y \in \mathbb{R}^{n\times C}$, the cross-entropy loss for a single sample $(x_i, y_i)$ is given by:*

$$\ell(\hat{y}_i, y_i; \Theta, W) = -\sum_{c=1}^{C} Y_{ic}\log\left(\hat{Y}_{ic}\right).$$

*The partial derivatives of $\ell(\hat{y}_i, y_i; \Theta, W)$ with respect to $\theta_k$ and $W_{pq}$ are:*

$$\frac{\partial\ell(\hat{y}_i, y_i; \Theta, W)}{\partial\theta_k} = \sum_{c=1}^{C}\left(\hat{Y}_{ic} - Y_{ic}\right)\left(\tilde{A}^kXW\right)_{ic},$$

$$\frac{\partial\ell(\hat{y}_i, y_i; \Theta, W)}{\partial W_{pq}} = \left(\hat{Y}_{iq} - Y_{iq}\right)\left(\sum_{k=0}^{K}\theta_k\tilde{A}^kX\right)_{ip}.$$

*Proof.* We begin with the following definitions:

$$Z = \sum_{k=0}^{K}\theta_k\tilde{A}^kXW, \quad \hat{Y}_{ic} = \frac{e^{Z_{ic}}}{\sum_{c'=1}^{C} e^{Z_{ic'}}}, \quad \ell(\hat{y}_i, y_i; \Theta, W) = -\sum_{c=1}^{C} Y_{ic}\log(\hat{Y}_{ic}),$$

where $Z \in \mathbb{R}^{n\times C}$ represents the feature matrix after aggregation, $\hat{Y}_{ic}$ is the softmax output for class $c$, and $\ell(\hat{y}_i, y_i; \Theta, W)$ is the cross-entropy loss for sample $(x_i, y_i)$. We then compute the following partial derivatives:

$$\frac{\partial\ell(\hat{y}_i, y_i; \Theta, W)}{\partial\hat{Y}_{ic}} = -\frac{Y_{ic}}{\hat{Y}_{ic}},$$

$$\frac{\partial\hat{Y}_{ic}}{\partial Z_{ic'}} = \hat{Y}_{ic}(\delta_{cc'} - \hat{Y}_{ic'}),$$

where $\delta_{cc'}$ is the Kronecker delta, which equals 1 if $c = c'$ and 0 otherwise.

(1) **Gradient w.r.t. $\theta_k$:** We have:

$$\frac{\partial Z_{ic}}{\partial\theta_k} = (\tilde{A}^kXW)_{ic}.$$

By the chain rule of gradient, we have:

$$\frac{\partial\ell(\hat{y}_i, y_i; \Theta, W)}{\partial\theta_k} = -\sum_{c=1}^{C}\frac{\ell(\hat{y}_i, y_i; \Theta, W)}{\partial\hat{Y}_{ic}}\cdot\left(\sum_{c'=1}^{C}\frac{\partial\hat{Y}_{ic}}{\partial Z_{ic'}}\cdot\frac{\partial Z_{ic'}}{\partial\theta_k}\right)$$

$$= -\sum_{c=1}^{C}\frac{Y_{ic}}{\hat{Y}_{ic}}\cdot\left(\sum_{c'=1}^{C}\hat{Y}_{ic}\left(\delta_{cc'} - \hat{Y}_{ic'}\right)\cdot\left(\tilde{A}^kXW\right)_{ic'}\right)$$

$$= -\sum_{c=1}^{C}Y_{ic}\cdot\left(\sum_{c'=1}^{C}\left(\delta_{cc'} - \hat{Y}_{ic'}\right)\cdot\left(\tilde{A}^kXW\right)_{ic'}\right)$$

$$= -\sum_{c=1}^{C}Y_{ic}\cdot\left(\left(\tilde{A}^kXW\right)_{ic} - \sum_{c'=1}^{C}\hat{Y}_{ic'}\left(\tilde{A}^kXW\right)_{ic'}\right)$$

$$= -\sum_{c=1}^{C}Y_{ic}\left(\tilde{A}^kXW\right)_{ic} + \sum_{c'=1}^{C}\hat{Y}_{ic'}\left(\tilde{A}^kXW\right)_{ic'}$$

$$= \sum_{c=1}^{C}\left(\hat{Y}_{ic} - Y_{ic}\right)\left(\tilde{A}^kXW\right)_{ic}$$

(2) **Gradient w.r.t.** $W$: Based on the following

$$Z_{ic} = \sum_{k=0}^{K} \theta_k \sum_{j=1}^{n} (\tilde{A}^k)_{ij} \sum_{r=1}^{f} X_{jr} W_{rc},$$

we have

$$\frac{\partial Z_{ic}}{\partial W_{pq}} = \sum_{k=0}^{K} \theta_k \sum_{j=1}^{n} (\tilde{A}^k)_{ij} X_{jp} \delta_{cq} = \delta_{cq} \sum_{k=0}^{K} \theta_k \left( \tilde{A}^k X \right)_{ip},$$

where $\delta_{cq}$ is the Kronecker delta, which is 1 if $c = q$ and 0 otherwise. Then, by the chain rule of gradient, we have:

$$\frac{\partial \ell(\hat{y}_i, y_i; \Theta, W)}{\partial W_{pq}} = -\sum_{c=1}^{C} \frac{\ell(\hat{y}_i, y_i; \Theta, W)}{\partial \hat{Y}_{ic}} \cdot \left( \sum_{c'=1}^{C} \frac{\partial \hat{Y}_{ic}}{\partial Z_{ic'}} \cdot \frac{\partial Z_{ic'}}{\partial W_{pq}} \right)$$

$$= -\sum_{c=1}^{C} \frac{Y_{ic}}{\hat{Y}_{ic}} \cdot \left( \sum_{c'=1}^{C} \hat{Y}_{ic} \left( \delta_{cc'} - \hat{Y}_{ic'} \right) \cdot \left( \delta_{c'q} \sum_{k=0}^{K} \theta_k \left( \tilde{A}^k X \right)_{ip} \right) \right)$$

$$= -\sum_{c=1}^{C} Y_{ic} \cdot \left( \sum_{c'=1}^{C} \left( \delta_{cc'} - \hat{Y}_{ic'} \right) \cdot \left( \delta_{c'q} \sum_{k=0}^{K} \theta_k \left( \tilde{A}^k X \right)_{ip} \right) \right)$$

$$= -\sum_{c=1}^{C} Y_{ic} \cdot \left( \left( \delta_{cq} \sum_{k=0}^{K} \theta_k \left( \tilde{A}^k X \right)_{ip} \right) - \sum_{c'=1}^{C} \hat{Y}_{ic'} \left( \delta_{c'q} \sum_{k=0}^{K} \theta_k \left( \tilde{A}^k X \right)_{ip} \right) \right)$$

$$= -\sum_{c=1}^{C} Y_{ic} \left( \delta_{cq} \sum_{k=0}^{K} \theta_k \left( \tilde{A}^k X \right)_{ip} \right) + \sum_{c'=1}^{C} \hat{Y}_{ic'} \left( \delta_{c'q} \sum_{k=0}^{K} \theta_k \left( \tilde{A}^k X \right)_{ip} \right)$$

$$= \sum_{c=1}^{C} \left( \hat{Y}_{ic} - Y_{ic} \right) \left( \delta_{cq} \sum_{k=0}^{K} \theta_k \left( \tilde{A}^k X \right)_{ip} \right)$$

$$= \sum_{c=1}^{C} \sum_{k=0}^{K} \theta_k \delta_{cq} \left( \hat{Y}_{ic} - Y_{ic} \right) \left( \tilde{A}^k X \right)_{ip}$$

$$= \left( \hat{Y}_{iq} - Y_{iq} \right) \left( \sum_{k=0}^{K} \theta_k \tilde{A}^k X \right)_{ip}.$$

$\square$

### B.2 PROOF OF THEOREM 8

**Theorem 8.** *Consider a spectral GNN $\Psi$ with polynomial order $K$ trained using full-batch gradient descent for $T$ iterations with a learning rate $\eta$ on a training dataset $S_m$ sampled from a graph $G \sim cSBM(n, f, \Pi, Q)$ with average node degree $d \ll n$. When $n \to \infty$ and $K \ll n$, under Assumptions 1, 2, and 4, for any node $v_i$, $i \in [n]$, and for a constant $\epsilon \in (0, 1)$, with probability at least $1 - \epsilon$, $\Psi$ satisfies $\gamma$-uniform transductive stability, where $\gamma = r\beta$ and*

$$\beta = \frac{1}{\epsilon} \left[ O \left( \mathbb{E} \left[ \|\hat{y}_i - y_i\|_F^2 \right] \right) + O \left( \|\pi_{y_i}^\top \pi_{y_i} + \Sigma_{y_i}\|_F \right) \right.$$

$$\left. + O \left( \sum_{k=1}^{K} \sum_{j=1}^{n} \mathbb{E}[A_{ij}^k] \left\| \sum_{t=1}^{n} \mathbb{E}[A_{it}^k] \pi_{y_j}^\top \pi_{y_t} + \mathbb{E}[A_{ij}^k] \Sigma_{y_j} \right\|_F \right) \right].$$

*Proof.* Any spectral GNN described in Eq. (1) with a linear feature transformation function and a polynomial basis expanded on a normalized graph matrix can be expressed in the following form:

$$\hat{Y} = \text{softmax} \left( \sum_{k=0}^{K} \theta_k \tilde{A}^k X W \right), \tag{9}$$

where $\tilde{A} = D^{-\frac{1}{2}} A D^{-\frac{1}{2}}$ is the normalized graph adjacency matrix, and $D$ is the diagonal degree matrix. Here, $Y \in \mathbb{R}^{n \times C}$ denotes the ground truth node label matrix.

(1) **Walk counting**: According to Definition 7, we have

$$\mathbb{E}[A_{ij}^k] = \sum_{p \in P_{ij}^k} \prod_{(v,v') \in p} Q_{yy'}$$

(2) **Feature expectation**: Since we have $G \sim cSBM(n, f, \Pi, Q)$, node classes have a uniform prior $y_i \sim \mathcal{U}(1, C)$. Thus,

$$
\begin{aligned}
\mathbb{E}\left[XW\right]_{ij} &= \frac{1}{n} \sum_{u=1}^{n} (\pi_{y_u} W)_j \\
&= \frac{1}{n} \sum_{u=1}^{n} \sum_{c=1}^{C} p(y_u = c)(\pi_c W)_j \\
&= \frac{1}{n} \sum_{u=1}^{n} \sum_{c=1}^{C} \frac{1}{C} (\pi_c W)_j \\
&= \frac{1}{C} \sum_{c=1}^{C} (\pi_c W)_j.
\end{aligned}
\tag{10}
$$

– When $k \geq 1$, we have

$$
\begin{aligned}
\mathbb{E}[(\tilde{A}^k XW)_{ij}] &= \mathbb{E}\left[\tilde{A}_{i:}^k\right] \mathbb{E}\left[(XW)_{:j}\right] \\
&= \sum_{s=1}^{n} \mathbb{E}\left[\tilde{A}_{is}^k\right] \mathbb{E}\left[(XW)_{sj}\right] \\
&= \sum_{s=1}^{n} \mathbb{E}\left[\tilde{A}_{is}^k\right] \cdot \frac{1}{C} \sum_{c=1}^{C} (\pi_c W)_j.
\end{aligned}
$$

– When $k = 0$, we have

$$
\begin{aligned}
\mathbb{E}[(IXW)_{ij}] &= \mathbb{E}\left[(XW)_{ij}\right] \\
&= \frac{1}{C} \sum_{c=1}^{C} (\pi_c W)_j.
\end{aligned}
$$

Thus,

$$
\mathbb{E}[(\tilde{A}^k XW)_{ij}] = \begin{cases} \frac{1}{C} \sum_{c=1}^{C} (\pi_c W)_j, & k = 0 \\ \sum_{s=1}^{n} \mathbb{E}\left[\tilde{A}_{is}^k\right] \cdot \frac{1}{C} \sum_{c=1}^{C} (\pi_c W)_j, & k \geq 1 \end{cases}
\tag{11}
$$

(3) **Gradient Norm**: The gradient norm can be relaxed as:

$$
\begin{aligned}
\mathbb{E}\left[\|\nabla \ell(\hat{y}_i, y_i; \Theta, W)\|_F\right] &\leq \mathbb{E}\left[\|\nabla \ell(\hat{y}_i, y_i; \Theta, W)\|_{\ell_1}\right] \\
&= \sum_{k=0}^{K} \mathbb{E}\left[\|\frac{\partial \ell(\hat{y}_i, y_i; \Theta, W)}{\partial \theta_k}\|_{\ell_1}\right] + \mathbb{E}\left[\|\frac{\partial \ell(\hat{y}_i, y_i; \Theta, W)}{\partial W}\|_{\ell_1}\right].
\end{aligned}
\tag{12}
$$

According to Eq. (9) and Lemma 23, we get the partial derivatives $\frac{\partial \ell(\hat{y}_i, y_i; \Theta, W)}{\partial \theta_k}$ and $\frac{\partial \ell(\hat{y}_i, y_i; \Theta, W)}{\partial W_{pq}}$. Specially, when $m = 1$, we get the partial derivatives of empirical loss on training sample $(x_i, y_i)$:

$$
\frac{\partial \ell(\hat{y}_i, y_i; \Theta, W)}{\partial \theta_k} = \sum_{c=1}^{C} \left(\hat{Y}_{ic} - Y_{ic}\right) \left(\tilde{A}^k XW\right)_{ic}
\tag{13}
$$

$$\frac{\partial \ell(\hat{y}_i, y_i; \Theta, W)}{\partial W_{pq}} = \left(\hat{Y}_{iq} - Y_{iq}\right) \left(\sum_{k=0}^{K} \theta_k \tilde{A}^k X\right)_{ip} \tag{14}$$

Thus, we have:

$$
\begin{aligned}
\mathbb{E}\left[\|\frac{\partial \ell(\hat{y}_i, y_i; \Theta, W)}{\partial \theta_k}\|_{\ell_1}\right] &= \mathbb{E}\left[|\sum_{c=1}^{C} \left(\hat{Y}_{ic} - Y_{ic}\right)\left(\tilde{A}^k X W\right)_{ic}|\right] \\
&\leq \sum_{c=1}^{C} \mathbb{E}\left[|\left(\hat{Y}_{ic} - Y_{ic}\right)\left(\tilde{A}^k X W\right)_{ic}|\right] \\
&= \sum_{c=1}^{C} \mathbb{E}\left[|\left(\hat{Y}_{ic} - Y_{ic}\right)| \cdot |\left(\tilde{A}^k X W\right)_{ic}|\right] \\
&\leq \sum_{c=1}^{C} \frac{1}{2}\left(\mathbb{E}\left[\left(\hat{Y}_{ic} - Y_{ic}\right)^2\right] + \mathbb{E}\left[\left(\tilde{A}^k X W\right)_{ic}^2\right]\right) \\
&\quad (Lemma\ 28) \\
&= \frac{1}{2}\left(\mathbb{E}\left[\|\hat{y}_i - y_i\|_F^2\right] + \mathbb{E}\left[\|\tilde{A}_{i:}^k X W\|_F^2\right]\right);
\end{aligned}
\tag{15}
$$

$$
\begin{aligned}
\mathbb{E}\left[\|\frac{\partial \ell(\hat{y}_i, y_i; \Theta, W)}{\partial W}\|_{\ell_1}\right] &= \sum_{p=1}^{f}\sum_{q=1}^{C} \mathbb{E}\left[\|\frac{\partial \ell(\hat{y}_i, y_i; \Theta, W)}{\partial W_{pq}}\|_{\ell_1}\right] \\
&= \sum_{p=1}^{f}\sum_{q=1}^{C} \mathbb{E}\left[|\left(\hat{Y}_{iq} - Y_{iq}\right)\left(\sum_{k=0}^{K} \theta_k \tilde{A}^k X\right)_{ip}|\right] \\
&\leq \sum_{p=1}^{f}\sum_{k=0}^{K} |\theta_k| \left(\sum_{q=1}^{C} \mathbb{E}\left[|\left(\hat{Y}_{iq} - Y_{iq}\right)| \cdot |\left(\tilde{A}^k X\right)_{ip}|\right]\right) \\
&\leq \sum_{p=1}^{f}\sum_{k=0}^{K} |\theta_k| \left(\mathbb{E}\left[\sum_{q=1}^{C}\left(\hat{Y}_{iq} - Y_{iq}\right)^2\right] + \mathbb{E}\left[\sum_{q=1}^{C}\left(\tilde{A}^k X\right)_{ip}^2\right]\right) \\
&\quad (Lemma\ 28) \\
&= \sum_{p=1}^{f}\sum_{k=0}^{K} |\theta_k| \left(\mathbb{E}\left[\|\hat{y}_i - y_i\|_F^2\right] + C\mathbb{E}\left[\left(\tilde{A}^k X\right)_{ip}^2\right]\right) \\
&= \sum_{k=0}^{K} |\theta_k| \left(f \cdot \mathbb{E}\left[\|\hat{y}_i - y_i\|_F^2\right] + C\mathbb{E}\left[\|\tilde{A}_{i:}^k X\|_F^2\right]\right).
\end{aligned}
\tag{16}
$$

(4) **Expectation** $\mathbb{E}\left[\|\tilde{A}_{i:}^k X W\|_F^2\right]$ **and** $\mathbb{E}\left[\|\tilde{A}_{i:}^k X\|_F^2\right]$: For sparse graphs $G$ with adjacency matrix $A$, when $d \ll n$ (average degree much smaller than the number of nodes) and $k \ll n$ (walk length much smaller than the number of nodes), $A_{ia}^k$ and $A_{ib}^k$ can be treated as independent variables due to the following reasons: (a). The overlap between walks of different lengths is limited due to the sparsity of the graph. (b). The existence of a $k$-length walk between two nodes is a rare event when $k \ll n$, and the joint occurrences of two rare events can be neglected. (c). When $d \ll n$, the variance of $A_{ij}^k$ is negligible compared to

$\left(\mathbb{E}[A_{ij}^k]\right)^2$. Thus, by Eq. (11), we derive the following for the case $k \geq 1$:

$$
\begin{aligned}
\mathbb{E}[\|\tilde{A}_{i:}^k XW\|_F^2] &= \mathbb{E}\left[\sum_{c=1}^C \left(\sum_{s=1}^n \tilde{A}_{is}^k (XW)_{sc}\right)^2\right] \\
&= \mathbb{E}\left[\sum_{c=1}^C \sum_{s=1}^n \sum_{t=1}^n \tilde{A}_{is}^k \tilde{A}_{it}^k (XW)_{sc} (XW)_{tc}\right] \\
&= \sum_{c=1}^C \sum_{s,t=1}^n \mathbb{E}\left[\tilde{A}_{is}^k \tilde{A}_{it}^k (XW)_{sc} (XW)_{tc}\right] \\
&= \sum_{c=1}^C \sum_{s,t=1}^n \mathbb{E}\left[\tilde{A}_{is}^k\right] \cdot \mathbb{E}\left[\tilde{A}_{it}^k\right] \cdot \mathbb{E}[(XW)_{sc}(XW)_{tc}] \\
&= \sum_{c=1}^C \sum_{s=1}^n \mathbb{E}\left[\tilde{A}_{is}^k\right] \Bigg[\sum_{t=1,t\neq s}^n \mathbb{E}\left[\tilde{A}_{it}^k\right] \cdot \mathbb{E}[(XW)_{sc}(XW)_{tc}] \\
&\quad + \mathbb{E}\left[\tilde{A}_{is}^k\right] \cdot \mathbb{E}\left[(XW)_{sc}^2\right]\Bigg] \\
&= \frac{1}{d^{2k}} \sum_{c=1}^C \sum_{s=1}^n \mathbb{E}\left[\tilde{A}_{is}^k\right] \Bigg[\sum_{t=1,t\neq s}^n \mathbb{E}\left[\tilde{A}_{it}^k\right] \cdot (\pi_{y_s} W)_c \cdot (\pi_{y_t} W)_c \\
&\quad + \mathbb{E}\left[\tilde{A}_{is}^k\right] \cdot W_{:c}^\top \left(\pi_{y_s}^\top \pi_{y_s} + \Sigma_{y_s}\right) W_{:c}\Bigg].
\end{aligned}
$$

When $k = 0$, we have:

$$
\begin{aligned}
\mathbb{E}\left[\|\tilde{A}_{i:}^k XW\|_F^2\right] &= \mathbb{E}\left[\|X_{i:}W\|_F^2\right] \\
&= \mathbb{E}\left[\sum_{c=1}^C (XW)_{ic}^2\right] \\
&= \sum_{c=1}^C W_{:c}^\top \left(\pi_{y_i}^\top \pi_{y_i} + \Sigma_{y_i}\right) W_{:c}.
\end{aligned}
$$

Thus, we obtain

$$
\mathbb{E}\left[\|\tilde{A}_{i:}^k XW\|_F^2\right] = \begin{cases}
\sum_{c=1}^C W_{:c}^\top \left(\pi_{y_i}^\top \pi_{y_k} + \Sigma_{y_i}\right) W_{:c}, k = 0 \\
\frac{1}{d^{2k}} \sum_{c=1}^C \sum_{s=1}^n \mathbb{E}\left[\tilde{A}_{is}^k\right] \Big[\sum_{t=1,t\neq s}^n \mathbb{E}\left[\tilde{A}_{it}^k\right] \cdot (\pi_{y_s} W)_c \cdot (\pi_{y_t} W)_c \\
\quad + \mathbb{E}\left[\tilde{A}_{is}^k\right] \cdot W_{:c}^\top \left(\pi_{y_s}^\top \pi_{y_s} + \Sigma_{y_s}\right) W_{:c}\Big], k \geq 1
\end{cases}
\tag{17}
$$

Similarly, by Eq. (10), we have

$$
\mathbb{E}\left[\|\tilde{A}_{i:}^k X\|_F^2\right] = \begin{cases}
\sum_{c=1}^C I_{:c}^\top \left(\pi_{y_i}^\top \pi_{y_k} + \Sigma_{y_i}\right) I_{:c}, k = 0 \\
\frac{1}{d^{2k}} \sum_{q=1}^f \sum_{s=1}^n \mathbb{E}\left[\tilde{A}_{is}^k\right] \Big[\sum_{t=1,t\neq s}^n \mathbb{E}\left[\tilde{A}_{it}^k\right] \cdot \pi_{y_s,q} \cdot \pi_{y_t,q} \\
\quad + \mathbb{E}\left[\tilde{A}_{is}^k\right] \cdot I_{:q}^\top \left(\pi_{y_s}^\top \pi_{y_s} + \Sigma_{y_s}\right) I_{:q}\Big], k \geq 1
\end{cases}
\tag{18}
$$

By substituting Eq. (17) into Eq. (15), Eq. (18) into Eq. (16), and combining Eq. (15) and Eq. (16) into Eq. (12), we obtain:

$$\mathbb{E}\left[\|\nabla\ell(\hat{y}_i, y_i; \Theta, W)\|_F\right]$$

$$\leq \frac{1}{2}\left(\mathbb{E}\left[\|\hat{y}_i - y_i\|_F^2\right] + \sum_{c=1}^{C} W_{:c}^{\top}\left(\pi_{y_i}^{\top}\pi_{y_i} + \Sigma_{y_i}\right)W_{:c}\right)$$

$$+ \sum_{k=1}^{K}\frac{1}{2}\left[\mathbb{E}\left[\|\hat{y}_i - y_i\|_F^2\right] + \frac{1}{d^{2k}}\sum_{c=1}^{C}\sum_{s=1}^{n}\mathbb{E}\left[\tilde{A}_{is}^k\right]\right.$$

$$\left.\cdot\left[\sum_{t=1, t\neq s}^{n}\mathbb{E}\left[\tilde{A}_{it}^k\right]\cdot(\pi_{y_s}W)_c\cdot(\pi_{y_t}W)_c + \mathbb{E}\left[\tilde{A}_{is}^k\right]\cdot W_{:c}^{\top}\left(\pi_{y_s}^{\top}\pi_{y_s} + \Sigma_{y_s}\right)W_{:c}\right]\right]$$

$$+ |\theta_0|\left(f\cdot\mathbb{E}\left[\|\hat{y}_i - y_i\|_F^2\right] + C\sum_{c=q}^{f}I_{:q}^{\top}\left(\pi_{y_i}^{\top}\pi_{y_i} + \Sigma_{y_i}\right)I_{:q}\right)$$

$$+ \sum_{k=1}^{K}|\theta_k|\left[f\cdot\mathbb{E}\left[\|\hat{y}_i - y_i\|_F^2\right] + C\frac{1}{d^{2k}}\sum_{c=1}^{C}\sum_{s=1}^{n}\mathbb{E}\left[\tilde{A}_{is}^k\right]\right.$$

$$\left.\cdot\left[\sum_{t=1, t\neq s}^{n}\mathbb{E}\left[\tilde{A}_{it}^k\right]\cdot\pi_{y_s,q}\cdot\pi_{y_t,q} + \mathbb{E}\left[\tilde{A}_{is}^k\right]\cdot I_{:q}^{\top}\left(\pi_{y_s}^{\top}\pi_{y_s} + \Sigma_{y_s}\right)I_{:q}\right]\right]$$

$$= \left(\frac{K+1}{2} + f\sum_{k=0}^{K}|\theta_k|\right)\mathbb{E}\left[\|\hat{y}_i - y_i\|_F^2\right]$$

$$+ \frac{1}{2}\sum_{c=1}^{C}W_{:c}^{\top}\left(\pi_{y_i}^{\top}\pi_{y_i} + \Sigma_{y_i}\right)W_{:c} + |\theta_0|C\sum_{c=1}^{C}I_{:c}^{\top}\left(\pi_{y_i}^{\top}\pi_{y_i} + \Sigma_{y_i}\right)I_{:c}$$

$$+ \sum_{k=1}^{K}\frac{1}{d^{2k}}\sum_{c=1}^{C}\sum_{s=1}^{n}\mathbb{E}\left[\tilde{A}_{is}^k\right]$$

$$\cdot\left[\sum_{t=1, t\neq s}^{n}\mathbb{E}\left[\tilde{A}_{it}^k\right]\cdot(\pi_{y_s}W)_c\cdot(\pi_{y_t}W)_c + \mathbb{E}\left[\tilde{A}_{is}^k\right]\cdot W_{:c}^{\top}\left(\pi_{y_s}^{\top}\pi_{y_s} + \Sigma_{y_s}\right)W_{:c}\right]$$

$$+ \sum_{k=1}^{K}\frac{C}{d^{2k}}|\theta_k|\sum_{q=1}^{f}\sum_{s=1}^{n}\mathbb{E}\left[\tilde{A}_{is}^k\right]$$

$$\cdot\left[\sum_{t=1, t\neq s}^{n}\mathbb{E}\left[\tilde{A}_{it}^k\right]\cdot\pi_{y_s,q}\cdot\pi_{y_t,q} + \mathbb{E}\left[\tilde{A}_{is}^k\right]\cdot I_{:q}^{\top}\left(\pi_{y_s}^{\top}\pi_{y_s} + \Sigma_{y_s}\right)I_{:q}\right]$$

$$= \left(\frac{K+1}{2} + f\sum_{k=0}^{K}|\theta_k|\right)\mathbb{E}\left[\|\hat{y}_i - y_i\|_F^2\right]$$

$$+ \frac{1}{2}\sum_{c=1}^{C}W_{:c}^{\top}\left(\pi_{y_i}^{\top}\pi_{y_i} + \Sigma_{y_i}\right)W_{:c} + |\theta_0|C\sum_{c=1}^{C}I_{:c}^{\top}\left(\pi_{y_i}^{\top}\pi_{y_i} + \Sigma_{y_i}\right)I_{:c}$$

$$+ \sum_{k=1}^{K}\frac{1}{d^{2k}}\sum_{c=1}^{C}\sum_{s=1}^{n}\mathbb{E}\left[\tilde{A}_{is}^k\right]\left[W_{:c}^{\top}\left(\sum_{\substack{t=1\\t\neq s}}^{n}\mathbb{E}\left[\tilde{A}_{it}^k\right]\pi_{y_s}^{\top}\pi_{y_t} + \mathbb{E}\left[\tilde{A}_{is}^k\right]\left(\pi_{y_s}^{\top}\pi_{y_s} + \Sigma_{y_s}\right)\right)W_{:c}\right]$$

$$+ \sum_{k=1}^{K}\frac{C|\theta_k|}{d^{2k}}\sum_{q=1}^{f}\sum_{s=1}^{n}\mathbb{E}\left[\tilde{A}_{is}^k\right]\left[\sum_{\substack{t=1\\t\neq s}}^{n}\mathbb{E}\left[\tilde{A}_{it}^k\right]\pi_{y_s,q}\pi_{y_t,q} + \mathbb{E}\left[\tilde{A}_{is}^k\right]I_{:q}^{\top}\left(\left(\pi_{y_s}^{\top}\pi_{y_s} + \Sigma_{y_s}\right)\right)I_{:q}\right].$$

Under Assumption 4, we can further simplify and relax the expression to:

$$\mathbb{E}\left[\|\nabla \ell(\hat{y}_i, y_i; \Theta, W)\|_F\right]$$

$$\leq \left(\frac{K+1}{2} + f \sum_{k=0}^{K} B_\Theta\right) \mathbb{E}\left[\|\hat{y}_i - y_i\|_F^2\right]$$

$$+ \frac{1}{2} Tr\left(W^T \left(\pi_{y_i}^\top \pi_{y_i} + \Sigma_{y_i}\right) W\right) + B_\Theta C Tr\left(\pi_{y_i}^\top \pi_{y_i} + \Sigma_{y_i}\right)$$

$$+ \sum_{k=1}^{K} \frac{1}{d^{2k}} \sum_{s=1}^{n} \mathbb{E}\left[\tilde{A}_{is}^k\right] Tr\left(\sum_{\substack{t=1 \\ t \neq s}}^{n} \mathbb{E}\left[\tilde{A}_{it}^k\right] \pi_{y_s}^\top \pi_{y_t} + \mathbb{E}\left[\tilde{A}_{is}^k\right] \left(\pi_{y_s}^\top \pi_{y_s} + \Sigma_{y_s}\right)\right)$$

$$+ \sum_{k=1}^{K} \frac{C B_\Theta}{d^{2k}} \sum_{s=1}^{n} \mathbb{E}\left[\tilde{A}_{is}^k\right] \left[\sum_{\substack{t=1 \\ t \neq s}}^{n} \mathbb{E}\left[\tilde{A}_{it}^k\right] Tr\left(\pi_{y_s}^\top \pi_{y_t}\right) + \mathbb{E}\left[\tilde{A}_{is}^k\right] Tr\left(\left(\pi_{y_s}^\top \pi_{y_s} + \Sigma_{y_s}\right)\right)\right]$$

$$\leq \left(\frac{K+1}{2} + f B_\Theta(K+1)\right) \mathbb{E}\left[\|\hat{y}_i - y_i\|_F^2\right]$$

$$+ \left(\frac{B_W^2}{2} + B_\Theta C\right) Tr\left(\pi_{y_i}^\top \pi_{y_i} + \Sigma_{y_i}\right)$$

$$+ \sum_{k=1}^{K} \frac{1 + C B_\Theta}{d^{2k}} \sum_{j=1}^{n} \mathbb{E}\left[A_{ij}^k\right] Tr\left(\sum_{\substack{t=1 \\ t \neq j}}^{n} \mathbb{E}\left[A_{it}^k\right] \pi_{y_j}^\top \pi_{y_t} + \mathbb{E}\left[A_{ij}^k\right] \left(\pi_{y_j}^\top \pi_{y_j} + \Sigma_{y_j}\right)\right).$$

$$(19)$$

With Lemma 22, we rewrite it as

$$\mathbb{E}\left[\|\nabla \ell(\hat{y}_i, y_i; \Theta, W)\|_F\right] \leq O\left(\mathbb{E}\left[\|\hat{y}_i - y_i\|_F^2\right]\right) + O\left(\|\pi_{y_i}^\top \pi_{y_i} + \Sigma_{y_i}\|_F\right)$$

$$+ O\left(\sum_{k=1}^{K} \sum_{j=1}^{n} \mathbb{E}\left[A_{ij}^k\right] \|\sum_{t=1}^{n} \mathbb{E}\left[A_{it}^k\right] \pi_{y_j}^\top \pi_{y_t} + \mathbb{E}\left[A_{ij}^k\right] \Sigma_{y_j}\|_F\right).$$

$$(20)$$

(5) **Concentration Bound**: By Jensen's inequality (Lemma 19), we have:

$$\mathbb{E}[\|\nabla \ell(\hat{y}_i, y_i; \Theta, W)\|_F]^2 \leq \mathbb{E}[\|\nabla \ell(\hat{y}_i, y_i; \Theta, W)\|_F^2],$$

which implies:

$$\mathbb{E}[\|\nabla \ell(\hat{y}_i, y_i; \Theta, W)\|_F] \leq \sqrt{\mathbb{E}[\|\nabla \ell(\hat{y}_i, y_i; \Theta, W)\|_F^2]}. \tag{21}$$

Using Markov's inequality (Lemma 20), for a positive constant $a$, we have:

$$\mathbb{P}(\|\nabla \ell(\hat{y}_i, y_i; \Theta, W)\|_F \geq a) \leq \frac{\mathbb{E}[\|\nabla \ell(\hat{y}_i, y_i; \Theta, W)\|_F]}{a} = \epsilon. \tag{22}$$

Solving for $a$, we obtain:

$$a = \frac{\mathbb{E}[\|\nabla \ell(\hat{y}_i, y_i; \Theta, W)\|_F]}{\epsilon}. \tag{23}$$

Therefore, combining Eq. (20), Eq. (21), Eq. (22), and Eq. (23), with probability at least $1 - \epsilon$, we have:

$$\|\nabla \ell(\hat{y}_i, y_i; \Theta, W)\|_F \leq \beta = \frac{1}{\epsilon} \mathbb{E}[\|\nabla \ell(\hat{y}_i, y_i; \Theta, W)\|_F].$$

When $\|\nabla \ell(\hat{y}_i, y_i; \Theta, W)\|_F \leq \beta$, according to Theorem 6, spectral GNNs on graphs $G \sim cSBM(n, f, \Pi, Q)$ have $\gamma$-uniform transductive stability. We rewrite this in Big-O notation

as:

$$\gamma = r \cdot \beta, \quad \beta = \frac{1}{\epsilon} \left[ O \left( \mathbb{E} \left[ \| \hat{y}_i - y_i \|_F^2 \right] \right) + O \left( \| \pi_{y_i}^\top \pi_{y_i} + \Sigma_{y_i} \|_F \right) \right.$$

$$\left. + O \left( \sum_{k=1}^K \sum_{j=1}^n \mathbb{E}[A_{ij}^k] \left\| \sum_{t=1}^n \mathbb{E}[A_{it}^k] \pi_{y_j}^\top \pi_{y_t} + \mathbb{E}[A_{ij}^k] \Sigma_{y_j} \right\|_F \right) \right],$$

where $r$ is the same constant as in Theorem 6.

$\square$

## C    GENERALIZATION ERROR BOUND OF SPECTRAL GNNs

We derive the generalization error bound of spectral GNNs based on their uniform transductive stability. Subsequently, we analyze how the number of training samples affects the generalization error bound.

We begin by introducing two lemmas for this proof.

**Lemma 24** (Inequality for permutation (El-Yaniv & Pechyony, 2006))**.** *Let $Z$ be a random permutation vector. Let $f(Z)$ be an $(m, q)$-symmetric permutation function satisfying $\| f(Z) - f(Z^{ij}) \| \le \beta$ for all $i \in I_1^m$ and $j \in I_{m+1}^{m+q}$. Define $H_2(n) \triangleq \sum_{i=1}^n \frac{1}{i^2}$ and $\Omega(m, q) \triangleq q^2 \left( H_2(m + q) - H_2(q) \right)$. Then*

$$\mathbb{P} \left( f(Z) - \mathbb{E}[f(Z)] \ge \epsilon \right) \le \exp \left( -\frac{\epsilon^2}{2\beta^2 \Omega(m, q)} \right).$$

**Lemma 25** (Risk and uniform stability (El-Yaniv & Pechyony, 2006))**.** *Given any training set $S_m$ and test set $\mathcal{D}_u$, the following holds:*

$$\mathbb{E} \left[ \mathcal{L}_{\mathcal{D}_u}(\Theta, W) - \mathcal{L}_{S_m}(\Theta, W) \right] = \mathbb{E} \left[ \Delta(i, j, i, i) \right], \quad i \in I_1^m, \ j \in I_{m+1}^{m+q},$$

*where $\Delta(i, j, i, i)$ denotes the change in the loss of sample $(x_i, y_i)$ when the model is trained on two datasets: one with $(x_i, y_i)$ in the training set and another with $(x_j, y_j)$ from the test set exchanged with $(x_i, y_i)$.*

### C.1    PROOF OF THEOREM 9

**Theorem 9** (Generalization Error Bound)**.** *Let $H_2(n) \triangleq \sum_{i=1}^n \frac{1}{i^2}$ and $\Omega(m, n - m) \triangleq (n - m)^2 \left( H_2(n) - H_2(n - m) \right)$. For $\epsilon \in (0, 1)$, if a spectral GNN is $\gamma$-uniform transductive stability with probability $1 - \epsilon$, then under Assumption 3, for $\delta \in (0, 1)$, with probability at least $(1 - \delta)(1 - \epsilon)$, the generalization error $\mathcal{L}_{\mathcal{D}_u}(\Theta, W) - \mathcal{L}_{S_m}(\Theta, W)$ is upper-bounded by:*

$$\gamma + \left( 2\gamma + \left( \frac{1}{n - m} + \frac{1}{m} \right) (B_\ell - \gamma) \right) \sqrt{2\Omega(m, n - m) \log \frac{1}{\delta}}. \tag{3}$$

*Proof.* Let $\Delta(i, j, s, t) \triangleq \ell(\hat{y}_t, y_t; \Theta_{ij}^T, W_{ij}^T) - \ell(\hat{y}_s, y_s; \Theta^T, W^T)$, where $\Theta_{ij}^T, W_{ij}^T$ are model parameters trained on dataset $S_m^{ij}$ for $T$ iterations and $\Theta^T, W^T$ are model parameters trained on dataset $S_m$. We first derive a bound on the permutation stability of the function $f(S_m, \mathcal{D}_u) \triangleq \mathcal{L}_{\mathcal{D}_u}(\Theta, W) - \mathcal{L}_{S_m}(\Theta, W)$, where $q = n - m$. The bound is given as:

$$\left\| \left( \mathcal{L}_{\mathcal{D}_u}(\Theta, W) - \mathcal{L}_{S_m}(\Theta, W) \right) - \left( \mathcal{L}_{\mathcal{D}_u}(\Theta^{ij}, W^{ij}) - \mathcal{L}_{S_m}(\Theta^{ij}, W^{ij}) \right) \right\| \le$$

$$\frac{1}{q} \sum_{r=m+1, r \ne j}^{m+q} \| \Delta(i, j, r, r) \| + \frac{1}{q} \| \Delta(i, j, i, j) \| + \frac{1}{m} \sum_{r=1, r \ne i}^m \| \Delta(i, j, r, r) \| + \frac{1}{m} \| \Delta(i, j, j, i) \|. \tag{24}$$

According to Definition 5, Assumption 3 and Theorem 6, we have

$$\max_{1 \le r \le m+q} \| \Delta(i, j, r, r) \| \le \gamma = \alpha_1 \sum_{t=1}^T (1 + \eta \alpha_2)^{t-1} \frac{2\eta\beta}{m}$$

Thus, Eq. (24) is bounded:

$$\| \left(\mathcal{L}_{\mathcal{D}_u}(\Theta, W) - \mathcal{L}_{S_m}(\Theta, W)\right) - \left(\mathcal{L}_{\mathcal{D}_u}(\Theta^{ij}, W^{ij}) - \mathcal{L}_{S_m}(\Theta^{ij}, W^{ij})\right) \|$$

$$\leq \frac{q-1}{q}\gamma + \frac{1}{q}B_\ell + \frac{m-1}{m}\gamma + \frac{1}{m}B_\ell$$

$$= \left(\frac{q-1}{q} + \frac{m-1}{m}\right)\gamma + \left(\frac{1}{q} + \frac{1}{m}\right)B_\ell$$

Let $\tilde{\beta} = \left(\frac{q-1}{q} + \frac{m-1}{m}\right)\gamma + \left(\frac{1}{q} + \frac{1}{m}\right)B_\ell$. Then, the function $f(S_m, \mathcal{D}_u) = \mathcal{L}_{\mathcal{D}_u}(\Theta, W) - \mathcal{L}_{S_m}(\Theta, W)$ has transductive stability $\tilde{\beta}$. Apply Lemma 24 to $f(S_m, \mathcal{D}_u)$, equating the bound to $\delta$

$$\exp\left(-\frac{\epsilon^2}{2\tilde{\beta}^2 \Omega(m, q)}\right) = \delta,$$

we get

$$\epsilon = \tilde{\beta}\sqrt{2\Omega(m, q)\log\frac{1}{\delta}}$$

Therefore, we obtain that the probability at least $1 - \delta$ that

$$\mathcal{L}_{\mathcal{D}_u}(\Theta, W) - \mathcal{L}_{S_m}(\Theta, W) - \mathbb{E}\left[\mathcal{L}_{\mathcal{D}_u}(\Theta^{ij}, W^{ij}) - \mathcal{L}_{S_m}(\Theta^{ij}, W^{ij})\right] \leq \tilde{\beta}\sqrt{2\Omega(m, q)\log\frac{1}{\delta}} \quad (25)$$

According to Lemma 25 and Theorem 6, for $1 \leq i \leq m, m+1 \leq j \leq n$, we have

$$\mathbb{E}\left[\mathcal{L}_{\mathcal{D}_u}(\Theta^{ij}, W^{ij}) - \mathcal{L}_{S_m}(\Theta^{ij}, W^{ij})\right] = \mathbb{E}\left[\Delta(i, j, i, i)\right] \leq \gamma \quad (26)$$

Substitute Eq. (26) into Eq. (25), we get:

$$\mathcal{L}_{\mathcal{D}_u}(\Theta, W) \leq \mathcal{L}_{S_m}(\Theta, W) + \gamma + \tilde{\beta}\sqrt{2\Omega(m, q)\log\frac{1}{\delta}}$$

It is rewritten as:

$$\mathcal{L}_{\mathcal{D}_u}(\Theta, W) - \mathcal{L}_{S_m}(\Theta, W) \leq \gamma + \left(2\gamma + \left(\frac{1}{n-m} + \frac{1}{m}\right)(B_\ell - \gamma)\right)\sqrt{2\Omega(m, n-m)\log\frac{1}{\delta}}$$

$$\square$$

## C.2 PROOF OF LEMMA 10

**Lemma 10.** *Consider a spectral GNN trained with $m$ samples as $n \to \infty$. As the sample size $m$ increases, the generalization error bound decreases at the rate $O(1/m) + O\left(\sqrt{2\log(1/\delta)/m}\right)$.*

*Proof.* The proof is proceeded in three steps:

(1) $\frac{1}{n-m}$ is neglectable compared with $\frac{1}{m}$: As $m < n$, we have $m = o(n)$.

$\frac{m}{n-m} = \frac{m}{n} \cdot \frac{1}{1-\frac{m}{n}}$ when $n \to \infty$, we have $\frac{m}{n} \to 0$ and $\frac{1}{1-\frac{m}{n}} \to 1$ as $m = o(n)$. Therefore,

$$\lim_{n\to\infty}\frac{m}{n-m} = 0, \lim_{n\to\infty}\frac{\frac{1}{n-m}}{\frac{1}{m}} = 0;$$

which indicates

$$\frac{1}{n-m} = o(\frac{1}{m})$$

(2) $\Omega(m, n-m)$ increase with $m$: As $H_2(k) = \sum_{i=1}^{k}\frac{1}{i^2}$, we have:

$$H_2(n) - H_2(n-m) = \sum_{i=n-m+1}^{n}\frac{1}{i^2}$$

As

$$m \cdot \frac{1}{n^2} \leq \sum_{i=n-m+1}^{n} \frac{1}{i^2} \leq m \cdot \frac{1}{(n-m)^2},$$

we have

$$m \cdot \frac{1}{n^2} \leq H_2(n) - H_2(n-m) \leq m \cdot \frac{1}{(n-m)^2}.$$

Multiple two sides with $(n-m)^2$, we have:

$$(n-m)^2 \cdot m \cdot \frac{1}{n^2} \leq (n-m)^2 \cdot (H_2(n) - H_2(n-m)) \leq (n-m)^2 \cdot m \cdot \frac{1}{(n-m)^2},$$

As $\Omega(m, n-m) = (n-m)^2 \left( H_2(n) - H_2(n-m) \right)$, we have:

$$\frac{m(n-m)^2}{n^2} \leq \Omega(m, n-m) \leq m$$

i.e.,

$$\Omega(m, n-m) = O(m)$$

(3) Generalization error bound: From Theorem 6, we have $\gamma = O(\frac{1}{m})$. Therefore:

$$\gamma + \left( 2\gamma + \left( \frac{1}{n-m} + \frac{1}{m} \right) (B_\ell - \gamma) \right) \sqrt{2\Omega(m, n-m) \log \frac{1}{\delta}}$$

$$= O(\frac{1}{m}) + \left( O(\frac{1}{m}) + \left( o(\frac{1}{m}) + \frac{1}{m} \right) \left( B_\ell - O(\frac{1}{m}) \right) \right) \sqrt{2O(m) \log \frac{1}{\delta}}$$

$$= O(\frac{1}{m}) + B_\ell O(\frac{1}{m}) O(m^{1/2}) \sqrt{2 \log \frac{1}{\delta}}$$

$$= O\left( \frac{1}{m} + B_\ell \sqrt{\frac{2 \log(\frac{1}{\delta})}{m}} \right)$$

In summary, the decay rate of generalization error bound is $O\left( \frac{1}{m} + O(\sqrt{\frac{2 \log(\frac{1}{\delta})}{m}}) \right)$.

$\square$

**Proposition 11.** *For a spectral GNN $\Psi_{\tilde{\sigma}}$ with a non-linear feature transformation function $f_W(X) = \tilde{\sigma}(XW)$, assume the gradient norm bound $\beta$ in Theorem 9 is the same for $\Psi$ and $\Psi_{\tilde{\sigma}}$. If $Lip(\tilde{\sigma}) \leq 1$ and $Smt(\tilde{\sigma}) \leq 1$, then $\gamma_{\tilde{\sigma}} \leq \gamma$, where $\gamma_{\tilde{\sigma}}$ is the stability of $\Psi_{\tilde{\sigma}}$.*

*Proof.* We consider spectral GNN $\Psi$:

$$\Psi(M, X) = \sigma(\sum_{k=0}^{K} \tilde{A}^k XW)$$

and spectral GNN $\Psi_{\tilde{\sigma}}$:

$$\Psi_{\tilde{\sigma}}(M, X) = \sigma(\sum_{k=0}^{K} \tilde{\sigma} \left( \tilde{A}^k XW \right))$$

(1) **Lipschitz Constant:** For any two sets of parameters $(\Theta_1, W_1)$ and $(\Theta_2, W_2)$, we have:

$$\|\Psi_{\tilde{\sigma}}(\Theta_1, W_1) - \Psi_{\tilde{\sigma}}(\Theta_2, W_2)\|$$

$$= \|\sigma(\sum_{i=0}^{K} \theta_{1k}\tilde{\sigma}(\tilde{A}^k XW_1)) - \sigma(\sum_{i=0}^{K} \theta_{2k}\tilde{\sigma}(\tilde{A}^k XW_2))\|$$

$$\leq Lip(\sigma)\|\sum_{i=0}^{K} \theta_{1k}\tilde{\sigma}(\tilde{A}^k XW_1) - \sum_{i=0}^{K} \theta_{2k}\tilde{\sigma}(\tilde{A}^k XW_2)\|$$

$$\leq Lip(\sigma)\|\sum_{i=0}^{K} (\theta_{1k} - \theta_{2k})\tilde{\sigma}(\tilde{A}^k XW_1) + \sum_{i=0}^{K} \theta_{2k}(\tilde{\sigma}(\tilde{A}^k XW_1) - \tilde{\sigma}(\tilde{A}^k XW_2))\|$$

$$\leq Lip(\sigma)(\|\sum_{i=0}^{K} (\theta_{1k} - \theta_{2k})\tilde{\sigma}(\tilde{A}^k XW_1)\| + \|\sum_{i=0}^{K} \theta_{2k}(\tilde{\sigma}(\tilde{A}^k XW_1) - \tilde{\sigma}(\tilde{A}^k XW_2))\|)$$

$$\leq Lip(\sigma)(\|\Theta_1 - \Theta_2\|_F \cdot \max_k \|\tilde{\sigma}(\tilde{A}^k XW_1)\|_2 + \|\Theta_2\|_F \cdot Lip(\tilde{\sigma}) \cdot \max_k \|\tilde{A}^k X(W_1 - W_2)\|_2)$$

Since $Lip(\tilde{\sigma}) \leq 1$, we have:

$$\|\Psi_{\tilde{\sigma}}(\Theta_1, W_1) - \Psi_{\tilde{\sigma}}(\Theta_2, W_2)\| \leq Lip(\sigma)(\|\Theta_1 - \Theta_2\|_F \cdot C_1 + \|\Theta_2\|_F \cdot \|W_1 - W_2\|_F \cdot C_2)$$

where $C_1, C_2$ are constants depending on $X, \tilde{A}$. The right hand side is identical to the bound we get for $\Psi$ without the activation function. Therefore, $Lip(\Psi_{\tilde{\sigma}}) \leq Lip(\Psi)$.

(2) **Smoothness Constant:** We first get partial derivatives of $\Psi$ and $\Psi_{\tilde{\sigma}}$ with respect to $\theta_k$:

$$\frac{\partial \Psi}{\partial \theta_k} = \nabla \sigma(\sum_{i=0}^{K} \theta_i \tilde{A}^i XW) \cdot \tilde{A}^k XW$$

$$\frac{\partial \Psi_{\tilde{\sigma}}}{\partial \theta_k} = \nabla \sigma(\sum_{i=0}^{K} \theta_i \tilde{\sigma}(\tilde{A}^i XW)) \cdot \tilde{\sigma}(\tilde{A}^k XW)$$

Partial derivatives of $\Psi$ and $\Psi_{\tilde{\sigma}}$ with respect to $W$ are:

$$\frac{\partial \Psi}{\partial W} = \nabla \sigma(\sum_{i=0}^{K} \theta_i \tilde{A}^i XW) \cdot \sum_{i=0}^{K} \theta_i \tilde{A}^i X$$

$$\frac{\partial \Psi_{\tilde{\sigma}}}{\partial W} = \nabla \sigma(\sum_{i=0}^{K} \theta_i \tilde{\sigma}(\tilde{A}^i XW)) \cdot \sum_{i=0}^{K} \theta_i \nabla \tilde{\sigma}(\tilde{A}^i XW) \cdot \tilde{A}^i X$$

The Lipschitz constant of these gradients determine the smoothness. For $\Psi_{\tilde{\sigma}}$, the additional $\tilde{\sigma}$ and $\nabla \tilde{\sigma}$ terms do not increase the Lipschitz constant of the gradient as $Lip(\tilde{\sigma}) \leq 1, Smt(\tilde{\sigma}) \leq 1$:

- $\tilde{\sigma}$ is 1-Lipschitz, so it doesn't increase the difference between inputs.
- $\nabla \tilde{\sigma}$ is bounded by 1 (since $Smt(\tilde{\sigma}) \leq 1$), so it doesn't amplify the gradient.

Therefore, the Lipschitz constant of the gradient of $\Psi_{\tilde{\sigma}}$ is at most equal to that of $\Psi$, i.e., :

$$Smt(\Psi_{\tilde{\sigma}}) \leq Smt(\Psi)$$

(3) **Stability** $\gamma_{\tilde{\sigma}}$**:** According to Theorem 6, we have $\alpha_1 = Lip(\ell) \cdot Lip(\Psi)$ and $\alpha_2 = Smt(\Psi)\beta_1 + Smt(\ell)Lip(\Psi)\beta_2$. Thus, we have a smaller $\alpha_{1\tilde{\sigma}}, \alpha_{2\tilde{\sigma}}$ as $Lip(\Psi_{\tilde{\sigma}}) \leq Lip(\Psi)$ and $\Psi_{\tilde{\sigma}}) \leq Smt(\Psi)$. Then, we have $r_{\tilde{\sigma}} \leq r$.

As $\beta$ is the same for $\Psi_{\tilde{\sigma}}$ and $\Psi$ and $\gamma_{\tilde{\gamma}} = \beta r_{\tilde{\sigma}}, \gamma = \beta r$, we have

$$\gamma_{\tilde{\sigma}} \leq \gamma$$

$\square$

# D STABILITY ON SPECIALIZED CSBM

We establish the uniform transductive stability of spectral GNNs with the architecture described in Eq. (1) on graphs generated by $G \sim cSBM(n, f, \mu, u, \lambda, d)$. Theorem 13 is a specialized form of Theorem 8, where the data model is specialized to nodes with binary classes and Gaussian node features.

We present lemmas essential for calculating node features after graph convolution in Appendix D.1. Then we derive the expectation and variance of the element $A_{ij}^k$ in the adjacency matrix and the expectation and variance of node features after graph convolution in Appendix D.2. Using these results, we derive the transductive stability of spectral GNNs on the specialized data model in Appendix D.3.

## D.1 LEMMAS FOR THEOREM 13

**Lemma 26** (Poisson Limit Theorem (Durrett, 2019)). *For each $n$, let $X_{n,m}, 1 \leq m \leq n$, be independent random variables with $\mathbb{P}(X_{n,m} = 1) = p_{n,m}$ and $\mathbb{P}(X_{n,m} = 0) = 1 - p_{n,m}$. Suppose:*

1. $\sum_{m=1}^{n} p_{n,m} \to \lambda \in (0, \infty)$, *and*

2. $\max_{1 \leq m \leq n} p_{n,m} \to 0$,

*then if $S_n = \sum_{m=1}^{n} X_{n,m}$, $S_n$ converges in distribution to a Poisson random variable with mean $\lambda$, i.e., $S_n \sim \text{Poisson}(\lambda)$.*

*Remark.* The Poisson limit theorem, also known as the law of rare events, states that the total number of events will follow a Poisson distribution if the probability of occurrence of an event is small in each trial but there are a large number of trials. For more details, see (Durrett, 2019).

**Lemma 27** (Binomial Coefficient Approximation). *When $n \gg k$, the binomial coefficient $\binom{n}{k}$ can be approximated as:*

$$\binom{n}{k} \approx \frac{n^k}{k!}.$$

*Proof.* The binomial coefficient is defined as:

$$\binom{n}{k} = \frac{n!}{k!(n-k)!}.$$

Expanding the factorial terms for $n!$, we have:

$$\binom{n}{k} = \frac{n \cdot (n-1) \cdot (n-2) \cdot \ldots \cdot (n-k+1) \cdot (n-k)!}{k! \cdot (n-k)!}.$$

Canceling the $(n-k)!$ terms in the numerator and denominator gives:

$$\binom{n}{k} = \frac{n \cdot (n-1) \cdot (n-2) \cdot \ldots \cdot (n-k+1)}{k!}.$$

When $n \gg k$, the terms $(n-1), (n-2), \ldots, (n-k+1)$ are approximately equal to $n$. Therefore, the product simplifies as:

$$n \cdot (n-1) \cdot (n-2) \cdot \ldots \cdot (n-k+1) \approx n^k.$$

Substituting this approximation, we obtain:

$$\binom{n}{k} \approx \frac{n^k}{k!}, \quad \text{for } n \gg k.$$

$\square$

**Lemma 28** (Expecatations of $\mathbb{E}[AB]$). *For any two random variables $A$ and $B$, the following inequality holds:*

$$\mathbb{E}[AB] \leq \frac{1}{2}\mathbb{E}[A^2] + \frac{1}{2}\mathbb{E}[B^2].$$

*Proof.* Define a function $f(t)$ for any real number $t$:

$$f(t) = \mathbb{E}\left[\left(\frac{1}{\sqrt{2}}A - \frac{t}{\sqrt{2}}B\right)^2\right].$$

Since $f(t)$ is the expectation of a squared term, it is non-negative for any real $t$, i.e., $f(t) \geq 0$. Expanding $f(t)$, we get:

$$f(t) = \mathbb{E}\left[\frac{1}{2}A^2 - tAB + \frac{t^2}{2}B^2\right].$$

Rearranging terms, this becomes:

$$f(t) = \frac{1}{2}\mathbb{E}[A^2] - t\mathbb{E}[AB] + \frac{t^2}{2}\mathbb{E}[B^2].$$

Since $f(t) \geq 0$ for all $t$, substitute $t = 1$ to simplify:

$$f(1) = \frac{1}{2}\mathbb{E}[A^2] - \mathbb{E}[AB] + \frac{1}{2}\mathbb{E}[B^2] \geq 0.$$

Rearranging this inequality gives:

$$\mathbb{E}[AB] \leq \frac{1}{2}\mathbb{E}[A^2] + \frac{1}{2}\mathbb{E}[B^2].$$

Thus, the result holds. $\qquad\square$

**Lemma 29** (Monotonicity of $g(\lambda)$). *The function* $g(\lambda) = \left(\left(d + \lambda\sqrt{d}\right)^k - \left(d - \lambda\sqrt{d}\right)^k\right)^2$ *satisfies the following properties:*

- *It monotonically increases on* $\lambda \in [0, \sqrt{d}]$.
- *It monotonically decreases on* $\lambda \in [-\sqrt{d}, 0]$.
- *It achieves its minimum value when* $\lambda = 0$.

*Proof.* First, observe that $g(\lambda)$ is an even function because:

$$g(-\lambda) = \left(\left(d - \lambda\sqrt{d}\right)^k - \left(d + \lambda\sqrt{d}\right)^k\right)^2 = \left(\left(d + \lambda\sqrt{d}\right)^k - \left(d - \lambda\sqrt{d}\right)^k\right)^2 = g(\lambda).$$

Thus, it is symmetric about $\lambda = 0$. Therefore, we only need to analyze its behavior for $\lambda \geq 0$, and the results for $\lambda < 0$ follow by symmetry.

Define:

$$A = d + \lambda\sqrt{d}, \quad B = d - \lambda\sqrt{d}.$$

Then, the function $g(\lambda)$ can be rewritten as:

$$g(\lambda) = (A^k - B^k)^2.$$

Using the chain rule:

$$g'(\lambda) = 2(A^k - B^k) \cdot \frac{\partial}{\partial\lambda}(A^k - B^k).$$

The derivative of $A^k - B^k$ with respect to $\lambda$ is:

$$\frac{\partial}{\partial\lambda}(A^k - B^k) = k\sqrt{d}(A^{k-1} + B^{k-1}).$$

Thus:

$$g'(\lambda) = 2k\sqrt{d}(A^k - B^k)(A^{k-1} + B^{k-1}).$$

When $\lambda \geq 0$, $A \geq B > 0$, we have:

$$A^k - B^k \geq 0, \quad A^{k-1} + B^{k-1} \geq 0.$$

Therefore:

$$g'(\lambda) \geq 0 \quad \text{for } \lambda \geq 0.$$

This shows that $g(\lambda)$ is monotonically increasing on $[0, \sqrt{d}]$.

By the even symmetry of $g(\lambda)$, we have:

$$g'(-\lambda) = -g'(\lambda).$$

Since $g'(\lambda) \geq 0$ for $\lambda \geq 0$, it follows that $g'(\lambda) \leq 0$ for $\lambda \leq 0$. Thus, $g(\lambda)$ monotonically decreases on $[-\sqrt{d}, 0]$.

At $\lambda = 0$, $A = B = d$, we have:

$$g(0) = (d^k - d^k)^2 = 0.$$

Thus, $g(\lambda)$ achieves its minimum value when $\lambda = 0$.

The proof is complete. $\qquad\square$

**Lemma 30** (Monotonicity of $g(\lambda)$). *The function $g(\lambda) = \sum_{s=1}^{k} \left(d + \lambda\sqrt{d}\right)^{k-s} \left(d - \lambda\sqrt{d}\right)^{s}$ satisfies the following properties:*

- *It monotonically decreases on $\lambda \in [0, \sqrt{d}]$.*
- *It monotonically increases on $\lambda \in [-\sqrt{d}, 0]$.*
- *It achieves its maximum value at $\lambda = 0$.*

*Proof.* The function $g(\lambda)$ can be rewritten as:

$$g(\lambda) = (2d)^k - \left(d + \lambda\sqrt{d}\right)^k - \left(d - \lambda\sqrt{d}\right)^k.$$

Differentiate $g(\lambda)$ with respect to $\lambda$:

$$g'(\lambda) = k\sqrt{d}\left[\left(d - \lambda\sqrt{d}\right)^{k-1} - \left(d + \lambda\sqrt{d}\right)^{k-1}\right].$$

- When $\lambda > 0$, we have $\left(d - \lambda\sqrt{d}\right) < \left(d + \lambda\sqrt{d}\right)$. This implies $\left(d - \lambda\sqrt{d}\right)^{k-1} < \left(d + \lambda\sqrt{d}\right)^{k-1}$ and $g'(\lambda) < 0$. Therefore, $g(\lambda)$ is strictly decreasing on $\lambda \in [0, \sqrt{d}]$.

- When $\lambda < 0$, we have $\left(d - \lambda\sqrt{d}\right) > \left(d + \lambda\sqrt{d}\right)$. This implies $\left(d - \lambda\sqrt{d}\right)^{k-1} > \left(d + \lambda\sqrt{d}\right)^{k-1}$ and $g'(\lambda) > 0$. Therefore, $g(\lambda)$ is strictly increasing on $\lambda \in [-\sqrt{d}, 0]$.

- When $\lambda = 0$, we have $\left(d + \lambda\sqrt{d}\right) = \left(d - \lambda\sqrt{d}\right) = d$ and $g(0) = (2d)^k - 2d^k$. This is the maximum value of $g(\lambda)$, as $g'(\lambda)$ changes sign from positive to negative at $\lambda = 0$.

The proof is complete. $\qquad\square$

## D.2 Expectation and Variance of $A_{ij}^k$ and $(\tilde{A}^k XW)_{ij}$

**Theorem 31** (Expectation and Variance of $A_{ij}^k$). *Let the graph be generated by $G \sim cSBM(n, f, \mu, u, \lambda, d)$. For $n \to \infty$, $d \ll n$, and $2 \le k \le k^2 \ll n$, the number of $k$-length walks connecting nodes $v_i$ and $v_j$ follows a Poisson distribution, $Poisson(\rho')$, where:*

$$\rho' = \begin{cases} \rho_= = \frac{(k-1)!}{n \cdot 2^{k-1}} \sum_{a=2}^{k+1} O\left( \sum_{s=\min(2,2(a-2),2(k+1-a))}^{\min(2(a-1),2(k+1-a))} c_{in}^{k-s} \cdot c_{out}^s \right), & \text{if } y_i = y_j, \\ \rho_{\neq} = \frac{(k-1)!}{n \cdot 2^{k-1}} \sum_{a=1}^{k} O\left( \sum_{s=1}^{\min(2a-1,2(k-a)+1)} c_{in}^{k-s} \cdot c_{out}^s \right), & \text{if } y_i \neq y_j. \end{cases}$$

*The expectation and variance are:*

$$\mathbb{E}[A_{ij}^k] = \rho', \quad \mathbb{V}\left[A_{ij}^k\right] = \rho'.$$

*When $k = 1$, the 1-length walk (i.e., a single edge) connecting nodes $v_i$ and $v_j$ follows a Bernoulli distribution, $Ber(p)$, where:*

$$p = \begin{cases} p_= = \frac{c_{in}}{n}, & \text{if } y_i = y_j, \\ p_{\neq} = \frac{c_{out}}{n}, & \text{if } y_i \neq y_j. \end{cases}$$

*The expectation and variance in this case are:*

$$\mathbb{E}[A_{ij}^k] = p, \quad \mathbb{V}\left[A_{ij}^k\right] = p(1-p).$$

*Proof.* According to Definition 7, the expectation of $A_{ij}^k$, the number of $k$-length walks between nodes $v_i$ and $v_j$, is given by:

$$\mathbb{E}[A_{ij}^k] = \sum_{p \in \mathcal{P}_{ij}^k} \prod_{(v,v') \in p} Q_{yy'},$$

where $\mathcal{P}_{ij}^k$ represents the set of all $k$-length walks between $v_i$ and $v_j$, and $Q_{yy'}$ is the probability of an edge between nodes $v$ and $v'$, conditioned on their respective classes $y$ and $y'$.

When $C = 2$ (binary classes), the edge probabilities $Q_{yy'}$ are:

$$Q_{yy'} = \begin{cases} \frac{c_{in}}{n}, & \text{if } y = y', \\ \frac{c_{out}}{n}, & \text{if } y \neq y', \end{cases}$$

where $c_{in}$ and $c_{out}$ are the intra-class and inter-class edge probabilities, respectively.

**Case 1:** $y_i = y_j$ **and** $k \ge 2$

For nodes $v_i$ and $v_j$ sharing the same class $y_i$, we consider walks of length $k$ that include $a$ nodes sharing the class $y_i$ and $k + 1 - a$ nodes with different classes. Since $v_i$ and $v_j$ both belong to class $y_i$, we need to choose $a - 2$ nodes from the same cluster and $k - a + 1$ nodes from the other cluster. The total number of ways to arrange these nodes in a walk is $(k - 1)!$, as there are $k - 1$ positions to fill. The probability of each edge depends on whether it connects nodes of the same class or different classes.

The number of ways to choose the nodes is as follows:

- Choose $a - 2$ nodes from $\frac{n}{2} - 2$ nodes in the same cluster: $\binom{\frac{n}{2}-2}{a-2}$.

- Choose $k - a + 1$ nodes from $\frac{n}{2}$ nodes in the other cluster: $\binom{\frac{n}{2}}{k-a+1}$.

The number of ways to arrange these nodes is $(k - 1)!$. Considering the class changes in the $k$-length walk, let $s$ denote the number of walk class changes:

- If $2a \ge k + 1$, then $s_{\min} = \min(2, 2(k + 1 - a))$ and $s_{\max} = 2(k + 1 - a)$.
- If $2a \le k + 1$, then $s_{\min} = \min(2, 2(a - 2))$ and $s_{\max} = 2(a - 1)$.

The probability of a $k$-length walk with $a$ nodes sharing the same class as $v_i$ is:

$$
p_k^a(v_i, v_j \mid y_i = y_j) =
$$

$$
\begin{cases}
\binom{\frac{n}{2}-2}{a-2} \cdot \binom{\frac{n}{2}}{k-a+1} \cdot (k-1)! \cdot \left( \displaystyle\sum_{s=\min(2,2(k+1-a))}^{2(k+1-a)} \left(\frac{c_{in}}{n}\right)^{k-s} \cdot \left(\frac{c_{out}}{n}\right)^s \right), & \text{if } 2a \geq k+1; \\[2em]
\binom{\frac{n}{2}-2}{a-2} \cdot \binom{\frac{n}{2}}{k-a+1} \cdot (k-1)! \cdot \left( \displaystyle\sum_{s=\min(2,2(a-2))}^{2(a-1)} \left(\frac{c_{in}}{n}\right)^{k-s} \cdot \left(\frac{c_{out}}{n}\right)^s \right), & \text{if } 2a < k+1.
\end{cases}
$$

The total probability of a $k$-length walk connecting $v_i$ and $v_j$ when $y_i = y_j$ is:

$$
p_k(v_i, v_j \mid y_i = y_j)
$$

$$
= \sum_{a=2}^{\frac{k+1}{2}} \binom{\frac{n}{2}-2}{a-2} \cdot \binom{\frac{n}{2}}{k-a+1} \cdot (k-1)! \cdot \sum_{s=\min(2,2(a-2))}^{2(a-1)} \left(\frac{c_{in}}{n}\right)^{k-s} \cdot \left(\frac{c_{out}}{n}\right)^s
$$

$$
+ \sum_{\frac{k+1}{2}}^{k+1} \binom{\frac{n}{2}-2}{a-2} \cdot \binom{\frac{n}{2}}{k-a+1} \cdot (k-1)! \cdot \sum_{s=\min(2,2(k+1-a))}^{2(k+1-a)} \left(\frac{c_{in}}{n}\right)^{k-s} \cdot \left(\frac{c_{out}}{n}\right)^s .
$$

$$(27)$$

Using Lemma 27, the binomial coefficients simplify as:

$$
\binom{\frac{n}{2}-2}{a-2} = \frac{\left(\frac{n}{2}-2\right)^{a-2}}{(a-2)!}, \qquad \binom{\frac{n}{2}}{k-a+1} = \frac{\left(\frac{n}{2}\right)^{k-a+1}}{(k-a+1)!}.
$$

Thus, we have

$$
\binom{\frac{n}{2}-2}{a-2} \cdot \binom{\frac{n}{2}}{k-a+1} \cdot (k-1)! = O\left( \left(\frac{n}{2}\right)^{k-1} \cdot \binom{k-1}{a-2} \right).
$$

Substituting into Eq. (27), we get:

$$
p_k(v_i, v_j \mid y_i = y_j)
$$

$$
= \sum_{a=2}^{\frac{k+1}{2}} O\left( \left(\frac{n}{2}\right)^{k-1} \cdot \binom{k-1}{a-2} \right) \cdot \left( \sum_{s=\min(2,2(a-2))}^{2(a-1)} \left(\frac{c_{in}}{n}\right)^{k-s} \cdot \left(\frac{c_{out}}{n}\right)^s \right)
$$

$$
+ \sum_{\frac{k+1}{2}}^{k+1} O\left( \left(\frac{n}{2}\right)^{k-1} \cdot \binom{k-1}{a-2} \right) \cdot \left( \sum_{s=\min(2,2(k+1-a))}^{2(k+1-a)} \left(\frac{c_{in}}{n}\right)^{k-s} \cdot \left(\frac{c_{out}}{n}\right)^s \right)
$$

$$
= \frac{1}{n \cdot 2^{k-1}} \sum_{a=2}^{\frac{k+1}{2}} O\left( \binom{k-1}{a-2} \cdot \left( \sum_{s=\min(2,2(a-2))}^{2(a-1)} c_{in}^{k-s} \cdot c_{out}^s \right) \right) \tag{28}
$$

$$
+ \frac{1}{n \cdot 2^{k-1}} \sum_{\frac{k+1}{2}}^{k+1} O\left( \binom{k-1}{a-2} \cdot \left( \sum_{s=\min(2,2(k+1-a))}^{2(k+1-a)} c_{in}^{k-s} \cdot c_{out}^s \right) \right)
$$

$$
= \frac{(k-1)!}{n \cdot 2^{k-1}} \sum_{a=2}^{k+1} O\left( \sum_{s=\min(2,2(a-2),2(k+1-a))}^{\min(2(a-1),2(k+1-a))} c_{in}^{k-s} \cdot c_{out}^s \right).
$$

**Case 2:** $y_i \neq y_j$ and $k \geq 2$

For nodes $v_i$ and $v_j$, when they belong to different classes ($y_i \neq y_j$), we count the walks of length $k$ where there are $a$ nodes of the same class as $v_i$ and $k + 1 - a$ nodes of the class of $v_j$. We need to choose $a - 1$ nodes from the same cluster as $v_i$ and $k - a$ nodes from the cluster of $v_j$. The total number of ways to arrange these nodes in a walk is $(k-2)!$, as there are $k - 2$ positions to fill.

The number of ways to choose the nodes is:

- Choose $a - 1$ nodes from $\frac{n}{2} - 1$ nodes in the same cluster as $v_i$: $\binom{\frac{n}{2} - 1}{a - 1}$;

- Choose $k - a$ nodes from $\frac{n}{2} - 1$ nodes in the same cluster as $v_j$: $\binom{\frac{n}{2} - 1}{k - a}$.

The number of ways to arrange these nodes is $(k - 1)!$. Considering the class changes in the $k$-length walk, let $s$ denote the number of class changes. The minimum and maximum values of $s$ are:

- If $2a \geq k + 1$, then $s_{\min} = 1$ and $s_{\max} = 2(k - a) + 1$;

- If $2a \leq k + 1$, then $s_{\min} = 1$ and $s_{\max} = 2a - 1$.

The probability of a $k$-length walk with $a$ nodes sharing the same class as $v_i$ is:

$$p_k^a(v_i, v_j | y_i \neq y_j) =$$
$$\begin{cases} \binom{\frac{n}{2} - 1}{a - 1} \cdot \binom{\frac{n}{2} - 1}{k - a} \cdot (k - 1)! \cdot \left( \sum_{s=1}^{2(k-a)+1} \left( \frac{c_{in}}{n} \right)^{k-s} \cdot \left( \frac{c_{out}}{n} \right)^s \right), & \text{if } 2a \geq k + 1 \\ \binom{\frac{n}{2} - 1}{a - 1} \cdot \binom{\frac{n}{2} - 1}{k - a} \cdot (k - 1)! \cdot \left( \sum_{s=1}^{2a-1} \left( \frac{c_{in}}{n} \right)^{k-s} \cdot \left( \frac{c_{out}}{n} \right)^s \right), & \text{if } 2a < k + 1 \end{cases}$$

The total probability of a $k$-length walk connecting $v_i$ and $v_j$ when $y_i \neq y_j$ is:

$$p_k(v_i, v_j | y_i \neq y_j) =$$

$$\sum_{a=1}^{\frac{k+1}{2}} \binom{\frac{n}{2} - 1}{a - 1} \cdot \binom{\frac{n}{2} - 1}{k - a} \cdot (k - 1)! \cdot \left( \sum_{s=1}^{2a-1} \left( \frac{c_{in}}{n} \right)^{k-s} \cdot \left( \frac{c_{out}}{n} \right)^s \right)$$

$$+ \sum_{a=\frac{k+1}{2}}^{k} \binom{\frac{n}{2} - 1}{a - 1} \cdot \binom{\frac{n}{2} - 1}{k - a} \cdot (k - 1)! \cdot \left( \sum_{s=1}^{2(k-a)+1} \left( \frac{c_{in}}{n} \right)^{k-s} \cdot \left( \frac{c_{out}}{n} \right)^s \right) \tag{29}$$

When $k \ll n$, using Lemma 27, we have

$$\binom{\frac{n}{2} - 1}{a - 1} = \frac{(\frac{n}{2} - 1)^{a-1}}{(a - 1)!}, \binom{\frac{n}{2} - 1}{k - a} = \frac{(\frac{n}{2} - 1)^{k-a}}{(k - a)!}.$$

Then:

$$\binom{\frac{n}{2} - 1}{a - 1} \cdot \binom{\frac{n}{2} - 1}{k - a} \cdot (k - 1)! = \frac{(\frac{n}{2} - 1)^{a-1}}{(a - 1)!} \cdot \frac{(\frac{n}{2} - 1)^{k-a}}{(k - a)!} \cdot (k - 1)!$$

$$= \left( \frac{n}{2} - 1 \right)^{k-1} \cdot \binom{k - 1}{a - 1}$$

We simplify Eq. (29) to

$$p_k(v_i, v_j | y_i \neq y_j) = \sum_{a=1}^{\frac{k+1}{2}} \left( \frac{n}{2} - 1 \right)^{k-1} \cdot \binom{k - 1}{a - 1} \cdot \left( \sum_{s=1}^{2a-1} \left( \frac{c_{in}}{n} \right)^{k-s} \cdot \left( \frac{c_{out}}{n} \right)^s \right)$$

$$+ \sum_{a=\frac{k+1}{2}}^{k} \left( \frac{n}{2} - 1 \right)^{k-1} \cdot \binom{k - 1}{a - 1} \cdot \left( \sum_{s=1}^{2(k-a)+1} \left( \frac{c_{in}}{n} \right)^{k-s} \cdot \left( \frac{c_{out}}{n} \right)^s \right)$$

$$= \frac{1}{n \cdot 2^{k-1}} \sum_{a=1}^{\frac{k+1}{2}} O \left( \binom{k - 1}{a - 1} \cdot \left( \sum_{s=1}^{2a-1} c_{in}^{k-s} \cdot c_{out}^s \right) \right) \tag{30}$$

$$+ \frac{1}{n \cdot 2^{k-1}} \sum_{a=\frac{k+1}{2}}^{k} O \left( \binom{k - 1}{a - 1} \cdot \left( \sum_{s=1}^{2(k-a)+1} c_{in}^{k-s} \cdot c_{out}^s \right) \right)$$

$$= \frac{(k - 1)!}{n \cdot 2^{k-1}} \sum_{a=1}^{k} O \left( \sum_{s=1}^{\min(2a-1, 2(k-a)+1)} c_{in}^{k-s} \cdot c_{out}^s \right).$$

**Case 3:** $k = 1$

When $k = 1$, we have $A^k = A$ and

$$\mathbb{E}[A_{ij}] = \begin{cases} \frac{c_{in}}{n}, & \text{if } y_i = y_j, \\ \frac{c_{out}}{n}, & \text{if } y_i \neq y_j. \end{cases}$$

In the following, we show that when a graph is sparse and $k$ is small, $A_{ij}^k$ can be modeled using a Poisson distribution.

- For sparse graphs with a large number of nodes ($n \to \infty, d \ll n$), the probability of a potential $k$-length walk existing is very small.

- When $k \ll n$, the dependence between two different $k$-length walks is negligible.

- The number of potential $k$-length walks is large ($n^{k-1}$ as $n \to \infty$).

Thus, according to Lemma 26, the number of $k$-length walks connecting nodes $v_i$ and $v_j$, $A_{ij}^k$, follows a Poisson distribution $Poisson(\rho')$ when $k \geq 2$, where:

$$\rho' = \begin{cases} \rho_= = \frac{(k-1)!}{n \cdot 2^{k-1}} \sum_{a=2}^{k+1} O\left( \sum_{s=\min(2,2(a-2),2(k+1-a))}^{\min(2(a-1),2(k+1-a))} c_{in}^{k-s} \cdot c_{out}^s \right), & \text{if } y_i = y_j, \\ \rho_{\neq} = \frac{(k-1)!}{n \cdot 2^{k-1}} \sum_{a=1}^{k} O\left( \sum_{s=1}^{\min(2a-1,2(k-a)+1)} c_{in}^{k-s} \cdot c_{out}^s \right), & \text{if } y_i \neq y_j. \end{cases}$$

When $k = 1$, $p(v_i, v_j)$ follows a Bernoulli distribution $Ber(p)$, where:

$$p = \begin{cases} \frac{c_{in}}{n}, & \text{if } y_i = y_j, \\ \frac{c_{out}}{n}, & \text{if } y_i \neq y_j. \end{cases}$$

This completes the proof.

$\square$

**Theorem 32** (Expectation and variance of $(\tilde{A}^k XW)_{ij}$)**.** *Given a graph generated by $G \sim cSBM(n, f, \mu, u, \lambda, d)$. The input node feature matrix is $X$ and the normalized adjacency matrix is $\tilde{A}$. The $k$-th power matrix $\tilde{A}^k$ is applied to obtain a new feature matrix $\tilde{A}^k XW$, then the expectation and the variance of $(\tilde{A}^k XW)_{ij}$ are as follows:*

*For $k = 1$:*

$$\mathbb{E}\left[(\tilde{A}^k XW)_{ij}\right] = \frac{1}{2d}\sqrt{\frac{\mu}{n}}\, (c_{in} - c_{out})\, y_i u W_{:j}$$

$$\mathbb{V}\left[(\tilde{A}^k XW)_{ij}\right] = \frac{1}{2 \cdot d^2}\left(d - \frac{c_{in}^2 + c_{out}^2}{n}\right) \cdot \left(\frac{\mu}{n}(uW_{:j})^2 + \frac{||W_{:j}||_2^2}{f}\right)$$

*For $k \geq 2$:*

$$\mathbb{E}\left[(\tilde{A}^k XW)_{ij}\right] = \frac{(k-1)!}{d^k \cdot 2^{k-1}} O\left(c_{in}^k - c_{out}^k\right)\sqrt{\frac{\mu}{n}}\, y_i u W_{:j}$$

$$\mathbb{V}\left[(\tilde{A}^k XW)_{ij}\right] = \frac{(k-1)!}{d^{2k} \cdot 2^k}\left(\sum_{a=2}^{k+1} O\left(\sum_{s=\min(2,2(a-2),2(k+1-a))}^{\min(2(a-1),2(k+1-a))} c_{in}^{k-s} \cdot c_{out}^s\right)\right.$$
$$\left. + \sum_{a=1}^{k} O\left(\sum_{s=1}^{\min(2a-1,2(k-a)+1)} c_{in}^{k-s} \cdot c_{out}^s\right)\right)\left(\frac{\mu}{n}(uW_{:j})^2 + \frac{||W_{:j}||_2^2}{f}\right)$$

*Proof.* Given that the node feature $x_i$ for node $v_i$, generated by a conditional Stochastic Block Model (cSBM) conditioned on $u$ and node class $y_i$, is distributed as:

$$x_i \sim \mathcal{N}\left(\sqrt{\frac{\mu}{n}}y_i u, \frac{I_f}{f}\right)$$

For a linear transformation matrix $W$, the transformed node feature is given by:

$$x_i W \sim \mathcal{N}\left(\sqrt{\frac{\mu}{n}}y_i u W, \frac{W^T W}{f}\right)$$

Feature after transformation with $W$ and propagation with $\tilde{A}^k$ is

$$\left(\tilde{A}^k X W\right)_{ij} = \sum_{r=1}^{n} \tilde{A}_{ir}^k (XW)_{rj}$$

$$= \sum_{r=1}^{n} \tilde{A}_{ir}^k \left(\sqrt{\frac{\mu}{n}}y_r u W_{:j} + \frac{\epsilon_r W_{:j}}{\sqrt{f}}\right)$$

$$= \sum_{r=1}^{n} \tilde{A}_{ir}^k \sqrt{\frac{\mu}{n}}\, y_r u W_{:j}$$

and

$$\mathbb{E}\left[\left(\tilde{A}^k X W\right)_{ij}\right] = \sqrt{\frac{\mu}{n}}\left(\sum_{r=1}^{n} \mathbb{E}\left[\tilde{A}_{ir}^k\right] y_r\right) u W_{:j} \tag{31}$$

We now derive the expectation $\mathbb{E}[A_{ij}^k]$ of the adjacency matrix $A$ raised to the power $k$.

**1. Expectation $\mathbb{E}\left[\left(\tilde{A}^k X W\right)_{ij}\right]$ when $k \geq 2$**

Two clusters generated by cSBM are in equal size. According to Theorem 31, we have

$$\mathbb{E}\left[\left(\tilde{A}^k X W\right)_{ij}\right] = \sqrt{\frac{\mu}{n}}\left(\sum_{r=1}^{n} \mathbb{E}\left[\tilde{A}_{ir}^k\right] y_r\right) u W_{:j}$$

$$= \frac{1}{d^k}\sqrt{\frac{\mu}{n}}\left(\sum_{r=1}^{n}\left(\mathbb{E}\left[A_{ir}^k | y_i = y_r\right] + \mathbb{E}\left[A_{ir}^k | y_i \neq y_r\right]\right) y_r\right) u W_{:j}$$

$$= \frac{1}{d^k}\sqrt{\frac{\mu}{n}}\Bigg(\sum_{r=1}^{n}\bigg(\frac{(k-1)!}{n \cdot 2^{k-1}}\sum_{a=2}^{k+1} O\Bigg(\sum_{s=\min(2,2(a-2),2(k+1-a))}^{\min(2(a-1),2(k+1-a))} c_{in}^{k-s} \cdot c_{out}^{s}\Bigg)$$

$$+ \frac{(k-1)!}{n \cdot 2^{k-1}}\sum_{a=1}^{k} O\Bigg(\sum_{s=1}^{\min(2a-1,2(k-a)+1)} c_{in}^{k-s} \cdot c_{out}^{s}\Bigg)\bigg) y_r\Bigg) u W_{:j}$$

$$= \frac{(k-1)!}{d^k \cdot 2^{k-1}} O\Bigg(\sum_{a=2}^{k+1}\sum_{s=\min(2,2(a-2),2(k+1-a))}^{\min(2(a-1),2(k+1-a))} c_{in}^{k-s} \cdot c_{out}^{s}$$

$$- \sum_{a=1}^{k}\sum_{s=1}^{\min(2a-1,2(k-a)+1)} c_{in}^{k-s} \cdot c_{out}^{s}\Bigg)\sqrt{\frac{\mu}{n}}y_i u W_{:j}$$

$$= \frac{(k-1)!}{d^k \cdot 2^{k-1}} O\left(c_{in}^k - c_{out}^k\right)\sqrt{\frac{\mu}{n}}y_i u W_{:j}$$

**2. Variance $\mathbb{E}\left[\left(\tilde{A}^k X W\right)_{ij}\right]$ when $k \geq 2$**

The variance of new feature $X'_{ij}$ given $u, Y$ can be expressed as:

$$
\mathbb{V}\left[(\tilde{A}^k XW)_{ij}\right] = \mathbb{V}\left[\sum_{r=1}^n \tilde{A}_{ir}^k (\sqrt{\frac{\mu}{n}} y_r u W_{:j} + \frac{\epsilon_r W_{:j}}{\sqrt{f}})\right]
$$

$$
= \sum_{r=1}^n \mathbb{V}\left[\tilde{A}_{ir}^k (\sqrt{\frac{\mu}{n}} y_r u W_{:j} + \frac{\epsilon_r W_{:j}}{\sqrt{f}})\right], \quad \text{feature dimension independent}
$$

$$
= \sum_{r=1}^n \left[ \mathbb{E}\left[(\tilde{A}^k)_{ir}^2\right] \mathbb{E}\left[(\sqrt{\frac{\mu}{n}} y_r u W_{:j} + \frac{\epsilon_r W_{:j}}{\sqrt{f}})^2\right] - \left(\mathbb{E}\left[\tilde{A}_{ir}^k\right]\right)^2 \left(\mathbb{E}\left[\sqrt{\frac{\mu}{n}} y_r u W_{:j} + \frac{\epsilon_r W_{:j}}{\sqrt{f}}\right]\right)^2 \right]
$$

$$
= \sum_{r=1}^n \left[ \mathbb{E}\left[(\tilde{A}^k)_{ir}^2\right] \left(\left(\sqrt{\frac{\mu}{n}} y_r u W_{:j}\right)^2 + \frac{||W_{:j}||_2^2}{f}\right) - \left(\mathbb{E}\left[\tilde{A}_{ir}^k\right]\right)^2 (\mathbb{E}\left[\sqrt{\frac{\mu}{n}} y_r u W_{:j} + \frac{\epsilon_r W_{:j}}{\sqrt{f}}\right])^2 \right]
$$

$$
= \sum_{r=1}^n \left[ \mathbb{E}\left[(\tilde{A}^k)_{ir}^2\right] \left(\left(\sqrt{\frac{\mu}{n}} y_r u W_{:j}\right)^2 + \frac{||W_{:j}||_2^2}{f}\right) - \left(\mathbb{E}\left[\tilde{A}_{ir}^k\right]\right)^2 \left(\sqrt{\frac{\mu}{n}} y_r u W_{:j}\right)^2 \right]
$$

$$
= \sum_{r=1}^n \left[ \left(\left(\mathbb{E}\left[\tilde{A}_{ir}^k\right]\right)^2 + \mathbb{V}\left[\tilde{A}_{ir}^k\right]\right) \cdot \left(\left(\sqrt{\frac{\mu}{n}} y_r u W_{:j}\right)^2 + \frac{||W_{:j}||_2^2}{f}\right) \right.
$$

$$
\left. - \left(\mathbb{E}\left[\tilde{A}_{ir}^k\right]\right)^2 \left(\sqrt{\frac{\mu}{n}} y_r u W_{:j}\right)^2 \right]
$$

$$
= \frac{1}{d^{2k}} \sum_{r=1}^n \left[ \left(\left(\mathbb{E}\left[A_{ir}^k\right]\right)^2 + \mathbb{V}\left[A_{ir}^k\right]\right) \cdot \left(\frac{\mu}{n} (u W_{:j})^2 + \frac{||W_{:j}||_2^2}{f}\right) \right.
$$

$$
\left. - \left(\mathbb{E}\left[A_{ir}^k\right]\right)^2 \frac{\mu}{n} (u W_{:j})^2 \right]
$$

$$
= \frac{1}{d^{2k}} \sum_{r=1}^n \left[ \left(\mathbb{E}\left[A_{ir}^k\right]\right)^2 \cdot \frac{||W_{:j}||_2^2}{f} + \mathbb{V}\left[A_{ir}^k\right] \cdot \left(\frac{\mu}{n} (u W_{:j})^2 + \frac{||W_{:j}||_2^2}{f}\right) \right]
$$

$$
= \frac{1}{d^{2k}} \frac{n}{2} \left(\left(\mathbb{E}\left[A_{ir}^k | y_i = y_r\right]\right)^2 + \left(\mathbb{E}\left[A_{ir}^k | y_i \neq y_r\right]^2\right)\right) \cdot \frac{||W_{:j}||_2^2}{f}
$$

$$
+ \frac{1}{d^{2k}} \frac{n}{2} \left(\mathbb{V}\left[A_{ir}^k | y_i = y_r\right] + \mathbb{V}\left[A_{ir}^k | y_i \neq y_r\right]\right) \cdot \left(\frac{\mu}{n} (u W_{:j})^2 + \frac{||W_{:j}||_2^2}{f}\right)
$$

$$
\tag{32}
$$

According to Theorem 31, when $k \geq 2$, we have

$$
\left(\mathbb{E}\left[A_{ij}^k | y_i = y_j\right]\right)^2 = \left(\frac{(k-1)!}{n \cdot 2^{k-1}} \sum_{a=2}^{k+1} O\left(\sum_{s=\min(2,2(a-2),2(k+1-a))}^{\min(2(a-1),2(k+1-a))} c_{in}^{k-s} \cdot c_{out}^s\right)\right)^2
$$

$$
\left(\mathbb{E}\left[A_{ij}^k | y_i \neq y_j\right]\right)^2 = \left(\frac{(k-1)!}{n \cdot 2^{k-1}} \sum_{a=1}^{k} O\left(\sum_{s=1}^{\min(2a-1,2(k-a)+1)} c_{in}^{k-s} \cdot c_{out}^s\right)\right)^2
$$

Two clusters generated by cSBM are in equal size. Then, Eq. (32) is written as:

$$\mathbb{V}\left[(\tilde{A}^k XW)_{ij}\right] = \frac{1}{d^{2k}} \frac{n}{2} \left( \left( \mathbb{E}\left[A_{ir}^k | y_i = y_r\right]\right)^2 + \left(\mathbb{E}\left[A_{ir}^k | y_i \neq y_r\right]\right)^2 \right) \cdot \frac{||W_{:j}||_2^2}{f}$$

$$+ \frac{1}{d^{2k}} \frac{n}{2} \left( \mathbb{V}\left[A_{ir}^k | y_i = y_r\right] + \mathbb{V}\left[A_{ir}^k | y_i \neq y_r\right] \right) \cdot \left( \frac{\mu}{n} \left(uW_{:j}\right)^2 + \frac{||W_{:j}||_2^2}{f} \right)$$

$$= \frac{((k-1)!)^2}{n \cdot d^{2k} \cdot 2^{2k-1}} O\Bigg( \left( \sum_{a=2}^{k+1} O\left( \sum_{s=\min(2,2(a-2),2(k+1-a))}^{\min(2(a-1),2(k+1-a))} c_{in}^{k-s} \cdot c_{out}^s \right) \right)^2$$

$$+ \left( \sum_{a=1}^{k} O\left( \sum_{s=1}^{\min(2a-1,2(k-a)+1)} c_{in}^{k-s} \cdot c_{out}^s \right) \right)^2 \Bigg) \cdot \frac{||W_{:j}||_2^2}{f}$$

$$+ \frac{(k-1)!}{d^{2k} \cdot 2^k} \Bigg( \sum_{a=2}^{k+1} O\left( \sum_{s=\min(2,2(a-2),2(k+1-a))}^{\min(2(a-1),2(k+1-a))} c_{in}^{k-s} \cdot c_{out}^s \right)$$

$$+ \sum_{a=1}^{k} O\left( \sum_{s=1}^{\min(2a-1,2(k-a)+1)} c_{in}^{k-s} \cdot c_{out}^s \right) \Bigg) \left( \frac{\mu}{n} \left(uW_{:j}\right)^2 + \frac{||W_{:j}||_2^2}{f} \right)$$

$$= \frac{(k-1)!}{d^{2k} \cdot 2^k} \Bigg( \sum_{a=2}^{k+1} O\left( \sum_{s=\min(2,2(a-2),2(k+1-a))}^{\min(2(a-1),2(k+1-a))} c_{in}^{k-s} \cdot c_{out}^s \right)$$

$$+ \sum_{a=1}^{k} O\left( \sum_{s=1}^{\min(2a-1,2(k-a)+1)} c_{in}^{k-s} \cdot c_{out}^s \right) \Bigg) \left( \frac{\mu}{n} \left(uW_{:j}\right)^2 + \frac{||W_{:j}||_2^2}{f} \right), \quad n \to \infty$$

**3. Expectation and variance of $\left(\tilde{A}^k XW\right)_{ij}$ when $k = 1$**

$$\mathbb{E}\left[(\tilde{A}XW)_{ij}\right] = \sqrt{\frac{\mu}{n}} \left( \sum_{r=1}^{n} \mathbb{E}\left[\tilde{A}_{ir}\right] y_r \right) uW_{:j}$$

$$= \frac{1}{d} \sqrt{\frac{\mu}{n}} \left( \sum_{r=1}^{n} \mathbb{E}\left[A_{ir} | y_i = y_r\right] y_i - \sum_{r=1}^{n} \mathbb{E}\left[A_{ir} | y_i \neq y_r\right] y_i \right) uW_{:j}$$

$$= \frac{1}{d} \sqrt{\frac{\mu}{n}} \left( \frac{n}{2} \frac{c_{in}}{n} y_i - \frac{n}{2} \frac{c_{out}}{n} y_i \right) uW_{:j}$$

$$= \frac{1}{2d} \sqrt{\frac{\mu}{n}} \left( c_{in} - c_{out} \right) y_i uW_{:j}$$

when $k = 1$, we have

$$\left( \mathbb{E}\left[A_{ij}^k | y_i = y_j\right] \right)^2 = \left( \frac{c_{in}}{n} \right)^2$$

$$\left( \mathbb{E}\left[A_{ij}^k | y_i \neq y_j\right] \right)^2 = \left( \frac{c_{out}}{n} \right)^2$$

Eq. (32) is written as:

$$\mathbb{V}\left[(\tilde{A}XW)_{ij}\right] = \frac{1}{d^2}\frac{n}{2}\left(\left(\mathbb{E}\left[A_{ir}^k|y_i = y_r\right]\right)^2 + \left(\mathbb{E}\left[A_{ir}^k|y_i \neq y_r\right]\right)^2\right)\cdot\frac{||W_{:j}||_2^2}{f}$$

$$+ \frac{1}{d^2}\frac{n}{2}\left(\mathbb{V}\left[A_{ir}^k|y_i = y_r\right] + \mathbb{V}\left[A_{ir}^k|y_i \neq y_r\right]\right)\cdot\left(\frac{\mu}{n}\left(uW_{:j}\right)^2 + \frac{||W_{:j}||_2^2}{f}\right)$$

$$= \frac{1}{d^2}\frac{n}{2}\left(\left(\frac{c_{in}}{n}\right)^2 + \left(\frac{c_{out}}{n}\right)^2\right)\cdot\frac{||W_{:j}||_2^2}{f}$$

$$+ \frac{1}{d^2}\frac{n}{2}\left(\frac{c_{in}}{n}\left(1 - \frac{c_{in}}{n}\right) + \frac{c_{out}}{n}\left(1 - \frac{c_{out}}{n}\right)\right)\cdot\left(\frac{\mu}{n}\left(uW_{:j}\right)^2 + \frac{||W_{:j}||_2^2}{f}\right)$$

$$= \frac{1}{2n\cdot d^2}\left(c_{in}^2 + c_{out}^2\right)\cdot\frac{||W_{:j}||_2^2}{f}$$

$$+ \frac{1}{2\cdot d^2}\left(d - \frac{c_{in}^2 + c_{out}^2}{n}\right)\cdot\left(\frac{\mu}{n}\left(uW_{:j}\right)^2 + \frac{||W_{:j}||_2^2}{f}\right)$$

$$= \frac{1}{2\cdot d^2}\left(d - \frac{c_{in}^2 + c_{out}^2}{n}\right)\cdot\left(\frac{\mu}{n}\left(uW_{:j}\right)^2 + \frac{||W_{:j}||_2^2}{f}\right), \quad n \to \infty$$

$\square$

### D.3   PROOF OF THEOREM 13

We first give a lemma about the order of $\mathbb{E}\left[A_{ij}^k\right]$, which will be used in proof of Theorem 13.

**Lemma 33** (order of $\mathbb{E}\left[A_{ij}^k\right]$). *The order of $\mathbb{E}\left[A_{ij}^k\right]$ is $O\left(\frac{k!\cdot d^k}{n\cdot 2^k}\right)$.*

*Proof.* According to Theorem 31, $A_{ij}^k|y_i = y_j$ and $A_{ij}^k|y_i \neq y_j$ obeys different Poisson distributions. As

$$c_{in}^{k-s}\cdot c_{out}^s = O\left(d^k\right),$$

we have,

$$\rho_= = \frac{(k-1)!}{n\cdot 2^{k-1}}\sum_{a=2}^{k+1}O\left(\sum_{s=\min(2,2(a-2),2(k+1-a))}^{\min(2(a-1),2(k+1-a))}c_{in}^{k-s}\cdot c_{out}^s\right)$$

$$= \frac{(k-1)!}{n\cdot 2^{k-1}}\sum_{a=2}^{k+1}O\left(\sum_{s=\min(2,2(a-2),2(k+1-a))}^{\min(2(a-1),2(k+1-a))}d^k\right)$$

$$= \frac{(k-1)!}{n\cdot 2^{k-1}}\sum_{a=2}^{k+1}O\left(k\cdot d^k\right)$$

$$= \frac{(k-1)!}{n\cdot 2^{k-1}}O\left(k^2\cdot d^k\right)$$

$$= O\left(\frac{k!\cdot d^k}{n\cdot 2^k}\right)$$

similarly, we have $\rho_{\neq} = O\left(\frac{k!\cdot d^k}{n\cdot 2^k}\right)$

$\square$

Below, we prove Theorem 13, which is a specific case of Theorem 8 when the graph is generated by $G \sim cSBM(n, f, \mu, u, \lambda, d)$.

**Theorem 13.** *Consider a spectral GNN $\Psi$ parameterized by $\Theta, W$ trained using full-batch gradient descent for $T$ iterations with a learning rate $\eta$ on a training dataset containing $m$ samples drawn from nodes on a graph $G \sim cSBM(n, f, \mu, u, \lambda, d)$. When $n \to \infty$, $k \ll n$, and $d \ll n$, under*

*Assumptions 1, 2, and 4, for any node $v_i$ on the graph, with probability at least $1 - \epsilon$ for a constant $\epsilon \in (0, 1)$, $\Psi$ satisfies $\gamma$-uniform transductive stability, where $\gamma = r\beta$ and*

$$\beta = \frac{1}{\epsilon}\left[ O\left(\mathbb{E}\left[\|\hat{y}_i - y_i\|_F^2\right]\right) + O\left(\sum_{k=2}^{K}\left(\mathbb{E}\left[\left(A_{ij}^k \mid y_i = y_j\right)^2\right] + \mathbb{E}\left[\left(A_{ij}^k \mid y_i \neq y_j\right)^2\right]\right)\right)\right].$$

*Proof.* Any spectral GNNs in Eq. (1) with linear feature transformation function, and polynomial basis expanded on normalized graph matrix can be transformed into the format:

$$\hat{Y} = softmax(\sum_{k=0}^{K} \theta_k \tilde{A}^k XW) \tag{33}$$

where $\tilde{A} = D^{-\frac{1}{2}}AD^{-\frac{1}{2}}$ is the normalized graph adjacency matrix, $D$ is the diagonal degree matrix. We denotes $Y \in \mathbb{R}^{n \times C}$ as the ground truth node label matrix.

When graph $G \sim cSBM(n, f, \mu, u, \lambda, d)$, the node feature

$$x_i \sim \mathcal{N}(y_i\sqrt{\mu/n}u, I_f/f)$$

Denote $B = XW$ and $S = BB^\top$, then we have

$$B_{ik} \sim \mathcal{N}(y_i\sqrt{\frac{\mu}{n}}uW_{:k}, \frac{\|W_{:k}\|_F^2}{f})$$

- when $i \neq j$, $B_{ik}, B_{jk}$ are independent, then

$$\mathbb{E}\left[S_{ij}\right] = \sum_{k=1}^{C}\mathbb{E}\left[B_{ik}B_{kj}^\top\right]$$

$$= \sum_{k=1}^{C}y_iy_j\frac{\mu}{n}\left(uW_{:k}\right)^2$$

$$= y_iy_j\frac{\mu}{n}\|uW\|_F^2;$$

- when $i = j$:

$$\mathbb{E}\left[S_{ii}\right] = \frac{\mu}{n}\|uW\|_F^2 + \frac{\|W\|_F^2}{f}$$

When node number $n \to \infty$, we have

$$\sum_{q=1,q\neq j}^{n}\mathbb{E}\left[S_{jq}\right] = \frac{n}{2}y_j^2\frac{\mu}{n}\|uW\|_F^2 + \frac{n}{2}y_j(-y_j)\frac{\mu}{n}\|uW\|_F^2 = 0.$$

Therefore,

$$\sum_{j=1}^{n}\sum_{q=1,q\neq j}^{n}\mathbb{E}\left[A_{ij}^kA_{iq}^k\right]\mathbb{E}\left[S_{jq}\right]$$

$$= \frac{n^2}{4}\rho_{k=}^2\frac{\mu}{n}\|uW\|_F^2; \quad (y_i = y_j = y_q)$$

$$+ \frac{n^2}{4}\rho_{k=}\rho_{k\neq}-\frac{\mu}{n}\|uW\|_F^2; \quad (y_i = y_j \neq y_q)$$

$$+ \frac{n^2}{4}\rho_{k\neq}\rho_{k=}-\frac{\mu}{n}\|uW\|_F^2; \quad (y_i \neq y_j = y_q) \tag{34}$$

$$+ \frac{n^2}{4}\rho_{k\neq}^2\frac{\mu}{n}\|uW\|_F^2; \quad (y_i = y_q \neq y_j)$$

$$= \frac{n^2}{4}\cdot\frac{\mu}{n}\|uW\|_F^2\cdot\left(\rho_{k=}^2 - 2\rho_{k\neq}\rho_{k=} + \rho_{k\neq}^2\right)$$

$$= \frac{n^2}{4}\cdot\frac{\mu}{n}\|uW\|_F^2\cdot\left(\rho_{k=} - \rho_{k\neq}\right)^2$$

According to Theorem 31,

- when $k \geq 2$, $A_{ij}^k \sim Poisson(\rho_k')$, then

$$
\begin{aligned}
\mathbb{E}\left[\|\tilde{A}_{i:}^k XW\|_F^2\right] &= \mathbb{E}\left[\tilde{A}_{i:}^k XW\left(XW\right)^\top \left(\tilde{A}_{i:}^k\right)^\top\right] \\
&= \mathbb{E}\left[\tilde{A}_{i:}^k S\left(\tilde{A}_{i:}^k\right)^\top\right] \\
&= \mathbb{E}\left[\sum_{q=1}^n \sum_{j=1}^n \left(\tilde{A}_{ij}^k \tilde{A}_{iq}^k S_{jq}\right)\right] \\
&= \frac{1}{d^{2k}}\mathbb{E}\left[\sum_{q=1}^n \sum_{j=1}^n \left(A_{ij}^k A_{iq}^k S_{jq}\right)\right] \\
&= \frac{1}{d^{2k}}\sum_{q=1}^n \sum_{j=1}^n \mathbb{E}\left[A_{ij}^k A_{iq}^k\right]\mathbb{E}\left[S_{jq}\right] \\
&= \frac{1}{d^{2k}}\sum_{j=1}^n \mathbb{E}\left[\left(A_{ij}^k\right)^2\right]\mathbb{E}\left[S_{jj}\right] + \frac{1}{d^{2k}}\sum_{j=1}^n \sum_{q=1,q\neq j}^n \mathbb{E}\left[A_{ij}^k A_{iq}^k\right]\mathbb{E}\left[S_{jq}\right] \\
&= \frac{1}{d^{2k}}\frac{n}{2}\mathbb{E}\left[\left(A_{ij}^k\right)^2 \mid y_i = y_j\right]\mathbb{E}\left[S_{jj}\right] + \frac{1}{d^{2k}}\frac{n}{2}\mathbb{E}\left[\left(A_{ij}^k\right)^2 \mid y_i \neq y_j\right]\mathbb{E}\left[S_{jj}\right] \\
&\quad + \frac{1}{d^{2k}}\frac{n^2}{4}\cdot\frac{\mu}{n}\|uW\|_F^2\cdot(\rho_{k=}-\rho_{k\neq})^2 \quad (Eq.\ (34)) \\
&= \frac{1}{d^{2k}}\frac{n}{2}\left(\rho_{k=}+\rho_{k=}^2+\rho_{k\neq}+\rho_{k\neq}^2\right)\left(\frac{\mu}{n}\|uW\|_F^2 + \frac{\|W\|_F^2}{f}\right) \\
&\quad + \frac{1}{d^{2k}}\frac{n^2}{4}\cdot\frac{\mu}{n}\|uW\|_F^2\cdot(\rho_{k=}-\rho_{k\neq})^2 \\
&= \frac{1}{2d^{2k}}\zeta_k\left(\mu\|uW\|_F^2 + \frac{n\|W\|_F^2}{f}\right) + \frac{n\mu}{4d^{2k}}\|uW\|_F^2\cdot(\rho_{k=}-\rho_{k\neq})^2
\end{aligned}
$$

where $\zeta_k = \rho_{k=}^2 + \rho_{k=} + \rho_{k\neq}^2 + \rho_{k\neq}$

- when $k = 1$, $A_{ij} \sim Ber(p)$, then

$$
\begin{aligned}
\mathbb{E}\left[\|\tilde{A}_{i:}XW\|_F^2\right] &= \frac{1}{d^2}\frac{n}{2}\left(p_=^2 + p_=(1-p_=) + p_{\neq}^2 + p_{\neq}(1-p_{\neq})\right)\left(\frac{\mu}{n}\|uW\|_F^2 + \frac{\|W\|_F^2}{f}\right) \\
&= \frac{1}{d^2}\frac{n}{2}\left(p_= + p_{\neq}\right)\left(\frac{\mu}{n}\|uW\|_F^2 + \frac{\|W\|_F^2}{f}\right) \\
&= \frac{1}{d^2}\frac{n}{2}\frac{2d}{n}\left(\frac{\mu}{n}\|uW\|_F^2 + \frac{\|W\|_F^2}{f}\right) \\
&= \frac{1}{d}\left(\frac{\mu}{n}\|uW\|_F^2 + \frac{\|W\|_F^2}{f}\right)
\end{aligned}
$$

Substituting $\mathbb{E}\left[\|\tilde{A}_{i:}XW\|_F^2\right]$ into Eq. (15), we have

$$
\mathbb{E}\left[\left|\frac{\partial\ell(\hat{y}_i, y_i; \Theta, W)}{\partial\theta_k}\right|\right] =
$$

$$
\begin{cases}
\frac{1}{2}\left(\mathbb{E}\left[\|\hat{y}_i - y_i\|_F^2\right] + \left(\frac{\mu}{n}\|uW\|_F^2 + \frac{\|W\|_F^2}{f}\right)\right), & \text{if } k = 0 \\
\frac{1}{2}\left(\mathbb{E}\left[\|\hat{y}_i - y_i\|_F^2\right] + \frac{1}{d}\left(\frac{\mu}{n}\|uW\|_F^2 + \frac{\|W\|_F^2}{f}\right)\right), & \text{if } k = 1 \\
\frac{1}{2}\left(\mathbb{E}\left[\|\hat{y}_i - y_i\|_F^2\right] + \frac{1}{2d^{2k}}\zeta_k\left(\mu\|uW\|_F^2 + \frac{n\|W\|_F^2}{f}\right) + \frac{n\mu}{4d^{2k}}\|uW\|_F^2\cdot(\rho_{k=}-\rho_{k\neq})^2\right), & \text{if } k \geq 2
\end{cases}
$$
$$(35)$$

Similarly, we have

$$
\mathbb{E}\left[\|\tilde{A}_{i:}^k X\|_F^2\right] = \begin{cases} \frac{\mu}{n}\|u\|_F^2 + 1, & \text{if} \quad k = 0 \\ \frac{1}{d}\left(\frac{\mu}{n}\|u\|_F^2 + 1\right), & \text{if} \quad k = 1 \\ \frac{1}{2d^{2k}}\zeta_k\left(\mu\|u\|_F^2 + 1\right) + \frac{n\mu}{4d^{2k}}\|u\|_F^2 \cdot (\rho_{k=} - \rho_{k\neq})^2, & \text{if} \quad k \geq 2 \end{cases}
$$

Substituting $\mathbb{E}\left[\|\tilde{A}_{i:}^k X\|_F^2\right]$ into Eq. (16), we have

$$
\mathbb{E}\left[\|\frac{\partial \ell(\hat{y}_i, y_i; \Theta, W)}{\partial W}\|_{\ell_1}\right] = |\theta_0|\left(f \cdot \mathbb{E}\left[\|\hat{y}_i - y_i\|_F^2\right] + C\left(\frac{\mu}{n}\|u\|_F^2 + 1\right)\right)
$$
$$
+ |\theta_1|\left(f \cdot \mathbb{E}\left[\|\hat{y}_i - y_i\|_F^2\right] + C\frac{1}{d}\left(\frac{\mu}{n}\|u\|_F^2 + 1\right)\right)
$$
$$
+ \sum_{k=2}^K \frac{1}{d^{2k}}|\theta_k|\left(f \cdot \mathbb{E}\left[\|\hat{y}_i - y_i\|_F^2\right] + C\frac{1}{2d^{2k}}\zeta_k\left(\mu\|u\|_F^2 + 1\right) + \frac{n\mu}{4d^{2k}}\|u\|_F^2 \cdot (\rho_{k=} - \rho_{k\neq})^2\right)
$$
$$
\tag{36}
$$

Substitute Eq. (35), Eq. (36) into Eq. (12), we have

$$
\mathbb{E}\left[\|\nabla \ell(\hat{y}_i, y_i; \Theta, W)\|_F\right] \leq \sum_{k=0}^K \mathbb{E}\left[\|\frac{\partial \ell(\hat{y}_i, y_i; \Theta, W)}{\partial \theta_k}\|_{\ell_1}\right] + \mathbb{E}\left[\|\frac{\partial \ell(\hat{y}_i, y_i; \Theta, W)}{\partial W}\|_{\ell_1}\right]
$$
$$
= \frac{1}{2}\left(\mathbb{E}\left[\|\hat{y}_i - y_i\|_F^2\right] + \left(\frac{\mu}{n}\|uW\|_F^2 + \frac{\|W\|_F^2}{f}\right)\right)
$$
$$
+ \frac{1}{2}\left(\mathbb{E}\left[\|\hat{y}_i - y_i\|_F^2\right] + \frac{1}{d}\left(\frac{\mu}{n}\|uW\|_F^2 + \frac{\|W\|_F^2}{f}\right)\right)
$$
$$
+ \sum_{k=2}^K \frac{1}{2}\left(\mathbb{E}\left[\|\hat{y}_i - y_i\|_F^2\right] + \frac{1}{2d^{2k}}\zeta_k\left(\mu\|uW\|_F^2 + \frac{n\|W\|_F^2}{f}\right) + \frac{n\mu}{4d^{2k}}\|uW\|_F^2 \cdot \tilde{\zeta}_k^2\right) \quad (37)
$$
$$
+ |\theta_0|\left(f \cdot \mathbb{E}\left[\|\hat{y}_i - y_i\|_F^2\right] + C\left(\frac{\mu}{n}\|u\|_F^2 + 1\right)\right)
$$
$$
+ |\theta_1|\left(f \cdot \mathbb{E}\left[\|\hat{y}_i - y_i\|_F^2\right] + C\frac{1}{d}\left(\frac{\mu}{n}\|u\|_F^2 + 1\right)\right)
$$
$$
+ \sum_{k=2}^K \frac{1}{d^{2k}}|\theta_k|\left(f \cdot \mathbb{E}\left[\|\hat{y}_i - y_i\|_F^2\right] + C\frac{1}{2d^{2k}}\zeta_k\left(\mu\|u\|_F^2 + 1\right) + \frac{n\mu}{4d^{2k}}\|u\|_F^2 \cdot \tilde{\zeta}_k^2\right)
$$

where $\zeta_k = \rho_{k=}^2 + \rho_{k=} + \rho_{k\neq}^2 + \rho_{k\neq}, \tilde{\zeta}_k = \rho_{k=} - \rho_{k\neq}$.

According to Lemma 33, when $n \to \infty$, we have

$$
n\left(\tilde{\zeta}_k\right)^2 = n\left(\rho_= - \rho_{\neq}\right)^2
$$
$$
= n\left(O\left(\frac{k! \cdot d^k}{n \cdot 2^k}\right)\right)^2
$$
$$
= nO\left(\frac{\left(k! \cdot d^k\right)^2}{n^2 \cdot 2^{2k}}\right)
$$
$$
= O\left(\frac{\left(k! \cdot d^k\right)^2}{n \cdot 2^{2k}}\right)
$$
$$
\to 0
$$

Thus, $n\left(\tilde{\zeta}_k\right)^2$ can be neglected. Thus, we rewrite Eq. (37) as

$$
\begin{aligned}
\mathbb{E}\left[\|\nabla\ell(\hat{y}_i, y_i; \Theta, W)\|_F\right] &= \sum_{k=0}^{K} \mathbb{E}\left[\|\frac{\partial\ell(\hat{y}_i, y_i; \Theta, W)}{\partial\theta_k}\|_{\ell_1}\right] + \mathbb{E}\left[\|\frac{\partial\ell(\hat{y}_i, y_i; \Theta, W)}{\partial W}\|_{\ell_1}\right] \\
&= \frac{1}{2}\left(\mathbb{E}\left[\|\hat{y}_i - y_i\|_F^2\right] + \left(\frac{\mu}{n}\|uW\|_F^2 + \frac{\|W\|_F^2}{f}\right)\right) \\
&\quad + \frac{1}{2}\left(\mathbb{E}\left[\|\hat{y}_i - y_i\|_F^2\right] + \frac{1}{d}\left(\frac{\mu}{n}\|uW\|_F^2 + \frac{\|W\|_F^2}{f}\right)\right) \\
&\quad + \sum_{k=2}^{K}\frac{1}{2}\left(\mathbb{E}\left[\|\hat{y}_i - y_i\|_F^2\right] + \frac{1}{2d^{2k}}\zeta_k\left(\mu\|uW\|_F^2 + \frac{n\|W\|_F^2}{f}\right)\right) \\
&\quad + |\theta_0|\left(f \cdot \mathbb{E}\left[\|\hat{y}_i - y_i\|_F^2\right] + C\left(\frac{\mu}{n}\|u\|_F^2 + 1\right)\right) \\
&\quad + |\theta_1|\left(f \cdot \mathbb{E}\left[\|\hat{y}_i - y_i\|_F^2\right] + C\frac{1}{d^{2k-1}}\left(\frac{\mu}{n}\|u\|_F^2 + 1\right)\right) \\
&\quad + \sum_{k=2}^{K}\frac{1}{d^{2k}}|\theta_k|\left(f \cdot \mathbb{E}\left[\|\hat{y}_i - y_i\|_F^2\right] + C\frac{1}{2d^{2k}}\zeta_k\left(\mu\|u\|_F^2 + 1\right)\right) \\
&\leq \frac{1}{2}\left(\mathbb{E}\left[\|\hat{y}_i - y_i\|_F^2\right] + \left(\frac{\mu}{n}\|u\|_F^2 B_W^2 + \frac{B_W^2}{f}\right)\right) \\
&\quad + \frac{1}{2}\left(\mathbb{E}\left[\|\hat{y}_i - y_i\|_F^2\right] + \frac{1}{d}\left(\frac{\mu}{n}\|u\|_F^2 B_W^2 + \frac{B_W^2}{f}\right)\right) \\
&\quad + \sum_{k=2}^{K}\frac{1}{2}\left(\mathbb{E}\left[\|\hat{y}_i - y_i\|_F^2\right] + \frac{1}{2d^{2k}}\zeta_k\left(\mu\|u\|_F^2 B_W^2 + \frac{nB_W^2}{f}\right)\right) \\
&\quad + B_\Theta\left(f \cdot \mathbb{E}\left[\|\hat{y}_i - y_i\|_F^2\right] + C\left(\frac{\mu}{n}\|u\|_F^2 + 1\right)\right) \\
&\quad + B_\Theta\left(f \cdot \mathbb{E}\left[\|\hat{y}_i - y_i\|_F^2\right] + C\frac{1}{d}\left(\frac{\mu}{n}\|u\|_F^2 + 1\right)\right) \\
&\quad + \sum_{k=2}^{K}\frac{1}{d^{2k}}B_\Theta\left(f \cdot \mathbb{E}\left[\|\hat{y}_i - y_i\|_F^2\right] + C\frac{1}{2d^{2k}}\zeta_k\left(\mu\|u\|_F^2 + 1\right)\right) \\
&= \left(\frac{K+1}{2} + 2fB_\Theta + \sum_{k=2}^{K}\frac{f}{d^{2k}}B_\Theta\right)\mathbb{E}\left[\|\hat{y}_i - y_i\|_F^2\right] \\
&\quad + \left(1 + \frac{1}{d}\right)\left(\left(\frac{B_W^2}{2} + CB_\Theta\right)\frac{\mu}{n}\|u\|_F^2 + \frac{B_W^2}{2f} + CB_\Theta\right) \\
&\quad + \sum_{k=2}^{K}\frac{\zeta_k}{d^{2k}}\left(\left(\mu\|u\|_F^2 + \frac{n}{f}\right)\frac{B_W^2}{4} + \left(\mu\|u\|_F^2 + 1\right)\frac{B_\Theta}{d^{2k}}\right)
\end{aligned}
\tag{38}
$$

We express the result in big-$O$ notation:

$$
\mathbb{E}\left[\|\nabla\ell(\hat{y}_i, y_i; \Theta, W)\|_F\right] = O\left(\mathbb{E}\left[\|\hat{y}_i - y_i\|_F^2\right]\right) + O\left(\sum_{k=2}^{K}\zeta_k\right)
$$

where $\zeta_k = \mathbb{E}\left[\left(A_{ij}^k \mid y_i = y_j\right)^2\right] + \mathbb{E}\left[\left(A_{ij}^k \mid y_i \neq y_j\right)^2\right]$

After obtaining the upper bound of the gradient norm, and applying Theorem 6, we derive the uniform transductive stability of spectral GNNs on graphs $G \sim cSBM(n, f, \mu, u, \lambda, d)$ with two classes ($C = 2$) in big-$O$ notation as:

$$\gamma = r\beta; \beta = \frac{1}{\epsilon}\left[O\left(\mathbb{E}\left[\|\hat{y}_i - y_i\|_F^2\right]\right) + O\left(\sum_{k=2}^{K}\left(\mathbb{E}\left[\left(A_{ij}^k \mid y_i = y_j\right)^2\right] + \mathbb{E}\left[\left(A_{ij}^k \mid y_i \neq y_j\right)^2\right]\right)\right)\right]$$

where $r$ is the same as that in Theorem 6.

$\square$

## E ANALYSIS OF PROPERTIES

In this section, we first derive the relationship between the parameter $\lambda$ in cSBM and the edge homophilic ratio of the graph. We then analyze how the expected prediction error, $\mathbb{E}[\|\hat{y}_i - y_i\|_F^2]$, and $\zeta_k$ vary with $\lambda$ and $K$. Finally, we examine the impact of $\lambda$ and $K$ on the uniform transductive stability and generalization performance of spectral GNNs.

### E.1 PROOF OF PROPOSITION 12

**Proposition 12.** *For a graph $G \sim cSBM(n, \mu, u, \lambda, d)$, the expected edge homophily ratio is:*

$$\mathbb{E}[H_{edge}] = \frac{d + \lambda\sqrt{d}}{2d}; \quad \mathbb{E}[H_{edge}] = \frac{c_{in}}{c_{in} + c_{out}}. \tag{4}$$

*Proof.* Graphs generated with cSBM contain two clusters of equal size. Thus, there are $\frac{n}{2}$ nodes in each cluster belonging to the same class. The expected number of edges between nodes of the same class is given by:

$$\mathbb{E}[E_{\text{same}}] = \binom{\frac{n}{2}}{2} \cdot \frac{c_{in}}{n} = \frac{c_{in}(n-2)}{8},$$

where $\binom{\frac{n}{2}}{2}$ represents the number of possible edges between nodes within the same cluster, and $\frac{c_{in}}{n}$ is the probability of an edge existing between two nodes of the same class.

The expected number of edges between nodes of different classes is given by:

$$\mathbb{E}[E_{\text{diff}}] = \frac{n}{2} \cdot \frac{n}{2} \cdot \frac{c_{out}}{n} \cdot \frac{1}{2} = \frac{c_{out}n}{8},$$

where $\frac{n}{2} \cdot \frac{n}{2}$ represents the total number of possible edges between nodes in different clusters, $\frac{c_{out}}{n}$ is the probability of an edge existing between nodes of different classes, and the factor $\frac{1}{2}$ accounts for double-counting edges.

The expected value of $H_{edge}$, the ratio of the expected number of edges between nodes of the same class to the total expected number of edges, is given by:

$$\begin{aligned}
\mathbb{E}[H_{edge}] &= \frac{\mathbb{E}[E_{\text{same}}]}{\mathbb{E}[E_{\text{same}}] + \mathbb{E}[E_{\text{diff}}]} \\
&= \frac{\frac{c_{in}(n-2)}{8}}{\frac{c_{in}(n-2)}{8} + \frac{c_{out}n}{8}} \\
&= \frac{(d + \lambda\sqrt{d})(n-2)}{(d + \lambda\sqrt{d})(n-2) + (d - \lambda\sqrt{d})n} \\
&= \frac{d + \lambda\sqrt{d}}{2d}, \quad \text{as } n \to \infty.
\end{aligned}$$

Here, $d$ represents the average degree, and $\lambda$ measures the level of separation between clusters. As $n \to \infty$, the terms involving $(n-2)$ and $n$ simplify, yielding the final expression for $\mathbb{E}[H_{edge}]$.

We also derive the relationship between the expectation of $H_{edge}$ and the parameters $c_{in}$ and $c_{out}$ as follows:

$$
\begin{aligned}
\mathbb{E}[H_{edge}] &= \frac{\mathbb{E}[E_{\text{same}}]}{\mathbb{E}[E_{\text{same}}] + \mathbb{E}[E_{\text{diff}}]} \\
&= \frac{\frac{c_{in}(n-2)}{8}}{\frac{c_{in}(n-2)}{8} + \frac{c_{out}n}{8}} \\
&= \frac{c_{in}(n-2)}{c_{in}(n-2) + c_{out}n} \\
&= \frac{c_{in}}{c_{in} + c_{out}}, \quad \text{as } n \to \infty.
\end{aligned}
$$

$\square$

### E.2 PROOF OF THEOREM 14

**Theorem 14** ($\mathbb{E}\left[\|\hat{y}_i - y_i\|_F^2\right]$ and $\lambda, K$). *Given a graph $G \sim cSBM(n, \mu, u, \lambda, d)$ and a spectral GNN of order $K$, $\mathbb{E}[\|\hat{y}_i - y_i\|_F^2]$ for any node $v_i$ satisfies the following: it increases with $\lambda \in [-\sqrt{d}, 0]$, decreases with $\lambda \in [0, \sqrt{d}]$, and reaches its maximum at $\lambda = 0$; it increases with $K$ if $\sum_{k=2}^{K} \theta_k \frac{(k-1)!}{2^{k-1}}$ grows more slowly than $\sum_{k=2}^{K} \theta_k^2 \frac{(k-1)!}{2^k}$ as $K$ increases.*

*Proof.* Denote

$$
Z = \sum_{k=0}^{K} \theta_k \tilde{A}^k X W, \qquad \hat{Y} = \text{softmax}(Z).
$$

For any node $v_i$ with true class $y_i$, its prediction is denoted as:

$$
\hat{y}_i = \text{softmax}(Z_{i:}).
$$

In the case of binary classification ($C = 2$), for a node with true class $y_i = [1, 0]$, the predicted class is:

$$
\hat{y}_i = [\hat{y}_1, \hat{y}_2] = \text{softmax}([Z_{i1}, Z_{i2}]) = [\sigma(Z_{i1} - Z_{i2}), 1 - \sigma(Z_{i1} - Z_{i2})],
$$

where $\sigma(x) = \frac{1}{1+e^{-x}}$ is the sigmoid function.

Let $z_i = Z_{i1} - Z_{i2}$, then:
$$
\hat{y}_i = [\sigma(z_i), 1 - \sigma(z_i)].
$$

Thus, the squared Frobenius norm of the difference between $\hat{y}_i$ and $y_i$ is:

$$
\|\hat{y}_i - y_i\|_F^2 = (\sigma(z_i) - 1)^2 + (1 - \sigma(z_i))^2 = 2(1 - \sigma(z_i))^2.
$$

Taking the expectation, we have:

$$
\mathbb{E}[\|\hat{y}_i - y_i\|_F^2] = 2\mathbb{E}[(1 - \sigma(z_i))^2].
$$

As the node feature $x_i \sim \mathcal{N}(y_i \sqrt{\mu/n} u, I_f/f)$, any linear combination of Gaussian variables is still Gaussian. Therefore, we have:
$$
z_i \sim \mathcal{N}(\mu_{z_i}, \omega_{z_i}^2),
$$

where:
$$
\mu_{z_i} = \mathbb{E}[z_i] = \mathbb{E}[Z_{i1} - Z_{i2}] = \mathbb{E}[Z_{i1}] - \mathbb{E}[Z_{i2}].
$$

Given that $c_{in} = d + \lambda\sqrt{d}$, $c_{out} = d - \lambda\sqrt{d}$, and $\lambda \in [-\sqrt{d}, \sqrt{d}]$, we observe:

$$
c_{in}^k - c_{out}^k = O(d^k), \quad c_{in}^k = O(d^k), \quad c_{out}^k = O(d^k). \tag{39}
$$

Assuming $u \sim \mathcal{N}(0, I_f)$, $d \ll f$, and that $\Theta, W$ are bounded (as per Assumption 4), we analyze the dominant terms in $\mu_{z_i}$ and $\omega_{z_i}^2$. From Theorem 32, we derive the expectation of $(\tilde{A}^k X W)_{ij}$. Consequently, we obtain:

$$
\begin{aligned}
\mu_{z_i} = \mathbb{E}[Z_{i1}] - \mathbb{E}[Z_{i2}] &= \theta_0 \sqrt{\frac{\mu}{n}} y_i u (W_{:1} - W_{:2}) \\
&+ \theta_1 \frac{1}{2d} \sqrt{\frac{\mu}{n}} (c_{in} - c_{out}) y_i u (W_{:1} - W_{:2}) \\
&+ \sum_{k=2}^{K} \theta_k \frac{(k-1)!}{d^k \cdot 2^{k-1}} O(c_{in}^k - c_{out}^k) \sqrt{\frac{\mu}{n}} y_i u (W_{:1} - W_{:2}) \\
&= O\left( \sum_{k=2}^{K} \theta_k \frac{(k-1)!}{2^{k-1}} \right) \quad \text{(from Eq. (39))}.
\end{aligned}
\tag{40}
$$

Since $\tilde{A}^k$ and $X$ are independent, and the columns of $X$ are also independent, it follows that $\left( \sum_{k=0}^{K} \theta_k \tilde{A}^k X \right)_{ij}$ and $\left( \sum_{k=0}^{K} \theta_k \tilde{A}^k X \right)_{it}$ are independent. According to Theorem 32, we compute the variance of $(\tilde{A}^k X W)_{ij}$. Then, we have:

$$
\begin{aligned}
\omega_{z_i}^2 &= \text{Var}(Z_{i1} - Z_{i2}) \\
&= \text{Var}\left( \left( \sum_{k=0}^{K} \theta_k \tilde{A}^k X \right)_{i:} (W_{:1} - W_{:2}) \right) \\
&= \text{Var}\left( \sum_{j=1}^{f} \left( \sum_{k=0}^{K} \theta_k \tilde{A}^k X \right)_{ij} (W_{j1} - W_{j2}) \right) \\
&= \sum_{j=1}^{f} (W_{j1} - W_{j2})^2 \sum_{k=0}^{K} \theta_k^2 \text{Var}\left( \left( \tilde{A}^k X \right)_{ij} \right) \quad \text{(independence)} \\
&= \sum_{j=1}^{f} (W_{j1} - W_{j2})^2 \sum_{k=0}^{K} \theta_k^2 \Bigg[ \frac{1}{2 \cdot d^2} \left( d - \frac{c_{in}^2 + c_{out}^2}{n} \right) \cdot \left( \frac{\mu}{n} (u W_{:j})^2 + \frac{\|W_{:j}\|_2^2}{f} \right) \\
&\quad + \frac{(k-1)!}{d^{2k} \cdot 2^k} \Bigg( \sum_{a=2}^{k+1} O\left( \sum_{s=\min(2,2(a-2),2(k+1-a))}^{\min(2(a-1),2(k+1-a))} c_{in}^{k-s} \cdot c_{out}^s \right) \\
&\quad + \sum_{a=1}^{k} O\left( \sum_{s=1}^{\min(2a-1,2(k-a)+1)} c_{in}^{k-s} \cdot c_{out}^s \right) \Bigg) \cdot \left( \frac{\mu}{n} (u W_{:j})^2 + \frac{\|W_{:j}\|_2^2}{f} \right) \Bigg] \\
&= O\left( \sum_{k=2}^{K} \theta_k^2 \frac{(k-1)!}{2^k} \right) \quad \text{(from Eq. (39))}.
\end{aligned}
\tag{41}
$$

(1) $\mathbb{E}[\|\hat{y}_i - y_i\|_F^2]$ **and** $\lambda$: According to Lemma 29 and Lemma 30, we know that:

- $\mu_{z_i}$ monotonically decreases, and $\omega_{z_i}^2$ monotonically increases on $\lambda \in [-\sqrt{d}, 0]$;
- $\mu_{z_i}$ monotonically increases, and $\omega_{z_i}^2$ monotonically decreases on $\lambda \in [0, \sqrt{d}]$;
- $\mu_{z_i}$ achieves its minimum value, and $\omega_{z_i}^2$ achieves its maximum value when $\lambda = 0$.

The expectation of $(1 - \sigma(z_i))^2$ is given by:

$$
\mathbb{E}[(1 - \sigma(z_i))^2] = \int_{-\infty}^{\infty} (1 - \sigma(z_i))^2 \cdot \frac{1}{\sqrt{2\pi}\omega_{z_i}} e^{-\frac{(z - \mu_{z_i})^2}{2\omega_{z_i}^2}} \, dz_i.
\tag{42}
$$

Since the integral decreases with $\mu_{z_i}$ and increases with $\omega_{z_i}^2$, we conclude:

- $\mathbb{E}[(1 - \sigma(z_i))^2]$ increases on $\lambda \in [-\sqrt{d}, 0]$;
- $\mathbb{E}[(1 - \sigma(z_i))^2]$ decreases on $\lambda \in [0, \sqrt{d}]$;
- $\mathbb{E}[(1 - \sigma(z_i))^2]$ achieves its maximum value when $\lambda = 0$.

Since $\mathbb{E}[\|\hat{y}_i - y_i\|_F^2]$ has the same trend as $\mathbb{E}[(1 - \sigma(z_i))^2]$, we observe the same behavior for $\mathbb{E}[\|\hat{y}_i - y_i\|_F^2]$.

(2) $\mathbb{E}[\|\hat{y}_i - y_i\|_F^2]$ **and** $K$: We rewrite $z$ as:

$$z = \mu_{z_i} + \omega_{z_i} y,$$

where $y \sim \mathcal{N}(0, 1)$. Substituting into Eq. (42), we have:

$$\mathbb{E}[(1 - \sigma(z_i))^2] = \int_{-\infty}^{\infty} (1 - \sigma(\mu_{z_i} + \omega_{z_i} y))^2 \frac{1}{\sqrt{2\pi}} e^{-\frac{y^2}{2}} \, dy.$$

(a) If $\mu_{z_i}$ increases faster than $\omega_{z_i}^2$ as $K$ increases: In this case, $z$ is dominated by $\mu_{z_i}$, and we have:

$$\begin{aligned}
\mathbb{E}[(1 - \sigma(z))^2] &= \int_{-\infty}^{\infty} (1 - \sigma(\mu_{z_i}))^2 \frac{1}{\sqrt{2\pi}} e^{-\frac{y^2}{2}} \, dy \\
&= (1 - \sigma(\mu_{z_i}))^2 \\
&\leq 0.25.
\end{aligned}$$

(b) If $\mu_{z_i}$ increases slower than $\omega_{z_i}^2$ as $K$ increases: In this case, $z$ is dominated by $\omega_{z_i} y$, and we have:

$$\begin{aligned}
\mathbb{E}[(1 - \sigma(z))^2] &= \int_{-\infty}^{\infty} (1 - \sigma(\omega_{z_i} y))^2 \frac{1}{\sqrt{2\pi}} e^{-\frac{y^2}{2}} \, dy \\
&= \int_{-\infty}^{0} (1 - 0) \cdot \frac{1}{\sqrt{2\pi}} e^{-\frac{y^2}{2}} \, dy + \int_{0}^{\infty} (1 - 1)^2 \cdot \frac{1}{\sqrt{2\pi}} e^{-\frac{y^2}{2}} \, dy \\
&= 0.5.
\end{aligned}$$

From this analysis, we conclude:

- If $\mu_{z_i}$ increases slower than $\omega_{z_i}^2$ as $K$ increases, $\mathbb{E}[(1 - \sigma(z))^2]$ approaches $0.5$.
- If $\mu_{z_i}$ increases faster than $\omega_{z_i}^2$ as $K$ increases, $\mathbb{E}[(1 - \sigma(z))^2]$ is at most $0.25$.

Briefly, when $\mu_{z_i}$ increases slower than $\omega_{z_i}^2$ as $K$ increases, $\mathbb{E}[\|\hat{y}_i - y_i\|_F^2]$ increases with $K$. From Eq. (40) and Eq. (41), we observe that the dominant term of $\mu_{z_i}$ is $\sum_{k=2}^{K} \theta_k \frac{(k-1)!}{2^{k-1}}$, while the dominant term of $\omega_{z_i}^2$ is $\sum_{k=2}^{K} \theta_k^2 \frac{(k-1)!}{2^k}$. Therefore, $\mathbb{E}[\|\hat{y}_i - y_i\|_F^2]$ increases with $K$ if $\sum_{k=2}^{K} \theta_k \frac{(k-1)!}{2^{k-1}}$ grows slower than $\sum_{k=2}^{K} \theta_k^2 \frac{(k-1)!}{2^k}$.

$\square$

### E.3 PROOF OF THEOREM 15

**Theorem 15** ($\zeta_k$ and $\lambda, K$). *Given a graph $G \sim cSBM(n, \mu, u, \lambda, d)$ and a spectral GNN of order $K$, $\zeta_k$ has the following properties: (1) it increases with $\lambda \in [-\sqrt{d}, 0]$, decreases with $\lambda \in [0, \sqrt{d}]$, and achieves its maximum value at $\lambda = 0$; (2) it increases with $k$ as $k$ grows, for $k \in [0, K]$.*

*Proof.* As

$$\begin{aligned}
\zeta_k &= \mathbb{E}[(A_{ij}^k | y_i = y_j)^2] + \mathbb{E}[(A_{ij}^k | y_i \neq y_j)^2] \\
&= \left(\mathbb{E}[A_{ij}^k | y_i = y_j]\right)^2 + \mathbb{V}\left[A_{ij}^k | y_i = y_j\right] + \left(\mathbb{E}[A_{ij}^k | y_i \neq y_j]\right)^2 + \mathbb{V}\left[A_{ij}^k | y_i \neq y_j\right].
\end{aligned} \tag{43}$$

According to Theorem 31, we have explicit forms of $\mathbb{E}[A_{ij}^k]$ and $\mathrm{Var}(A_{ij}^k)$ for the cases $y_i = y_j$ and $y_i \neq y_j$. Substituting these into Eq. (43), we get:

$$\zeta_k = \rho_=^2 + \rho_= + \rho_{\neq}^2 + \rho_{\neq}$$

$$= \left( \frac{(k-1)!}{n \cdot 2^{k-1}} \sum_{a=2}^{k+1} O \left( \sum_{s=\min(2,2(a-2),2(k+1-a))}^{\min(2(a-1),2(k+1-a))} c_{in}^{k-s} \cdot c_{out}^s \right) \right)^2$$

$$+ \frac{(k-1)!}{n \cdot 2^{k-1}} \sum_{a=2}^{k+1} O \left( \sum_{s=\min(2,2(a-2),2(k+1-a))}^{\min(2(a-1),2(k+1-a))} c_{in}^{k-s} \cdot c_{out}^s \right)$$

$$+ \left( \frac{(k-1)!}{n \cdot 2^{k-1}} \sum_{a=1}^{k} O \left( \sum_{s=1}^{\min(2a-1,2(k-a)+1)} c_{in}^{k-s} \cdot c_{out}^s \right) \right)^2$$

$$+ \frac{(k-1)!}{n \cdot 2^{k-1}} \sum_{a=1}^{k} O \left( \sum_{s=1}^{\min(2a-1,2(k-a)+1)} c_{in}^{k-s} \cdot c_{out}^s \right).$$

Given $c_{in} = d + \lambda\sqrt{d}$ and $c_{out} = d - \lambda\sqrt{d}$, all terms $\rho_=^2 + \rho_= + \rho_{\neq}^2 + \rho_{\neq}$ in $\zeta_k$ are in the form:

$$g(\lambda) = \sum_{s=1}^{k} (d + \lambda\sqrt{d})^{k-s} \cdot (d - \lambda\sqrt{d})^s.$$

According to Lemma 30, functions in this form $g(\lambda)$ strictly increase on $\lambda \in [-\sqrt{d}, 0]$ and strictly decrease on $\lambda \in [0, \sqrt{d}]$. Therefore, $\zeta_k$ strictly increases on $\lambda \in [-\sqrt{d}, 0]$ and strictly decreases on $\lambda \in [0, \sqrt{d}]$. When $k$ increases, $\zeta_k$ contains more terms, causing it to increase with $k$ in the order of $K$. $\qquad\square$

### E.4 PROOF OF PROPOSITION 16

**Proposition 16.** *For a fixed $K$, $\gamma$-uniform transductive stability and generalization error bound strictly increase as $\lambda$ moves from $-\sqrt{d}$ to 0, and decreases as $\lambda$ moves from 0 to $\sqrt{d}$. For a fixed $\lambda$, if $\sum_{k=2}^{K} \theta_k \frac{(k-1)!}{2^{k-1}}$ grows more slowly than $\sum_{k=2}^{K} \theta_k^2 \frac{(k-1)!}{2^k}$ as $K$ increases, then $\gamma$-uniform transductive stability and generalization error bound increase with $K$.*

*Proof.* According to Theorem 6 and Theorem 13, the uniform stability of spectral GNNs depends on the upper bound of the gradient norm $\beta$, and

$$\beta = \left( \frac{K+1}{2} + 2fB_\Theta + \sum_{k=2}^{K} \frac{f}{d^{2k}} B_\Theta \right) \mathbb{E} \left[ \|\hat{y}_i - y_i\|_F^2 \right]$$

$$+ \left( 1 + \frac{1}{d} \right) \left( \left( \frac{B_W^2}{2} + CB_\Theta \right) \frac{\mu}{n} \|u\|_F^2 + \frac{B_W^2}{2f} + CB_\Theta \right)$$

$$+ \sum_{k=2}^{K} \frac{\zeta_k}{d^{2k}} \left( \left( \mu\|u\|_F^2 + \frac{n}{f} \right) \frac{B_W^2}{4} + \left( \mu\|u\|_F^2 + 1 \right) \frac{B_\Theta}{d^{2k}} \right)$$

where $\zeta_k = \rho_=^2 + \rho_= + \rho_{\neq}^2 + \rho_{\neq}$, and $\rho_=$ and $\rho_{\neq}$ are the parameters of distribution in Theorem 31.

Denote

$$\psi_y = \left( \frac{K+1}{2} + 2fB_\Theta + \sum_{k=2}^{K} \frac{f}{d^{2k}} B_\Theta \right);$$

$$\psi_1 = \sum_{k=2}^{K} \frac{\zeta_k}{d^{2k}} \left( \left( \mu\|u\|_F^2 + \frac{n}{f} \right) \frac{B_W^2}{4} + \left( \mu\|u\|_F^2 + 1 \right) \frac{B_\Theta}{d^{2k}} \right).$$

We show that the terms $\mathbb{E} \left[ \|\hat{y}_i - y_i\|_F^2 \right], \psi_y$, and $\psi_1$ can all be affected by $\lambda, K$.

(1) **Term** $\mathbb{E}\left[\|\hat{y}_i - y_i\|_F^2\right]$

According to Theorem 14, the expected prediction error $\mathbb{E}\left[\|\hat{y}_i - y_i\|_F^2\right]$ strictly increases with $\lambda \in [-\sqrt{d}, 0]$ and decreases with $\lambda \in [0, \sqrt{d}]$. In addition, it increases with $K$ when $\sum_{k=2}^{K} \theta_k \frac{(k-1)!}{2^{k-1}}$ grows slower than $\sum_{k=2}^{K} \theta_k^2 \frac{(k-1)!}{2^k}$.

(2) **Term** $\psi_y$

As $\psi_y = \left(\frac{K+1}{2} + \sum_{k=0}^{K} |\theta_k| f\right)$ which does not contain $\lambda$, the class distribution has no effect on $\psi_y$. It also increases with order $K$.

(3) **Terms** $\psi_1$

According to Theorem 15, $\zeta_k$ strictly increases on $\lambda \in [-\sqrt{d}, 0]$, decreases on $\lambda \in [0, \sqrt{d}]$ and it increases with order $K$.

Since all the other elements in $\psi_1$ except $\zeta_k$ are positive, $\psi_1$ and $\zeta_k$ has same trend when $\lambda$ and $K$ changes.

According to Proposition 12, we have

$$\lambda \in [0, \sqrt{d}] \Leftrightarrow H_{edge} \in [0.5, 1] \text{ and } \lambda \in [-\sqrt{d}, 0] \Leftrightarrow H_{edge} \in [0, 0.5].$$

According to Theorem 9, any factors affecting $\gamma$ affect the generalization error bound. Thus, we conclude the following cases:

(a) uniform transductive stability $\gamma$, generalization error bound and $\lambda$

From the above analysis, we know that $\phi_y$ is not affected by $\lambda$, and terms $\mathbb{E}\left[\|\hat{y}_i - y_i\|_F^2\right]$, $\psi_1$ strictly increase on $\lambda \in [-\sqrt{d}, 0]$ and decrease on $\lambda \in [0, \sqrt{d}]$. According to Theorem 6 and Theorem 9, this shows that the stability decreases and the generalization error bound increases when $H_{edge} \in (0, 0.5]$. The stability increases and the generalization error bound decreases when $H_{edge} \in [0, 5, 1)$. Spectral GNNs are stable and generalize well on strong homophilic and heterophilic graphs.

(b) uniform transductive stability $\gamma$, generalization error bound, and $K$

From the above analysis, we know that terms $\phi_y, \psi_1$ increase with $K$. According to Theorem 14, when the condition $\sum_{k=2}^{K} \theta_k \frac{(k-1)!}{2^{k-1}}$ grows slower than $\sum_{k=2}^{K} \theta_k^2 \frac{(k-1)!}{2^k}$ is satisfied, the expected prediction error $\mathbb{E}\left[\|\hat{y}_i - y_i\|_F^2\right]$ increases with $K$.

Therefore, when above condition is satisfied, the gradient norm bound $\beta$ increase with $K$. According to Theorem 6 and Theorem 9, this indicates that the uniform transductive stability $\gamma$ and generalization error bound also increases with $K$.

$\square$

# F    DETAILS OF EXPERIMENTS

## F.1    DATASETS

The statistical properties of real-world datasets, including the number of nodes, edges, feature dimensions, node classes, and edge homophily ratios, are summarized in Table 2 and Table 3. We use the directed and cleaned versions of the Chameleon and Squirrel datasets provided by (Platonov et al., 2023), where repeated nodes have been removed.

## F.2    SPECTRAL GNNS

In the literature, there are generally two kinds of architectures for spectral GNNs:

- Early spectral GNNs architecture: It is given by $Y = X_L$, $X_l = \alpha\left(\sum_{k=1}^{K} M^k X_{l-1} H_{lk}\right)$, where $M$ is a graph matrix, $X_l$ is the feature at the $l$-th layer, $H_{lk} \in \mathbb{R}^{f_l \times f_{l-1}}$, $f_l$ is

| Statistics | Texas | Wisconsin | Cornell | Actor | Chameleon | Squirrel | Citeseer | Pubmed | Cora |
|---|---|---|---|---|---|---|---|---|---|
| # Nodes | 183 | 251 | 183 | 7,600 | 890 | 2,223 | 3,327 | 19,717 | 2,708 |
| # Edges | 295 | 466 | 295 | 26,752 | 27,168 | 131,436 | 4,676 | 44,327 | 5,278 |
| # Features | 1,703 | 1,703 | 1,703 | 932 | 2,325 | 2,089 | 3,703 | 500 | 1,433 |
| # Classes | 5 | 5 | 5 | 5 | 5 | 6 | 5 | 7 | |
| Edge Homophily | 0.11 | 0.21 | 0.22 | 0.24 | 0.22 | 0.74 | 0.8 | 0.81 | |

Table 2: Statistics of real-world datasets.

| Statistics | OGBN-Arxiv | OGBN-Products |
|---|---|---|
| # Nodes | 169,343 | 2,449,029 |
| # Edges | 2,315,598 | 61,859,140 |
| # Features | 128 | 100 |
| # Classes | 40 | 47 |
| Edge Homophily | 0.65 | 0.81 |

Table 3: Statistics of OGBN datasets.

the feature dimension of the $l$-th layer, and $\alpha$ is an activation function. This describes the architecture of earlier spectral GNNs, such as GCN ($M^k = D^{-1/2}(I + A)D^{-1/2}$) and ChebNet (where $M^k$ represents the Chebyshev polynomial basis expanded on the normalized graph Laplacian matrix).

- Modern spectral GNNs architecture: Recent advances in spectral GNNs do not adhere to this multi-layer architecture. Instead, state-of-the-art spectral GNNs employ a single-layer structure as described in Eq. (1) of our paper:

$$\Psi(M, X) = \sigma(g_\Theta(M)f_W(X)),$$

where $M \in \mathbb{R}^{n \times n}$ is a graph matrix (e.g., Laplacian or adjacency matrix), $g_\Theta(M) = \sum_{k=0}^{K} \theta_k T_k(M)$ performs graph convolution using the $k$-th polynomial basis $T_k(\cdot)$ and learnable parameters $\Theta = \{\theta_k\}_{k=0}^{K}$, $f_W(X)$ is a feature transformation parameterized by $W$, and $\sigma$ is a non-linear activation function (e.g., softmax). Recent spectral GNNs, such as GPRGNN, JacobiConv, BernNet, ChebBase, and ChebNetII, adopt this architecture (Chien et al., 2021; Wang & Zhang, 2022; He et al., 2021; 2022b), and it serves as the basis for theoretical analysis of spectral GNNs (Wang & Zhang, 2022; Balcilar et al., 2021).

We study spectral GNNs with modern architecture. We detail the spectral GNNs used in our experiments below. For a graph with adjacency matrix $A$, degree matrix $D$, and identity matrix $I$, we define the following matrices: the normalized Laplacian matrix $\hat{L} = I - D^{-1/2}AD^{-1/2}$, the shifted normalized Laplacian matrix $\tilde{L} = -D^{-1/2}AD^{-1/2}$, the normalized adjacency matrix $\tilde{A} = D^{-1/2}AD^{-1/2}$, and the normalized adjacency matrix with self-loops $\tilde{A}' = (D + I)^{-1/2}(A + I)(D + I)^{-1/2}$.

**ChebNet** (Defferrard et al., 2016): This model uses the Chebyshev basis to approximate a spectral filter:

$$\hat{Y} = \sum_{k=0}^{K} \theta_k T_k(\tilde{L})f_W(X)$$

where $X$ is the raw feature matrix, $\Theta = [\theta_0, \theta_1, \ldots, \theta_K]$ is the graph convolution parameter, $W$ is the feature transformation parameter and $f_W(X)$ is usually a 2-layer MLP. $T_k(\tilde{L})$ is the $k$-th Chebyshev basis expanded on the shifted normalized graph Laplacian matrix $\tilde{L}$ and is recursively calculated:

$$T_0(\tilde{L}) = I$$
$$T_1(\tilde{L}) = \tilde{L}$$
$$T_k(\tilde{L}) = 2\tilde{L}T_{k-1}(\tilde{L}) - T_{k-2}(\tilde{L})$$

**ChebNetII** (He et al., 2022a): The model is formulated as

$$\hat{Y} = \frac{2}{K+2} \sum_{k=0}^{K} \sum_{j=0}^{K} \theta_j T_k(x_j) T_k(\tilde{L}) f_W(X),$$

where $X$ is the input feature matrix, $W$ is the feature transformation parameter, $f_W(X)$ is usually a 2-layer MLP, $T_k(\cdot)$ is the $k$-th Chebyshev basis expanded on $\cdot$, $x_j = \cos\left((j + 1/2)\pi/(K+1)\right)$ is the $j$-th Chebyshev node, which is the root of the Chebyshev polynomials of the first kind with degree $K+1$, and $\theta_j$ is a learnable parameter. Graph convolution parameter in ChebNet is reparameterized with Chebyshev nodes and learnable parameters $\theta_j$.

**JacobiConv** (Wang & Zhang, 2022): This model uses the Jacobi basis to approximate a filter as:

$$\hat{Y} = \sum_{k=0}^{K} \theta_k T_k^{a,b}(\tilde{A}) f_W(X),$$

where $X$ is the input feature matrix, $\Theta = [\theta_0, \theta_1, \ldots, \theta_K]$ is the graph convolution parameter, $W$ is the feature transformation parameter and $f_W(X)$ is usually a 2-layer MLP. $T_k^{a,b}(\tilde{A})$ is the Jacobi basis on normalized graph adjacency matrix $\tilde{A}$ and is recursively calculated as

$$T_k^{a,b}(\tilde{A}) = I$$
$$T_k^{a,b}(\tilde{A}) = \frac{1-b}{2}I + \frac{a+b+2}{2}\tilde{A}$$
$$T_k^{a,b}(\tilde{A}) = \gamma_k \tilde{A} T_{k-1}^{a,b}(\tilde{A}) + \gamma_k' T_{k-1}^{a,b}(\tilde{A}) + \gamma_k'' T_{k-2}^{a,b}(\tilde{A})$$

where $\gamma_k = \frac{(2k+a+b)(2k+a+b-1)}{2k(k+a+b)}, \gamma_k' = \frac{(2k+a+b-1)(a^2-b^2)}{2k(k+a+b)(2k+a+b-2)}, \gamma_k'' = \frac{(k+1-1)(k+b-1)(2k+a+b)}{k(k+a+b)(2k+a+b-2)}$. $a$ and $b$ are hyper-parameters. Usually, grid search is used to find the optimal $a$ and $b$ values.

**GPRGNN** (Chien et al., 2021): This model uses the monomial basis to approximate a filter:

$$\hat{Y} = \sum_{k=0}^{K} \theta_k \tilde{A}'^k f_W(X)$$

where $X$ is the input feature matrix, $\Theta = [\theta_0, \theta_1, \ldots, \theta_K]$ is the graph convolution parameter, $W$ is the feature transformation parameter and $f_W(X)$ is usually a 2-layer MLP. $\tilde{A}'$ is the normalized adjacency matrix with self-loops.

**BernNet** (He et al., 2021): This model uses the Bernstein basis for approximation:

$$\hat{Y} = \sum_{k=0}^{K} \theta_k \frac{1}{2^K} \binom{K}{k} (2I - \hat{L})^{K-k} \hat{L}^k f_W(X)$$

where $X$ is the input feature matrix, $\Theta = [\theta_0, \theta_1, \ldots, \theta_K]$ is the graph convolution parameter, $W$ is the feature transformation parameter and $f_W(X)$ is usually a 2-layer MLP. $\hat{L}$ is the normalized Laplacian matrix.

### F.3 HYPER-PARAMETER SETTINGS

All experiments were conducted on an NVIDIA RTX A6000 GPU with 48GB of memory.

We employ a two-layer Multi-Layer Perceptron (MLP) with a hidden layer size of 64 for the feature transformation function $f_W$, using ReLU as the activation function across all spectral GNN models.

Following (Tang & Liu, 2023a; Cong et al., 2021), the dropout rate and weight decay are set to 0.0. The Adam optimizer is used for optimization. Each experiment runs for a maximum of 300 iterations and is repeated 10 times to report the mean and variance of the results. A grid search is conducted to determine the best learning rate from $\{0.05, 0.01, 0.001\}$.

### F.4 DETAILED EXPERIMENTAL RESULTS

| $H_{edge}$ | 0.1 | 0.2 | 0.3 | 0.4 | 0.5 | 0.6 | 0.7 | 0.8 | 0.9 |
|---|---|---|---|---|---|---|---|---|---|
| ChebNet | 94.92±0.24 | 86.08±0.43 | 81.09±0.63 | 75.11±0.73 | 72.69±0.66 | 74.66±0.65 | 79.62±0.78 | 86.03±0.6 | 94.64±0.39 |
| Acc Gap | 5.08±0.24 | 13.92±0.41 | 18.91±0.57 | 24.89±0.72 | 27.3±0.62 | 25.34±0.68 | 20.38±0.74 | 13.97±0.61 | 5.36±0.41 |
| Loss Gap | 0.64±0.07 | 3.15±0.14 | 3.72±0.2 | 5.42±0.24 | 5.88±0.5 | 6.01±0.27 | 4.62±0.3 | 3.04±0.18 | 0.98±0.06 |
| ChebNetII | 92.19±0.51 | 85.03±0.58 | 79.83±0.43 | 77.55±0.64 | 77.34±0.54 | 77.7±0.57 | 78.22±0.73 | 83.68±0.41 | 91.43±0.48 |
| Acc Gap | 7.81±0.47 | 14.97±0.58 | 20.17±0.41 | 22.45±0.66 | 22.66±0.49 | 22.3±0.57 | 21.77±0.71 | 16.32±0.44 | 8.57±0.47 |
| Loss Gap | 0.66±0.07 | 1.84±0.11 | 3.55±0.21 | 4.77±0.26 | 4.86±0.13 | 4.64±0.21 | 4.23±0.33 | 2.14±0.17 | 0.72±0.05 |
| JacobiConv | 89.25±3.35 | 77.23±4.51 | 77.19±0.66 | 77.0±0.55 | 79.06±0.61 | 80.2±0.57 | 84.64±0.39 | 90.48±0.24 | 96.91±0.24 |
| Acc Gap | 10.71±2.86 | 22.73±4.36 | 22.8±0.67 | 23.0±0.54 | 20.94±0.61 | 19.8±0.6 | 15.36±0.41 | 9.51±0.24 | 3.09±0.25 |
| Loss Gap | 0.69±0.26 | 1.58±0.45 | 4.08±0.21 | 4.33±0.14 | 5.36±0.33 | 1.95±0.13 | 1.58±0.13 | 0.99±0.06 | 0.16±0.01 |
| GPRGNN | 90.33±0.57 | 87.06±0.64 | 81.71±0.41 | 77.03±0.47 | 77.23±0.65 | 79.52±0.59 | 82.72±0.52 | 89.25±0.5 | 96.45±0.18 |
| Acc Gap | 9.66±0.54 | 12.94±0.67 | 18.29±0.42 | 22.96±0.49 | 22.77±0.64 | 20.48±0.6 | 17.27±0.52 | 10.75±0.54 | 3.55±0.2 |
| Loss Gap | 1.42±0.08 | 2.21±0.14 | 3.27±0.2 | 4.72±0.19 | 5.17±0.13 | 4.7±0.25 | 3.7±0.47 | 2.4±0.32 | 1.05±0.11 |
| BernNet | 87.44±0.5 | 82.92±0.67 | 79.3±0.44 | 77.69±0.53 | 77.97±0.54 | 77.49±0.72 | 76.58±0.79 | 79.73±1.3 | 85.68±1.05 |
| Acc Gap | 12.55±0.5 | 17.08±0.76 | 20.7±0.44 | 22.31±0.54 | 22.03±0.55 | 22.51±0.64 | 23.41±0.8 | 20.27±1.39 | 14.32±1.06 |
| Loss Gap | 1.2±0.06 | 2.45±0.21 | 3.69±0.16 | 4.77±0.24 | 4.72±0.15 | 4.7±0.17 | 4.35±0.35 | 2.92±0.31 | 1.36±0.14 |

Table 4: Testing accuracy, accuracy gap, loss gap of spectral GNNs on synthetic datasets with edge homophilic ratio $H_{edge} \in [0.1, 0.9]$. Small accuracy and loss gaps imply good generalization capability.

| Datasets | Texas | Wisconsin | Actor | Squirrel | Chameleon | Cornell | Citeseer | Pubmed | Cora |
|---|---|---|---|---|---|---|---|---|---|
| ChebNet | 40.82±7.25 | 52.23±3.77 | 26.63±0.53 | 30.08±1.14 | 33.94±1.58 | 44.88±6.19 | 64.16±0.82 | 84.74±0.37 | 74.95±0.96 |
| Acc Gap | 59.18±6.94 | 47.77±3.92 | 73.26±0.54 | 69.92±1.28 | 66.06±1.52 | 55.12±5.95 | 35.82±0.75 | 15.25±0.37 | 25.05±0.92 |
| Loss Gap | 5.91±0.66 | 5.77±0.87 | 21.64±0.8 | 35.68±2.33 | 36.17±3.04 | 6.57±0.82 | 4.68±0.22 | 1.44±0.06 | 3.9±0.29 |
| ChebNetII | 77.55±5.71 | 74.38±3.08 | 27.94±0.36 | 28.1±1.82 | 38.45±1.63 | 73.69±5.12 | 65.85±0.52 | 84.7±0.3 | 74.0±0.8 |
| Acc Gap | 22.45±5.2 | 25.62±3.31 | 71.94±0.33 | 71.83±1.77 | 61.47±1.53 | 26.31±5.0 | 34.12±0.48 | 15.16±0.28 | 26.0±0.75 |
| Loss Gap | 1.1±0.27 | 1.39±0.32 | 20.16±0.76 | 27.56±2.88 | 19.33±1.68 | 1.7±0.3 | 2.66±0.09 | 1.13±0.09 | 2.14±0.09 |
| JacobiConv | 78.06±5.31 | 77.62±2.92 | 27.89±0.63 | 26.78±1.28 | 32.2±2.08 | 80.41±3.98 | 73.56±0.64 | 86.33±0.47 | 84.31±0.49 |
| Acc Gap | 21.94±5.41 | 22.38±2.85 | 71.97±0.66 | 50.85±11.88 | 63.82±9.46 | 19.59±4.18 | 26.41±0.65 | 10.87±1.45 | 15.69±0.5 |
| Loss Gap | 0.94±0.26 | 1.19±0.22 | 31.67±0.86 | 32.75±11.57 | 38.77±7.16 | 0.91±0.16 | 2.16±0.06 | 0.51±0.14 | 1.28±0.09 |
| GPRGNN | 46.84±6.22 | 72.08±3.23 | 26.29±0.65 | 29.91±1.19 | 34.28±1.58 | 61.33±6.12 | 72.89±0.62 | 85.42±0.4 | 84.37±0.51 |
| Acc Gap | 53.16±6.12 | 27.92±2.92 | 71.52±4.82 | 70.09±1.09 | 65.72±1.69 | 38.67±6.43 | 27.08±0.67 | 14.58±0.37 | 15.63±0.54 |
| Loss Gap | 3.35±0.83 | 1.6±0.31 | 29.22±2.69 | 35.34±5.58 | 29.88±2.22 | 2.2±0.53 | 3.32±0.16 | 1.24±0.09 | 1.54±0.1 |
| BernNet | 75.92±5.31 | 81.85±2.23 | 27.28±0.76 | 33.42±1.14 | 33.72±1.38 | 81.43±3.46 | 67.17±0.59 | 84.82±0.25 | 73.39±0.87 |
| Acc Gap | 24.08±5.41 | 18.15±2.16 | 72.61±0.71 | 66.58±1.11 | 66.28±1.33 | 18.57±3.57 | 32.8±0.57 | 14.95±0.45 | 26.61±0.87 |
| Loss Gap | 1.24±0.31 | 0.87±0.26 | 24.68±0.71 | 28.17±1.47 | 27.83±1.75 | 1.06±0.18 | 2.66±0.09 | 1.13±0.13 | 2.18±0.08 |

Table 5: Testing accuracy, accuracy gap, loss gap of spectral GNNs on real world datasets with edge homophilic ratio $H_{edge} \in [0.11, 0.81]$. Small accuracy and loss gaps imply good generalization capability.

| Order K | 1 | 2 | 3 | 4 | 5 | 6 | 7 | 8 | 9 | 10 |
|---|---|---|---|---|---|---|---|---|---|---|
| ChebNet | 87.31±0.3 | 89.11±0.31 | 88.48±0.49 | 84.19±0.9 | 71.3±3.0 | 79.58±0.52 | 80.77±0.62 | 76.21±0.51 | 82.94±0.48 | 86.08±0.41 |
| Acc Gap | 12.7±0.32 | 10.89±0.31 | 11.52±0.5 | 15.8±0.92 | 28.7±3.54 | 20.42±0.51 | 19.23±0.57 | 23.79±0.47 | 17.06±0.45 | 13.92±0.42 |
| Loss Gap | 2.2±0.09 | 1.76±0.07 | 1.9±0.14 | 2.84±0.27 | 7.2±1.45 | 3.88±0.2 | 3.08±0.21 | 3.79±0.26 | 3.8±0.11 | 3.15±0.14 |
| ChebNetII | 85.92±0.56 | 80.1±0.99 | 82.65±0.7 | 85.56±0.45 | 84.64±0.8 | 84.62±0.59 | 85.27±0.51 | 86.2±0.64 | 86.39±0.5 | 85.03±0.57 |
| Acc Gap | 14.07±0.53 | 19.9±1.02 | 17.35±0.73 | 14.44±0.45 | 15.36±0.87 | 15.38±0.6 | 14.73±0.5 | 13.79±0.6 | 13.61±0.49 | 14.97±0.58 |
| Loss Gap | 1.94±0.08 | 3.23±0.31 | 2.62±0.14 | 2.06±0.14 | 1.94±0.21 | 1.95±0.17 | 1.99±0.15 | 1.75±0.14 | 1.83±0.11 | 1.84±0.11 |
| JacobiConv | 77.44±0.67 | 80.51±0.48 | 49.44±1.12 | 39.85±1.91 | 48.81±2.65 | 47.73±7.63 | 60.29±7.48 | 67.53±7.95 | 68.03±9.15 | 77.23±4.79 |
| Acc Gap | 22.55±0.62 | 19.49±0.46 | 50.56±1.18 | 60.13±1.98 | 51.19±2.63 | 52.25±7.08 | 39.7±7.32 | 32.45±7.76 | 31.96±9.19 | 22.73±4.82 |
| Loss Gap | 5.72±0.19 | 5.8±0.26 | 8.81±0.79 | 12.63±1.22 | 7.3±1.01 | 8.23±1.77 | 4.98±1.23 | 3.42±1.39 | 3.33±1.32 | 1.58±0.48 |
| GPRGNN | 83.61±0.66 | 86.14±0.29 | 79.44±1.05 | 88.36±0.28 | 87.25±0.5 | 88.0±0.39 | 87.57±0.47 | 87.5±0.3 | 87.17±0.3 | 87.06±0.59 |
| Acc Gap | 16.39±0.69 | 13.86±0.29 | 20.56±1.06 | 11.63±0.29 | 12.76±0.49 | 12.01±0.32 | 12.43±0.48 | 12.49±0.33 | 12.84±0.29 | 12.94±0.68 |
| Loss Gap | 2.37±0.11 | 2.21±0.1 | 3.18±0.19 | 1.83±0.1 | 2.14±0.2 | 1.93±0.09 | 2.06±0.13 | 2.12±0.09 | 2.19±0.13 | 2.21±0.14 |
| BernNet | 82.76±0.72 | 81.14±0.41 | 81.21±0.57 | 81.47±0.6 | 81.77±0.66 | 82.11±0.75 | 82.32±0.88 | 82.55±0.84 | 82.8±0.81 | 82.92±0.79 |
| Acc Gap | 17.24±0.71 | 18.86±0.39 | 18.79±0.56 | 18.53±0.7 | 18.23±0.62 | 17.89±0.85 | 17.68±0.84 | 17.45±0.79 | 17.2±0.79 | 17.08±0.7 |
| Loss Gap | 2.45±0.17 | 3.02±0.11 | 2.95±0.21 | 2.84±0.2 | 2.75±0.21 | 2.65±0.21 | 2.59±0.22 | 2.54±0.2 | 2.49±0.21 | 2.45±0.21 |

Table 6: Testing accuracy, accuracy gap, loss gap of spectral GNNs on synthetic dataset of edge homophilic ratio $H_{edge} = 0.2$ when $K \in [1, 10]$. Small accuracy and loss gaps imply good generalization capability.

| Order K | 1 | 2 | 3 | 4 | 5 | 6 | 7 | 8 | 9 | 10 |
|---|---|---|---|---|---|---|---|---|---|---|
| ChebNet | 83.78±2.45 | 80.61±4.59 | 80.51±3.47 | 61.73±5.0 | 63.37±8.57 | 36.33±5.72 | 44.18±5.0 | 24.39±2.14 | 30.2±4.8 | 40.82±7.35 |
| Acc Gap | 16.22±2.45 | 19.39±4.8 | 19.49±3.78 | 38.27±5.0 | 36.63±7.86 | 63.67±6.12 | 55.82±5.0 | 75.61±2.24 | 69.8±5.0 | 59.18±7.15 |
| Loss Gap | 1.49±0.44 | 1.26±0.44 | 1.48±0.31 | 2.77±0.53 | 3.08±0.59 | 8.98±0.68 | 6.09±0.72 | 7.99±0.93 | 9.0±1.03 | 5.91±0.69 |
| ChebNetII | 80.41±3.98 | 75.41±5.72 | 76.53±4.29 | 76.53±4.59 | 76.94±5.0 | 78.78±5.61 | 78.88±5.2 | 77.45±4.9 | 76.94±5.72 | 77.55±5.51 |
| Acc Gap | 19.59±3.78 | 24.59±5.2 | 23.47±4.59 | 23.47±4.49 | 23.06±4.8 | 21.22±5.61 | 21.12±5.82 | 22.55±4.49 | 23.06±5.61 | 22.45±5.31 |
| Loss Gap | 0.74±0.14 | 1.2±0.44 | 1.15±0.29 | 1.28±0.3 | 1.23±0.33 | 1.11±0.29 | 1.16±0.26 | 1.21±0.29 | 1.24±0.27 | 1.1±0.27 |
| JacobiConv | 52.24±5.41 | 80.92±3.78 | 75.31±5.31 | 74.39±3.78 | 79.08±3.67 | 78.67±4.08 | 80.0±3.06 | 73.67±6.33 | 77.65±5.41 | 78.06±5.61 |
| Acc Gap | 47.76±5.31 | 19.08±3.98 | 24.69±5.0 | 25.61±3.67 | 20.92±3.47 | 21.33±3.67 | 20.0±3.06 | 26.33±6.84 | 22.35±5.1 | 21.94±5.41 |
| Loss Gap | 2.54±0.42 | 0.89±0.2 | 1.1±0.25 | 1.18±0.27 | 0.9±0.17 | 0.97±0.16 | 0.93±0.13 | 1.22±0.39 | 0.97±0.26 | 0.94±0.24 |
| GPRGNN | 53.88±4.8 | 49.18±5.1 | 46.73±5.82 | 45.82±6.64 | 46.12±5.41 | 45.61±5.2 | 46.43±4.59 | 46.12±5.0 | 47.55±4.8 | 46.84±6.22 |
| Acc Gap | 46.12±4.9 | 50.82±5.31 | 53.27±5.61 | 54.18±6.63 | 53.88±5.72 | 54.39±5.2 | 53.57±4.9 | 53.88±4.9 | 52.45±5.1 | 53.16±6.43 |
| Loss Gap | 2.6±0.44 | 3.21±0.53 | 3.5±0.67 | 3.6±0.63 | 3.58±0.63 | 3.51±0.64 | 3.47±0.48 | 3.44±0.61 | 3.22±0.73 | 3.35±0.83 |
| BernNet | 76.73±3.67 | 75.92±2.45 | 75.61±3.67 | 77.04±3.88 | 77.14±4.39 | 75.2±4.7 | 74.9±5.72 | 75.2±5.2 | 74.8±5.92 | 75.71±5.71 |
| Acc Gap | 23.27±3.67 | 24.08±2.65 | 24.39±3.57 | 22.96±3.98 | 22.86±4.29 | 24.8±4.69 | 25.1±5.2 | 24.8±5.61 | 25.2±6.02 | 24.29±5.61 |
| Loss Gap | 0.96±0.22 | 0.95±0.18 | 1.01±0.17 | 1.02±0.21 | 1.06±0.21 | 1.13±0.25 | 1.19±0.31 | 1.18±0.26 | 1.27±0.34 | 1.25±0.31 |

Table 7: Testing accuracy, accuracy gap, loss gap of spectral GNNs on Texas dataset of edge homophilic ratio $H_{edge} = 0.11$ when $K \in [1, 10]$. Small accuracy and loss gaps imply good generalization capability.

