# OpenReview forum: "Generalization of Spectral Graph Neural Networks"
_ICLR.cc/2025/Conference — Submitted to ICLR 2025_

### Official Review · Reviewer_VkKC · 2024-10-22

**Soundness:** 3
**Presentation:** 3
**Contribution:** 4
**Rating:** 6
**Confidence:** 3

**Summary:**

This paper explored the generalization capabilities of Spectral GNNs, focusing on the relationship between generalization capabilities, graph homophily, and node class distribution in node classification tasks. It also investigated the connection between the polynomial order in the architecture of spectral GNNs and generalization error. Through rigorous theoretical analysis and experimental validation, the paper showed that spectral GNNs performed worse on graphs with weak homophily or heterophily but generalized well on graphs with strong homophilic or heterophilic structures. At the same time, it also provided a detailed analysis of the relationship between polynomial order and generalization error.

**Strengths:**

1. The structure of the paper is clear and easy to read, and it is well-organized；

2. The paper is highly theoretical, offering a rigorous theoretical derivation process for its research problem；

3. The paper is somewhat pioneering, providing a great perspective on the study of the generalization performance of spectral GNNs.

**Weaknesses:**

1. The research is based on generalization error. In my opinion, the authors should also incorporate additional metrics to validate the accuracy of their theory. For instance, comparing training and test loss over a specific number of epochs would closely align with the notion of generalization error. Additionally, computing one or two sets of generalization error on both synthetic and real-world datasets used in the paper would further illustrate their point effectively.

2. Datasets are small in experiments. The authors might consider using larger datasets, such as those mentioned in link [1], to further validate their theory. Only a few additional experiments, perhaps just one or two, would be sufficient to make the point, as certain properties of Spectral GNNs may differ when applied to larger datasets.

3. In line 512-571, the author wrote that "These observations align with our theoretical analysis." However, in Figure 4, the test accuracy for JacobiConv and in Figure 4 for ChebNet does not show a linear relationship with the K value, and the curves exhibit significant fluctuations. This seems inconsistent with the conclusions drawn by the authors in Proposition 16. Could the authors provide an explanation for the discrepancy between their conclusions and the experimental results observed in Figure 3 and Figure 4?

4. It would be better to provide codes to ensure the reproducibility.

&nbsp;&nbsp;&nbsp;[1] https://ogb.stanford.edu/docs/nodeprop/

**Questions:**

Refer to Weaknesses.

---

> ### Author Response · Authors · 2024-11-25
>
> > Q1: The research is based on generalization error. In my opinion, the authors should also incorporate additional metrics to validate the accuracy of their theory. For instance, comparing training and test loss over a specific number of epochs would closely align with the notion of generalization error. Additionally, computing one or two sets of generalization error on both synthetic and real-world datasets used in the paper would further illustrate their point effectively.
>
> A1:
>
> (1) This is already addressed in our experiments. We use two empirical metrics to evaluate the generalization error bound: (1) the gap between training and testing accuracy, and (2) the gap between training and testing loss. Following established practices from prior work [1,2], we compare these two gaps after 300 iterations.
>
> - Results for synthetic datasets are presented in Fig.1(a-c).
> - Results for real-world datasets are shown in Fig.1(g-i).
>
> These metrics provide a clear and practical validation of our theoretical findings on both synthetic and real-world datasets.
>
> (2) Calculating the exact values of the constants ($Lip(\ell)$, $Lip(\Psi)$,$Smt(\ell)$, $Smt(\Psi)$) required for our generalization bound is infeasible because determining these values is NP-hard [3]. As a result, directly plotting the precise generalization bound in Theorem 9 is not practical. This challenge is not unique to our work and applies to all generalization bounds that rely on Lipschitz constants, including those based on Rademacher complexity [1,4] or uniform stability [5,6]. Following established practices in validating bound empirically, we correlate it with the accuracy gap (difference between training and testing accuracy) and the loss gap (difference between training and testing loss).
>
> > Q2: Datasets are small in experiments. The authors might consider using larger datasets, such as those mentioned in link [1], to further validate their theory. Only a few additional experiments, perhaps just one or two, would be sufficient to make the point, as certain properties of Spectral GNNs may differ when applied to larger datasets.
>
> A2:
>
> We conducted additional experiments on the larger datasets Ogbn-Arxiv and Ogbn-Products to validate our theory. The relationship between the generalization error bound and graph homophily $H_{edge}$ for these datasets is shown in Fig.1(g-i) with a fixed training sample size of $m = 100$. The results on these larger datasets exhibit similar trends to those observed on smaller real-world datasets, confirming the consistency of our findings.
>
>
> These additional experiments strengthen our results and demonstrate that the properties of spectral GNNs generalize well to larger datasets.
>
>
>
> References:
>
> [1] Tang, Huayi, and Yong Liu. "Towards understanding generalization of graph neural networks." International Conference on Machine Learning. PMLR, 2023.
>
> [2] Cong, Weilin, Morteza Ramezani, and Mehrdad Mahdavi. "On provable benefits of depth in training graph convolutional networks." Advances in Neural Information Processing Systems 34 (2021): 9936-9949.
>
> [3] Virmaux, Aladin, and Kevin Scaman. "Lipschitz regularity of deep neural networks: analysis and efficient estimation." Advances in Neural Information Processing Systems 31 (2018).
>
> [4] Esser, Pascal, Leena Chennuru Vankadara, and Debarghya Ghoshdastidar. "Learning theory can (sometimes) explain generalisation in graph neural networks." Advances in Neural Information Processing Systems 34 (2021): 27043-27056.

---

> ### Author Response · Authors · 2024-11-25
>
> > Q3: In line 512-571, the author wrote that "These observations align with our theoretical analysis." However, in Figure 4, the test accuracy for JacobiConv and in Figure 4 for ChebNet does not show a linear relationship with the K value, and the curves exhibit significant fluctuations. This seems inconsistent with the conclusions drawn by the authors in Proposition 16. Could the authors provide an explanation for the discrepancy between their conclusions and the experimental results observed in Figure 3 and Figure 4?
>
> A3: We conduct experiments to explain the condition in Theorem 14 and Proposition 16, details are provided in Section 6 in our revised version.
>
> **(1) Experimental Validation:**
>
> We conducted additional experiments to demonstrate that when the condition in Proposition 16 is satisfied, the generalization error bound tends to increase with the polynomial order $K$. Specifically, we analyzed the relationship between the accuracy gap, loss gap, and the conditions $\rho_1 = \sum_{k=2}^K \theta_k \frac{(k-1)!}{2^{k-1}}$ and $\rho_2 = \sum_{k=2}^K \theta_k^2 \frac{(k-1)!}{2^k}$.
>
> Using JacobiConv and BernNet on both a synthetic dataset ($H_{\text{edge}} = 0.2$) and the real-world Chameleon dataset ($H_{\text{edge}} = 0.24$), we observed that when the slope of $\rho_1$ is smaller than that of $\rho_2$, the accuracy gap and loss gap increase (shown in Figure 2). This aligns with Proposition 16, which predicts that if $\rho_1$ grows slower than $\rho_2$, the generalization error bound will increase with $K$. More details are provided in Section 6 of the revised version.
>
> **(2) Intuitive Explanation of Conditions:**
>
> 1. **Non-Negative $\theta_k$:**
>    When $\theta_k$ is constrained to $0 \leq \theta_k \leq 1$, spectral GNNs tend to generalize well. In this case, $\theta_k \geq \theta_k^2$, ensuring that $\rho_1$ grows faster than $\rho_2$. This violates the condition in Proposition 16, preventing the generalization error bound from increasing with $K$. For example, BernNet enforces non-negative $\theta_k$, and as shown in Fig. 2(e-h), its accuracy and loss gaps remain stable as $K$ increases.
>
> 2. **Unrestricted $\theta_k$:**
>    When $\theta_k$ is not restricted (allowing both positive and negative values), spectral GNNs may generalize poorly. If $\theta_k < 0$, it often results in $\rho_1 \leq \rho_2$. Additionally, when some $\theta_k$ are negative ($\theta_{k_1} < 0$) and others are positive ($\theta_{k_2} > 0$), $\rho_1$ typically grows slower than $\rho_2$. This satisfies the condition in Proposition 16, causing the generalization error bound to increase with $K$. For example, JacobiConv, which allows unrestricted $\theta_k$, shows increasing accuracy and loss gaps with $K$ in Fig. 2(a-b).
>
> > Q4: It would be better to provide codes to ensure the reproducibility.
>
> A4: Link of codes will be provided in the camera-ready version.

---

> > ### Comment · Reviewer_VkKC · 2024-11-25
> >
> > Thanks for your explanation, it answers most of my questions, I will raise my score!

---

> > > ### Author Response · Authors · 2024-11-26
> > >
> > > We sincerely appreciate your constructive feedback and the time you've dedicated to reviewing our work. Thank you for your response and support.

---

### Official Review · Reviewer_5ECx · 2024-10-29

**Soundness:** 3
**Presentation:** 3
**Contribution:** 2
**Rating:** 5
**Confidence:** 4

**Summary:**

This paper studies the impact of node label distribution and polynomial degree on the generalization capability of spectral GNNs for the task of node classification.

Through a uniform transductive stability analysis of spectral GNNs, the authors provide a generic error bound for spectral GNNs, highlighting the impact of expected prediction error and $k$-length walks on the stability and generalization of the GNNs. Then, the authors specify the bound in the case of a graph with 2 classes, showing that in this case high or low homophily correspond to good generalization, while medium homophily hinders generalization, and they prove that, under some assumptions, increasing the order of the polynomial in the spectral GNN leads to worse generalization.

**Strengths:**

- The paper is well-written and contributions are clearly highlighted
- The theoretical results are interesting, well introduced and clearly explained
- The experiments on synthetic datasets validate the theoretical claims

**Weaknesses:**

- Throughout the paper, both in the theoretical and in the experimental results, no analysis of the depth and width of these spectral GNNs is conducted. It seems at some that the work focuses on a single layer GNN but this is not explicit. As such, the claims of the paper do not hold for a general setting as GNNs are typically deeper and wider (multiple features), which imply layers of filter banks. The effect of these filter banks and how their combined spectrum affects generalisation abilities is unclear.
- The experiments on real-world datasets show, in some cases, inconsistent results compared to the theoretical claims (e.g., as the authors point out in lines 484-485, the accuracy and loss gaps do not decrease as $H_{edge}$ increases from high to medium heterophily). One possible explanation of this is the varying dataset size, as the authors mention, but a more thorough analysis of this behavior would increase the understanding of how the theoretical results apply in practice. One aspect to consider in this sense might be the fact that the final generalization bound is computed in the case of a graph with 2 classes, whereas the real-world graphs used in experiments contain more. This might have a relevant impact, since homophily itself does not characterize GNNs' behavior well and more sophisticated measures of label distributions provide deeper insights, especially when there are more than 2 classes [1,2].
- The above as well as the limitation of the generalisation in terms of transitive stability and node classification, limit drastically the impact of this contribution

**Questions:**

- Regarding the condition in Theorem 14 "when $\sum_{k=2}^K\theta_k(k-1)!/2^{k-1}$ increases more slowly than $\sum_{k=2}^K\theta_k^2(k-1)!/2^{k}$": it is not clear what this means in practice, e.g., how likely this is to hold? Could the authors provide more details about this?

---

> ### Author Response · Authors · 2024-11-25
>
> > Q1: Throughout the paper, both in the theoretical and in the experimental results, no analysis of the depth and width of these spectral GNNs is conducted. It seems at some that the work focuses on a single layer GNN but this is not explicit. As such, the claims of the paper do not hold for a general setting as GNNs are typically deeper and wider (multiple features), which imply layers of filter banks. The effect of these filter banks and how their combined spectrum affects generalisation abilities is unclear.
>
>
> A1:
> Spatial GNNs and spectral GNNs are two main streams of GNNs, each with distinct mechanisms and application areas [1,2]. Spatial GNNs employ a message-passing mechanism and are widely used for graph classification tasks. In contrast, spectral GNNs process data in the spectral domain and achieve strong performance in node classification. Both paradigms are essential and cater to different types of problems.
>
> The architecture discussed in our paper represents a significant branch of popular and widely used spectral GNNs that utilize polynomial graph filters, including ChebNet, GPRGNN, BernNet, etc. Unlike spatial GNNs, which are composed of multiple layers, spectral GNNs do not have a "layer" concept. Instead, their complexity is determined by the highest polynomial order $K$ in their graph filter. This determines the range of information aggregation, allowing the central node to incorporate information from up to $K$-hop neighbors.
>
> The polynomial order $K$ in spectral GNNs plays a role analogous to the number of layers in spatial GNNs in terms of information flow. Our theoretical analysis in Proposition 16 provides conditions under which the generalization error bound increases with $K$. This framework addresses how the polynomial order $K$ in spectral GNNs influences generalization, offering insights relevant to their unique architecture.
>
>
>
> > Q2: The experiments on real-world datasets show, in some cases, inconsistent results compared to the theoretical claims (e.g., as the authors point out in lines 484-485, the accuracy and loss gaps do not decrease as increases from high to medium heterophily). One possible explanation of this is the varying dataset size, as the authors mention, but a more thorough analysis of this behavior would increase the understanding of how the theoretical results apply in practice. One aspect to consider in this sense might be the fact that the final generalization bound is computed in the case of a graph with 2 classes, whereas the real-world graphs used in experiments contain more. This might have a relevant impact, since homophily itself does not characterize GNNs' behavior well and more sophisticated measures of label distributions provide deeper insights, especially when there are more than 2 classes [1,2].
>
> A2:  Both training sample number $m$ and class number $C$ will affect the generalization error bound.
>
> (1) **Effect of Training sample Number ($m$):**
> For the real-world datasets in Figure 2 (original version), we follow the standard data splitting protocol [3,4,5], where 60\% of nodes are used for training. As a result, the number of training samples ($m$) varies significantly across datasets, ranging from 183 samples in Texas to 19,717 samples in Pubmed. This variation in $m$ contributes to the differences observed in the gaps on real-world datasets.
>
> To further investigate this, we conducted two experiments to isolate the effects of $m$ and the homophilic ratio $H_{edge}$:
>
> 1. **Experiment 1:** We fixed the number of training samples to $m = 100$ across all real-world datasets, which exhibit varying homophilic ratios.
> 2. **Experiment 2:** We varied the number of training samples $m$ on synthetic datasets while keeping the homophilic ratio fixed.
>
> The results, now presented in Figure 1 of the revised manuscript, reveal the following:
> - When $m = 100$ is used for all real-world datasets, the behavior of spectral GNNs aligns more closely with that observed on synthetic datasets (Figures 1(a-c) and 1(g-i)).
> - On synthetic datasets, the accuracy and loss gaps consistently decrease as the number of training samples increases (Figures 1(d-f)).
>
>
> These experiments confirm that controlling the training sample size resolves much of the discrepancy.
>
> (2) **Effect of Class Number $C$:**
>
> The generalization error bound increases with the number of classes ($C$) because a larger $C$ leads to a higher gradient norm bound $\beta$, as derived in Eq. (19) and Eq. (39) in the appendix. This, in turn, increases the stability parameter ($\gamma$) and the generalization error bound. However, the impact of $C$ is relatively minor compared to the effect of the training sample size $m$, as typically $C \ll m$.

---

> ### Author Response · Authors · 2024-11-25
>
> > Q3: The above as well as the limitation of the generalisation in terms of transitive stability and node classification, limit drastically the impact of this contribution.
>
> A3:
> We respectfully disagree with this statement.
>
> (1) **Different Learning Settings in GNNs:**
> Generalization analysis for graph classification differs the analysis for node classification due to their distinct learning settings:
> - Graph classification is in an inductive setting setting, where graphs are assumed to be i.i.d.
> - Node classification is in a transductive setting setting, where node features in the test set are available during training, creating a dependency between training and test sets.
>
> Because of these differences, generalization bounds derived for one setting can not be applied to the other. Inductive bounds are not meaningful for transductive settings, and vice versa, unless the bounds are entirely data-independent (e.g., VC bounds). However, data-independent bounds cannot capture the relationship between graph properties and generalization.
>
> (2) **Suitability of Uniform Transductive Stability:**
> Uniform transductive stability is the most appropriate method to analyze the relationship between generalization and graph homophily. Alternative methods, such as:
> - **PAC-Bayesian bounds:** These are unsuitable for transductive settings due to dependencies between training and test data.
> - **Rademacher complexity:** While useful for evaluating a model's capacity to fit noise, this approach focuses solely on graph node features and neglects the role of node labels. Consequently, it cannot capture the combined effects of graph structure and node labels on generalization.
>
> Uniform stability, on the other hand, incorporates node label distributions through gradient analysis, allowing it to effectively analyze the impact of graph homophily on generalization.
>
> (3) **Relevance and Impact of Our Work:**
> By leveraging uniform transductive stability, our work provides meaningful insights into the relationship between generalization and graph homophily in the transductive setting. These contributions are not limited, but rather address a critical aspect of GNN generalization that cannot be effectively analyzed using other methods.
>
> We have added a detailed comparison of generalization bounds in Section 2 to highlight the strengths of our approach and its relevance to existing works.

---

> ### Author Response · Authors · 2024-11-25
>
> > Q4:Regarding the condition in Theorem 14 "when $\sum_{k=2}^K \theta_k \frac{(k-1)!}{2^{k-1}}$ increases more slowly than $\sum_{k=2}^K \theta_k^2 \frac{(k-1)!}{2^k}$ ": it is not clear what this means in practice, e.g., how likely this is to hold? Could the authors provide more details about this?
>
> A4:
> We conduct experiments to explain the condition in Theorem 14 and Proposition 16, details are provided in Section 6 in our revised version. We also have added a new section (Section 5.3) to the revised manuscript that outlines the practical implications of our theoretical findings.
>
>
> **(1) Intuitive Explanation of Conditions:**
>
> 1. **Non-Negative $\theta_k$:**
>    When $\theta_k$ is constrained to $0 \leq \theta_k \leq 1$, spectral GNNs tend to generalize well. In this case, $\theta_k \geq \theta_k^2$, ensuring that $\rho_1$ grows faster than $\rho_2$. This violates the condition in Proposition 16, preventing the generalization error bound from increasing with $K$. For example, BernNet enforces non-negative $\theta_k$, and as shown in Fig. 2(e-h), its accuracy and loss gaps remain stable as $K$ increases.
>
> 2. **Unrestricted $\theta_k$:**
>    When $\theta_k$ is not restricted (allowing both positive and negative values), spectral GNNs may generalize poorly. If $\theta_k < 0$, it often results in $\rho_1 \leq \rho_2$. Additionally, when some $\theta_k$ are negative ($\theta_{k_1} < 0$) and others are positive ($\theta_{k_2} > 0$), $\rho_1$ typically grows slower than $\rho_2$. This satisfies the condition in Proposition 16, causing the generalization error bound to increase with $K$. For example, JacobiConv, which allows unrestricted $\theta_k$, shows increasing accuracy and loss gaps with $K$ in Fig. 2(a-b).
>
>
> **(2) Experimental Validation:**
>
> We conducted additional experiments to demonstrate that when the condition in Proposition 16 is satisfied, the generalization error bound tends to increase with the polynomial order $K$. Specifically, we analyzed the relationship between the accuracy gap, loss gap, and the conditions $\rho_1 = \sum_{k=2}^K \theta_k \frac{(k-1)!}{2^{k-1}}$ and $\rho_2 = \sum_{k=2}^K \theta_k^2 \frac{(k-1)!}{2^k}$.
>
> Using JacobiConv and BernNet on both a synthetic dataset ($H_{\text{edge}} = 0.2$) and the real-world Chameleon dataset ($H_{\text{edge}} = 0.24$), we observed that when the slope of $\rho_1$ is smaller than that of $\rho_2$, the accuracy gap and loss gap increase (shown in Figure 2). This aligns with Proposition 16, which predicts that if $\rho_1$ grows slower than $\rho_2$, the generalization error bound will increase with $K$. More details are provided in Section 6 in the revised version.
>
>
>
> **(3) Practical Guidelines:**
> Our results suggest two key guidelines for improving model design:
>
> (1) Rewiring Graphs:
>    Our analysis demonstrates that graph homophily strongly influences generalization error. Specifically, graphs with higher homophily (or heterophily) tend to exhibit lower variance in $k$-length walks, revealing clearer structural patterns. This reduction in variance leads to a smaller gradient norm bound $\beta$ (Theorem 8, Theorem 13), improving the $\gamma$-uniform transductive stability (Theorem 6). As a result, the generalization error bound decreases (Theorem 9). Therefore, practitioners can enhance the performance of spectral GNNs by focusing on graph structures that maximize homophily or heterophily.
>
> (2) Constrained Graph Convolution:
> Our results show that constraining the convolution parameters $0 \leq \theta_k \leq 1$ helps prevent the generalization error bound from increasing with the polynomial order $K$. This is because the constraint ensures that the condition in Proposition 16, where $\sum_{k=2}^K \theta_k \frac{(k-1)!}{2^{k-1}}$increases slower than $\sum_{k=2}^K \theta_k^2 \frac{(k-1)!}{2^k}$, is violated, as $\theta_k \geq \theta_k^2$. Previous work~\citep{bernnet} reports that constraining $\theta_k$ to non-negative values with Bernstein polynomial basis leads to valid polynomial filters. Our analysis further suggests adding the constraint $\theta_k \leq 1$ to maintain stable generalization error as $K$ increases.

---

> ### Author Response · Authors · 2024-11-25
>
> References:
> [1] Wang, Xiyuan, and Muhan Zhang. "How powerful are spectral graph neural networks." International conference on machine learning. PMLR, 2022.
>
> [2] Balcilar, Muhammet, et al. "Analyzing the expressive power of graph neural networks in a spectral perspective." International Conference on Learning Representations. 2021.
>
> [3] Chien, Eli, et al. "Adaptive universal generalized pagerank graph neural network." arXiv preprint arXiv:2006.07988 (2020).
>
> [4] He, Mingguo, Zhewei Wei, and Hongteng Xu. "Bernnet: Learning arbitrary graph spectral filters via bernstein approximation." Advances in Neural Information Processing Systems 34 (2021): 14239-14251.
>
> [5] He, Mingguo, Zhewei Wei, and Ji-Rong Wen. "Convolutional neural networks on graphs with chebyshev approximation, revisited." Advances in neural information processing systems 35 (2022): 7264-7276.

---

> > ### Comment · Reviewer_5ECx · 2024-11-25
> > **Follow-up Rebuttal**
> >
> > I wish to thank the authors for their replies. I have the following remarks.
> >
> > 	1.	The authors are confused about the distinction about spectral and spatial GNNs. It is incorrect to state the the polynomial order acts as the layer in the spectral GNN. The polynomial order defines the filter order in \emph{each layer}. For example, a ChebNet can have L layers, and in each layer it can in turn have a filter bank of F Chebyshev Polynomials of some order K. This order is the same analogy as the kernel dimension in regular CNN. See e.g., [R1, R2]. The order K does NOT play the same role as the layers. It rather indicates how intermediate features (at some layer l) are aggregated from neighbouring nodes to achieve some frequency response in the filter bank. As such, the analysis conducted here is limited.
> > 	2.	It is unclear to from the answer why there was a discrepancy between theory and experiments. Not it is stated this does not hold anymore by running new experiments. Unfortunately, this answer does not address my concern. A more thorough investigation is needed.
> > 	3.	Regarding Q4 — The arguments given are based on an experimental assessment. These are nice but I feel for these type of work a more theoretical investigation is needed.
> >
> > As such, I feel the paper needs to investigate a more general setting. Either a more general GNN (not single layer) or move beyond node classification, or two classes. There are too many assumptions that limit the significance. As such, I stand to my former judgement.
> >
> > [R1] Defferrard, M., Bresson, X., & Vandergheynst, P. (2016). Convolutional neural networks on graphs with fast localized spectral filtering. Advances in neural information processing systems, 29.
> >
> > [R2] Isufi, E., Gama, F., Shuman, D. I., & Segarra, S. (2024). Graph filters for signal processing and machine learning on graphs. IEEE Transactions on Signal Processing.

---

> > > ### Author Response · Authors · 2024-11-26
> > > **Response to remarks**
> > >
> > > Q1:	The authors are confused about the distinction about spectral and spatial GNNs. It is incorrect to state the the polynomial order acts as the layer in the spectral GNN. The polynomial order defines the filter order in \emph{each layer}. For example, a ChebNet can have L layers, and in each layer it can in turn have a filter bank of F Chebyshev Polynomials of some order K. This order is the same analogy as the kernel dimension in regular CNN. See e.g., [R1, R2]. The order K does NOT play the same role as the layers. It rather indicates how intermediate features (at some layer l) are aggregated from neighbouring nodes to achieve some frequency response in the filter bank. As such, the analysis conducted here is limited.
> > >
> > > A1: We appreciate your comment and provide the following clarifications regarding the distinction between two architectures of spectral GNNs:
> > >
> > > 1. **Early Spectral GNNs Architecture**: The architecture of GNNs in [R1, R2] is given by $ Y = X_L, \, X_l = \alpha \left(\sum_{k=1}^K M^k X_{l-1} H_{lk} \right) $, where $ M $ is a graph matrix, $ X_l $ is the feature at the $ l $-th layer, $ H_{lk} \in \mathbb{R}^{f_l \times f_{l-1}} $, $ f_l $ is the feature dimension of the $ l $-th layer, and $ \alpha $ is an activation function. This describes the architecture of earlier spectral GNNs, such as GCN ($ M^k = D^{-1/2}(I + A)D^{-1/2} $) and ChebNet (where $ M^k $ represents the Chebyshev polynomial basis expanded on the normalized graph Laplacian matrix).
> > >
> > > 2. **Modern Spectral GNNs Architecture**: Recent advances in spectral GNNs do not adhere to this multi-layer architecture. Instead, state-of-the-art spectral GNNs employ a single-layer structure as described in Equation (1) of our paper:
> > >    $\Psi(M, X) = \sigma(g_{\Theta}(M)f_W(X)),$  where $ M \in \mathbb{R}^{n \times n} $ is a graph matrix (e.g., Laplacian or adjacency matrix), $ g_{\Theta}(M) = \sum_{k=0}^K \theta_k T_k(M) $ performs graph convolution using the $ k $-th polynomial basis $ T_k(\cdot) $ and learnable parameters $ \Theta = \{\theta_k\}_{k=0}^K $, $ f_W(X) $ is a feature transformation parameterized by $ W $, and $ \sigma $ is a non-linear activation function (e.g., softmax).
> > >    Recent spectral GNNs, such as GPRGNN, JacobiConv, BernNet, ChebBase, and ChebNetII, adopt this architecture [1, 3, 4, 5], and it serves as the basis for theoretical analysis of spectral GNNs [1, 2, 6].
> > >
> > > We recommend reviewing recent analyses of spectral GNNs [1, 2] for further insights. These works demonstrate the advancements in understanding and formalizing the single-layer spectral GNN framework.
> > >
> > > [1] Wang, Xiyuan, and Muhan Zhang. "How powerful are spectral graph neural networks." International conference on machine learning. PMLR, 2022.
> > >
> > > [2] Balcilar, Muhammet, et al. "Analyzing the expressive power of graph neural networks in a spectral perspective." International Conference on Learning Representations. 2021.

---

> > > ### Author Response · Authors · 2024-11-26
> > > **response to remarks**
> > >
> > > Q2.	It is unclear to from the answer why there was a discrepancy between theory and experiments. Not it is stated this does not hold anymore by running new experiments. Unfortunately, this answer does not address my concern. A more thorough investigation is needed.
> > >
> > > A2:
> > > ### Original Manuscript
> > >
> > > In the original version of our paper, there is a discrepancy between the results of real-world databases and the results of synthetic datasets (Shown in Fig.1 and Fig. 2 in the original version). This was caused by the training sample size difference on real-world datasets. For synthetic datasets, all datasets share same training sample number $m=3,000$ and they vary in edge homophilic ratio $H_{edge}$. But for real-world datasets, we followed the data split in work [3,4] that 60\% nodes are for training. As all real-world datasets have different size, they vary both in training sample number $m$ and the edge homophic ratio $H_{edge}$.
> > >
> > > ### Additional Experiments to Address Discrepancy
> > >
> > > To isolate the effects of $ m $ and $ H_{edge} $, we conducted two new experiments:
> > >
> > > 1. **Experiment 1:**
> > >    Fixed the number of training samples ($ m = 100 $) across all real-world datasets, which exhibit varying homophilic ratios.
> > >
> > > 2. **Experiment 2:**
> > >    Varied the number of training samples ($ m $) on synthetic datasets while keeping the homophilic ratio constant.
> > >
> > > ### Results
> > >
> > > The results, now presented in Figure 1 of the revised manuscript, indicate the following:
> > >
> > > - **Real-World Datasets:**
> > >   When $ m = 100 $ is used for all datasets, the behavior of spectral GNNs aligns more closely with the trends observed on synthetic datasets (Figures 1(a-c) and 1(g-i)).
> > >
> > > - **Synthetic Datasets:**
> > >   On synthetic datasets, the accuracy and loss gaps consistently decrease as the number of training samples increases (Figures 1(d-f)).
> > >
> > > ### Conclusion
> > >
> > > These additional experiments demonstrate that the variations in training sample size ($ m $) are a critical factor contributing to the observed differences between synthetic and real-world datasets. By using same training sample number ($ m=100 $) on all real-world datasets, the relation between generalization error and $H_{edge}$ on real-world datasets show similar trend with that on synthetic datasets.
> > >
> > > Q3.	Regarding Q4 — The arguments given are based on an experimental assessment. These are nice but I feel for these type of work a more theoretical investigation is needed.
> > >
> > > A3: In Q4, the condition that the generalization error increases with the polynomial order ($ K $) is proved in Theorem 14 and we have added links to proof in our revised manuscript. The experiments given in Q4 are to verify this condition in Theorem 14.

---

> > > ### Author Response · Authors · 2024-11-26
> > > **response to remarks**
> > >
> > > S4: As such, I feel the paper needs to investigate a more general setting. Either a more general GNN (not single layer) or move beyond node classification, or two classes. There are too many assumptions that limit the significance. As such, I stand to my former judgement.
> > >
> > >
> > > A4:
> > > ### 1. Generality of the GNN Architecture
> > >
> > > Our generalization analysis is based on the state-of-the-art single-layer spectral GNN architecture. Advanced spectral GNNs such as **JacobiConv**, **BernNet**, and **ChebNetII** all follow this architecture, which represents a significant branch of spectral GNN research.
> > >
> > > ### 2. Node Classes
> > >
> > > Our analysis is not limited to two classes:
> > >
> > > - **Theorems 6, 8, 9, and Lemma 10** all hold for cases where the node class number $ C \geq 3 $.
> > > - **Theorem 13** is derived based on the cSBM model with two classes to provide an explicit, interpretable form. This specific case of cSBM is widely used in theoretical analysis [6, 7].
> > >
> > > ### 3. Generalization Beyond Node Classification
> > >
> > > Generalization analysis for graph classification differs fundamentally from node classification due to the distinct learning settings:
> > >
> > > - **Graph Classification:** Operates in an **inductive setting**, where graphs are assumed to be independent and identically distributed (i.i.d.).
> > > - **Node Classification:** Operates in a **transductive setting**, where node features in the test set are available during training, creating dependencies between training and test sets.
> > >
> > > #### Why These Differences Matter
> > >
> > > - Generalization bounds derived for one setting cannot be directly applied to the other. For example:
> > >   - **Inductive bounds** (e.g., those derived for graph classification) are not meaningful in a transductive setting.
> > >   - **Transductive bounds** (e.g., those derived for node classification) are not meaningful in an inductive setting.
> > > - Data-independent bounds (e.g., VC bounds) fail to capture the relationship between graph properties and generalization.
> > >
> > > #### Framework Differences
> > >
> > > - **Transductive Rademacher Complexity** and **Inductive Rademacher Complexity** follow distinct analysis frameworks [8, 9].
> > > - Similarly, **Transductive Uniform Stability** and **Inductive Uniform Stability** are analyzed through different methodologies [10, 11].
> > >
> > > If there exists a method capable of analyzing data-dependent generalization bounds (except VC) in both **transductive** and **inductive** settings (i.e., applicable to both graph classification and node classification), could you provide references to such work for further investigation?
> > >
> > > Should you have any further questions or require additional clarification, please feel free to reach out to us.
> > >
> > > References:
> > >
> > > [1] Wang, Xiyuan, and Muhan Zhang. "How powerful are spectral graph neural networks." International conference on machine learning. PMLR, 2022.
> > >
> > > [2] Balcilar, Muhammet, et al. "Analyzing the expressive power of graph neural networks in a spectral perspective." International Conference on Learning Representations. 2021.
> > >
> > > [3] Chien, Eli, et al. "Adaptive universal generalized pagerank graph neural network." arXiv preprint arXiv:2006.07988 (2020).
> > >
> > > [4] He, Mingguo, Zhewei Wei, and Hongteng Xu. "Bernnet: Learning arbitrary graph spectral filters via bernstein approximation." Advances in Neural Information Processing Systems 34 (2021): 14239-14251.
> > >
> > > [5] He, Mingguo, Zhewei Wei, and Ji-Rong Wen. "Convolutional neural networks on graphs with chebyshev approximation, revisited." Advances in neural information processing systems 35 (2022): 7264-7276.
> > >
> > > [6] Lu, Kangkang, et al. "Improving Expressive Power of Spectral Graph Neural Networks with Eigenvalue Correction." Proceedings of the AAAI Conference on Artificial Intelligence. Vol. 38. No. 13. 2024
> > >
> > > [6] Deshpande, Yash, et al. "Contextual stochastic block models." Advances in Neural Information Processing Systems 31 (2018).
> > >
> > > [7] Ma, Yao, et al. "Is homophily a necessity for graph neural networks?." arXiv preprint arXiv:2106.06134 (2021).
> > >
> > > [8] Bartlett, Peter L., and Shahar Mendelson. "Rademacher and Gaussian complexities: Risk bounds and structural results." Journal of Machine Learning Research 3.Nov (2002): 463-482.
> > >
> > > [9] El-Yaniv, Ran, and Dmitry Pechyony. "Transductive rademacher complexity and its applications." Journal of Artificial Intelligence Research 35 (2009): 193-234.
> > >
> > > [10] Bousquet, Olivier, and André Elisseeff. "Stability and generalization." The Journal of Machine Learning Research 2 (2002): 499-526.
> > >
> > > [11] El-Yaniv, Ran, and Dmitry Pechyony. "Stable transductive learning." Learning Theory: 19th Annual Conference on Learning Theory, COLT 2006, Pittsburgh, PA, USA, June 22-25, 2006. Proceedings 19. Springer Berlin Heidelberg, 2006.

---

> > > > ### Comment · Reviewer_5ECx · 2024-11-26
> > > > **Update on review**
> > > >
> > > > Thank you for the follow-up clarification. Given these recent spectral methods have shown practical benefits even for a single layer in node classification, I can consider the significance of this paper higher. Please make explicit in the paper that the results hold for a single layer GNN as one could build deeper solutions even with these polynomials. I'll raise my score.
> > > >
> > > > As a clarification, I still feel ChebNetII etc., with a single layer act as a linear input-output mapping and ignore all the hierarchical learning that is key to deep architectures as well as put the nonlinear model in the feature space. In this regard, I feel they remain limited in terms of insights. The fact that they perform so well may be a dataset / task related aspect. It may be good to clarify in the paper why this is the case as I could not find an explanation in the other works besides some standard polynomial approximation theory results.

---

> > > > > ### Author Response · Authors · 2024-11-27
> > > > > **response to update**
> > > > >
> > > > > Thank you for reading our response and also your support.
> > > > >
> > > > > > As a clarification, I still feel ChebNetII etc., with a single layer act as a linear input-output mapping and ignore all the hierarchical learning that is key to deep architectures as well as put the nonlinear model in the feature space. In this regard, I feel they remain limited in terms of insights. The fact that they perform so well may be a dataset / task related aspect. It may be good to clarify in the paper why this is the case as I could not find an explanation in the other works besides some standard polynomial approximation theory results.
> > > > >
> > > > >
> > > > > ### **Reasons**
> > > > > Stacking layers does not help in spectral GNNs compared with deep neural networks for images due to the following key reasons:
> > > > >
> > > > >  1. **Differences Between Graphs and Images**
> > > > >    - **Images:** Grid-like structures in images provide natural hierarchical representations. Adding layers increases the receptive field, enabling the learning of hierarchical and more abstract features.
> > > > >    - **Graphs:** Graphs are irregular and often lack inherent hierarchical structures. Many graphs, such as small-world networks, exhibit dense local community structure.
> > > > >
> > > > >
> > > > >
> > > > >  2. **Parameter Complexity**
> > > > >    GNN require significantly more parameters as the number of layers increases. This rapid increase in parameter number leads to higher computational costs and a greater risk of overfitting.
> > > > >
> > > > >
> > > > > ### **Experiments**
> > > > >
> > > > >
> > > > >
> > > > > We conducted additional experiments to demonstrate that GNN architectures with a single layer of order $K$ generally outperform their counterparts with multiple layers. The experimental results are shown in the table below.
> > > > >
> > > > > 1. **Experimental Setup:**
> > > > > - We follow the semi-supervised node classification setting as described in [1,2].
> > > > > - For the **Texas**, **Wisconsin**, **Actor**, **Chameleon**, and **Squirrel** datasets, we use a split of 2.5\% for training, 2.5\% for validation, and 95\% for testing.
> > > > > - For the **Citeseer**, **Pubmed**, and **Cora** datasets, we use 20 nodes per class for training, 500 nodes for validation, and 1,000 nodes for testing.
> > > > >
> > > > > 2. **Comparison:**
> > > > > We evaluate the performance of the following GNN architectures:
> > > > > - architecture with 2 layers and order 10.
> > > > > - architecture with 1 layer and order 10.
> > > > >
> > > > >
> > > > > The results highlight that the modern architecture's simplified design, even with a single layer, achieves better performance than architecture with stacking layers.
> > > > >
> > > > > | GNN       | Layer | Texas             | Wisconsin         | Actor            | Chameleon         | Squirrel         | Citeseer         | Pubmed          | Cora            |
> > > > > |-----------|-------|-------------------|-------------------|----------------------|-------------------|------------------|------------------|-----------------|-----------------|
> > > > > | **ChebNet** | 2     | 28.79±7.57      | 26.88±9.13      | 24.59±1.46     | 24.45±2.78      | 18.93±2.47     | 66.08±0.68     | 59.63±2.51    | 70.50±2.23    |
> > > > > | **ChebNet** | 1     | 36.76±6.76      | 33.17±7.83      | 24.94±1.33     | 25.65±3.56      | 22.58±2.56     | 65.49±1.39     | 74.88±1.47    | 80.26±0.86    |
> > > > > | **Δ ↑**     |       | +7.97            | +6.29            | +0.35            | +1.20            | +3.65            | -0.59           | +15.25         | +9.76          |
> > > > > | **ChebNetII** | 2    | 27.63±7.52      | 44.46±10.08     | 25.23±1.18     | 24.21±3.53      | 16.37±0.88     | 57.97±1.17     | 70.30±1.33    | 58.33±1.16    |
> > > > > | **ChebNetII** | 1    | 48.44±10.87     | 41.33±9.25      | 29.29±0.80     | 33.48±4.59      | 30.80±2.65     | 70.07±1.13     | 78.87±1.28    | 82.55±0.78    |
> > > > > | **Δ ↑**     |       | +20.81           | -3.13            | +4.06            | +9.27            | +14.43           | +12.10          | +8.57          | +24.22         |
> > > > > | **GPRGNN**  | 2     | 24.80±14.05     | 23.88±9.00      | 18.72±4.28     | 31.07±4.65      | 15.62±3.27     | 53.70±7.01     | 59.41±9.00    | 55.83±7.26    |
> > > > > | **GPRGNN**  | 1     | 48.55±7.00      | 40.79±3.62      | 30.20±0.95     | 30.44±4.29      | 24.33±2.68     | 70.23±1.29     | 79.86±1.81    | 83.50±0.94    |
> > > > > | **Δ ↑**     |       | +23.75           | +16.91           | +11.48           | -0.63            | +8.71            | +16.53          | +20.45         | +27.67         |
> > > > >
> > > > > **Table:** Test accuracy of different architectures of spectral GNNs on real-world datasets. Higher accuracy indicates better performance. **Δ ↑** represents the performance improvement.
> > > > >
> > > > > References:
> > > > >
> > > > > [1] Chien, Eli, et al. "Adaptive universal generalized pagerank graph neural network." arXiv preprint arXiv:2006.07988 (2020).
> > > > >
> > > > > [2] He, Mingguo, Zhewei Wei, and Ji-Rong Wen. "Convolutional neural networks on graphs with chebyshev approximation, revisited." Advances in neural information processing systems 35 (2022): 7264-7276.

---

> > > > > > ### Comment · Reviewer_5ECx · 2024-11-27
> > > > > > **Deeper GNN Follow-Up**
> > > > > >
> > > > > > Thank you for the clarification. I'm actually not convince by the latter comparisons for the following reasons.
> > > > > > 1- Graphs have a different hierarchy than images. More like coarsening, clustering and so on. This is well know in network science.
> > > > > > 2- Deeper architectures with nonlinearities between layers are needed to process "hierarchical representations". Single layer GNNs do not possess this capability, implying, that either their function mapping capabilities do not improve, or that the task requires simple input-output function mappings that a single layer suffices. The latter is often the case in semi-supervised learning.
> > > > > > 3-The parameter increases as in any other deep learning solution. You can control it via number of features and layers.
> > > > > > 4- The numerical comparisons you provided does not fully stand. A grid search over the different hyper parameters is needed to identify the right tuple Layer, Features, Filter Order as well as other regularization hyper parameters.

---

> > > > > > > ### Author Response · Authors · 2024-11-28
> > > > > > > **clarification for deeper GNNs**
> > > > > > >
> > > > > > > > I still feel ChebNetII etc., with a single layer act as a linear input-output mapping and ignore all the hierarchical learning that is key to deep architectures as well as put the nonlinear model in the feature space.
> > > > > > >
> > > > > > >  We would like to clarify that ChebNetII etc., with a single layer **do not** act as a linear input-output mapping. The Weierstrass approximation theorem [1] states that any continuous function defined on a closed interval $([a, b]$ can be uniformly approximated by a polynomial function to any desired accuracy. In the context of a single-layer special GNN architecture $\Psi(M, X) = \sigma(g_{\Theta}(M)f_W(X))$ (defined in our work), $g_{\Theta}(M)$ can be any non-linear function. Single-layer spectral GNNs can learn complex non-linear graph filters. The paper for BernNet [2] has reported several non-linear graph filters (see Figure 6 of the appendix)
> > > > > > >
> > > > > > >
> > > > > > >
> > > > > > > > I'm actually not convince by the latter comparisons for the following reasons. 1- Graphs have a different hierarchy than images. More like coarsening, clustering and so on. This is well know in network science. 2- Deeper architectures with nonlinearities between layers are needed to process "hierarchical representations". Single layer GNNs do not possess this capability, implying, that either their function mapping capabilities do not improve, or that the task requires simple input-output function mappings that a single layer suffices. The latter is often the case in semi-supervised learning. 3-The parameter increases as in any other deep learning solution. You can control it via number of features and layers. 4- The numerical comparisons you provided does not fully stand. A grid search over the different hyper parameters is needed to identify the right tuple Layer, Features, Filter Order as well as other regularization hyper parameters.
> > > > > > >
> > > > > > >
> > > > > > > **(1)** Deeper GNNs are often effective for graph classification but **not** for node classification. Graph classification tasks benefit from hierarchical structures like graph coarsening or multi-scale learning, which capture broader, high-level patterns. In contrast, node classification relies on fine-grained local structures, such as a node's neighbors, edges, and their features. Aggregating these local structures into higher-level representations may obscure the detailed information crucial for accurate node classification. Therefore, hierarchical learning in deeper GNNs does not necessarily improve node classification performance.
> > > > > > >
> > > > > > >
> > > > > > >
> > > > > > > **(2)** It is reasonable to assume that deeper GNNs generally involve more parameters compared to single-layer GNNs. This aligns with standard trends in deep learning architectures.
> > > > > > >
> > > > > > > **(3)** Our grid search focuses on hyperparameters such as learning rate, weight decay rate, and dropout rate. For hidden feature dimensions ($F_l$), layer numbers ($L$), and polynomial orders ($K$), we used fixed values for simplicity for comparisons.
> > > > > > >
> > > > > > > References:
> > > > > > >
> > > > > > > [1] Stone, Marshall H. "The generalized Weierstrass approximation theorem." Mathematics Magazine 21.5 (1948): 237-254.
> > > > > > >
> > > > > > > [2] He, Mingguo, Zhewei Wei, and Hongteng Xu. "Bernnet: Learning arbitrary graph spectral filters via bernstein approximation." Advances in Neural Information Processing Systems 34 (2021): 14239-14251.

---

> > > > > > > > ### Comment · Reviewer_5ECx · 2024-11-28
> > > > > > > > **Follow-up on depth**
> > > > > > > >
> > > > > > > > 1- To clarify, with linear mapping, I meant in the input to this filter. The result you are stating is classic in polynomial approximation theory with orthogonal polynomials. In the end, fixed those coefficients, we have a polynomial matrix H, and and output y = Hx as in graph filters. A deeper GNN will ensure to learn nonlinear mapping here in a hierarchical manner. In these solutions, nonlinearity is placed in the feature spaces via MLP in a form akin to kernel techniques.
> > > > > > > >
> > > > > > > > 2- Yes, I agree. But that is not necessarily a bad thing or not necessarily can lead to overfitting. One can say the same for higher order polynomials.
> > > > > > > >
> > > > > > > > 3- Indeed, fixing the most important aspects in a GNN, depth and width does not lead to a meaningful and conclusive comparison scientifically speaking.

---

> > > > > > > > > ### Author Response · Authors · 2024-11-29
> > > > > > > > > **summary**
> > > > > > > > >
> > > > > > > > > Thank you for your comment.
> > > > > > > > >
> > > > > > > > > Our work focuses on analyzing the generalization of spectral GNNs within the context of the latest architectures.
> > > > > > > > >
> > > > > > > > > **Developing a new spectral GNN architecture with multiple layers falls outside the scope of this study.** Could the reviewer kindly clarify how the inclusion of hierarchical or multilayer architectures relates to the significance of our contribution?
> > > > > > > > >
> > > > > > > > > Are you suggesting that the single-layer design of recent spectral GNNs, such as ChebNetII and BernNet, diminishes their significance?

---

### Official Review · Reviewer_5ppb · 2024-11-03

**Soundness:** 3
**Presentation:** 3
**Contribution:** 2
**Rating:** 6
**Confidence:** 4

**Summary:**

This work analyzes the generalization ability of spectral GNN in node classification tasks. The authors first bound the generalization error with uniform transductive stability, then analyze the effect of graph homophily/heterophily and polynomial order on stability. Specifically, the authors prove that graphs with strong homophily/heterophily improve model generalization by stabilizing models, while increased polynomial order will increase generalization error when specific conditions hold, leading to varying behavior in real world datasets. Finally, the authors provide experiments to support their findings.

**Strengths:**

- This work builds the bridge between generalization and uniform transductive stability on spectral GNNs, and explores the effect of graph homophily/heterophily and polynomial order on stability, which is overall original and provides some insight into this area.
- The paper is and well-organised and easy to follow.

**Weaknesses:**

This work does provide insight into the effect of graph homophily/heterophily and polynomial order K on generalization. However, I would suggest the following weaknesses:

1. The conditions for K may or may not hold in real-world datasets. As a result, this finding may not offer substantial guidance for future improvements.
2. The authors do not provide sufficient explanation or intuitive reasoning regarding the conditions for K.
3. Although the authors acknowledge that the work is limited to a specialized cSBM, I would suggest the authors include explanations on why this specialized cSBM is useful when analyzing generalization.

**Questions:**

Please refer to the weakness

---

> ### Author Response · Authors · 2024-11-25
>
> > Q1 and Q2: The conditions for K may or may not hold in real-world datasets. As a result, this finding may not offer substantial guidance for future improvements. The authors do not provide sufficient explanation or intuitive reasoning regarding the conditions for K.
>
> A1 and A2:
> Thanks for your comments. We conduct experiments to explain the condition in Theorem 14 and Proposition 16, details are provided in Section 6 in our revised version. We also have added a new section (Section 5.3) to the revised manuscript that outlines the practical implications of our theoretical findings.
>
> **(1) Experimental Validation:**
>
> We conducted additional experiments to demonstrate that when the condition in Proposition 16 is satisfied, the generalization error bound tends to increase with the polynomial order $K$. Specifically, we analyzed the relationship between the accuracy gap, loss gap, and the conditions $\rho_1 = \sum_{k=2}^K \theta_k \frac{(k-1)!}{2^{k-1}}$ and $\rho_2 = \sum_{k=2}^K \theta_k^2 \frac{(k-1)!}{2^k}$.
>
> Using JacobiConv and BernNet on both a synthetic dataset ($H_{\text{edge}} = 0.2$) and the real-world Chameleon dataset ($H_{\text{edge}} = 0.24$), we observed that when the slope of $\rho_1$ is smaller than that of $\rho_2$, the accuracy gap and loss gap increase (shown in Figure 2). This aligns with Proposition 16, which predicts that if $\rho_1$ grows slower than $\rho_2$, the generalization error bound will increase with $K$. More details are provided in Section 6 of the revised version.
>
> **(2) Intuitive Explanation of Conditions:**
>
> 1. **Non-Negative $\theta_k$:**
>    When $\theta_k$ is constrained to $0 \leq \theta_k \leq 1$, spectral GNNs tend to generalize well. In this case, $\theta_k \geq \theta_k^2$, ensuring that $\rho_1$ grows faster than $\rho_2$. This violates the condition in Proposition 16, preventing the generalization error bound from increasing with $K$. For example, BernNet enforces non-negative $\theta_k$, and as shown in Fig. 2(e-h), its accuracy and loss gaps remain stable as $K$ increases.
>
> 2. **Unrestricted $\theta_k$:**
>    When $\theta_k$ is not restricted (allowing both positive and negative values), spectral GNNs may generalize poorly. If $\theta_k < 0$, it often results in $\rho_1 \leq \rho_2$. Additionally, when some $\theta_k$ are negative ($\theta_{k_1} < 0$) and others are positive ($\theta_{k_2} > 0$), $\rho_1$ typically grows slower than $\rho_2$. This satisfies the condition in Proposition 16, causing the generalization error bound to increase with $K$. For example, JacobiConv, which allows unrestricted $\theta_k$, shows increasing accuracy and loss gaps with $K$ in Fig. 2(a-b).
>
> **(3) Practical Guidelines:**
> Our results suggest two key guidelines for improving model design:
>
> (1) Rewiring Graphs:
>    Our analysis demonstrates that graph homophily strongly influences generalization error. Specifically, graphs with higher homophily (or heterophily) tend to exhibit lower variance in $k$-length walks, revealing clearer structural patterns. This reduction in variance leads to a smaller gradient norm bound $\beta$ (Theorem 8, Theorem 13), improving the $\gamma$-uniform transductive stability (Theorem 6). As a result, the generalization error bound decreases (Theorem 9). Therefore, practitioners can enhance the performance of spectral GNNs by focusing on graph structures that maximize homophily or heterophily.
>
> (2) Constrained Graph Convolution:
> Our results show that constraining the convolution parameters $0 \leq \theta_k \leq 1$ helps prevent the generalization error bound from increasing with the polynomial order $K$. This is because the constraint ensures that the condition in Proposition 16, where $\sum_{k=2}^K \theta_k \frac{(k-1)!}{2^{k-1}}$increases slower than  $\sum_{k=2}^K \theta_k^2 \frac{(k-1)!}{2^k}$, is violated, as $\theta_k \geq \theta_k^2$. Previous work~\citep{bernnet} reports that constraining $\theta_k$ to non-negative values with Bernstein polynomial basis leads to valid polynomial filters. Our analysis further suggests adding the constraint $\theta_k \leq 1$ to maintain stable generalization error as $K$ increases.

---

> ### Author Response · Authors · 2024-11-25
>
> > Q3: Although the authors acknowledge that the work is limited to a specialized cSBM, I would suggest the authors include explanations on why this specialized cSBM is useful when analyzing generalization.
>
> A3:
> Thank you for the suggestions. We have added the motivation for using cSBM in Section 1. In summary, cSBM is useful for analyzing generalization due to the following reasons:
>
> 1. **Relevance to Real-World Datasets:** cSBM has been shown to effectively model real-world datasets such as Citeseer, Cora, and Polblogs, which are widely used in GNN research [1,2,3,4].
>
> 2. **Flexibility Graph Homophily:** cSBM can generate graphs with well-defined block structures. This allows us to model both homophilic graphs, where nodes within the same block are more likely to be connected, and heterophilic graphs, where connections are more likely between nodes in different blocks.
>
> 3. **Analytical Tractability:** cSBM provides a clear and systematic framework for theoretical analysis. By leveraging its properties, we can systematically vary graph homophily and study its impact on GNN generalization, which would be challenging with more complex or less structured graph models.
>
> These features make cSBM an ideal choice for exploring generalization properties in GNNs.
>
>
> References:
>
> [1] Deshpande, Yash, et al. "Contextual stochastic block models." Advances in Neural Information Processing Systems 31 (2018).
>
>
> [2] Lei, Jing. "A goodness-of-fit test for stochastic block models." (2016): 401-424.
>
>
> [3] Dreveton, Maximilien, Felipe Fernandes, and Daniel Figueiredo. "Exact recovery and Bregman hard clustering of node-attributed Stochastic Block Model." Advances in Neural Information Processing Systems 36 (2024).
>
>
> [4] Zhang, Yingxue, et al. "Detection and defense of topological adversarial attacks on graphs." International Conference on Artificial Intelligence and Statistics. PMLR, 2021.

---

> ### Author Response · Authors · 2024-11-29
>
> We would greatly appreciate your feedback on our rebuttal whenever you have a chance - thank you.

---

> ### Author Response · Authors · 2024-12-02
>
> The deadline for discussion is approaching.
>
> We would greatly appreciate your feedback on our rebuttal whenever you have a chance - thank you.

---

### Official Review · Reviewer_g817 · 2024-11-04

**Soundness:** 3
**Presentation:** 3
**Contribution:** 3
**Rating:** 6
**Confidence:** 2

**Summary:**

The paper studies the generalization capabilities of spectral graph neural network. The analysis  is based on the theory of uniform transductive stability. The paper theoretically show that generalisation bound is influenced by graph homophily, the polynomial order of the spectral GNN and the cardinality of the training set as follow: 1) a spectral GNN with polynoms of order K performs better on strongly homophilic and strongly heterophilic datasets; 2) if the spectral GNN parameters fulfill certain conditions, the generalisation error bound increases with the order of polynom K (which might explain the poor performance of spectral GNN with high K, despite the increase in expressivity) and 3) increased in the number of training samples improves generalisation. The theoretical results are validated both on synthetic and real-world datasets, for 5 different spectral GNN architectures.

**Strengths:**

- Analysing the connection between generalization bound, dataset caracteristiques ( degree of homophily/heterophiliy..) and architecture (polynom’s degree) is an interesting topic that can help developing more powerful architectures, better tuned for a specific problem.
- The paper is well-structured, with a smooth progression from one finding to the next one. Significant effort has been made to present the findings in an intuitive and practical manner.
- The design of the experiments is well aligned with the theoretical findings.

**Weaknesses:**

- While the synthetic experiments clearly match the theoretical results, for the real-world datasets some discrepancies are present. In Figure 2, models with medium degree of heterophily has a similar gap of accuracy compared to strongly homophilic/heterophilic datasets. The authors motivate these results by the influence of training set cardinality, since the datasets not only differ in the homophily level, but also in size.  Indeed, since the generalisation gap is influence by both number of training samples and homophily level, it is hard to disentangle the individual effect of each one of them in the presented real-world experiments. However, this makes it considerably harder  to asses if the theoretical results correlates or not with the practical findings. Does repeating the experiments with a different train-val-split such that the number of training samples remains the same alleviates these discrepancy? Or, given the assumption that n→ infinity it is better to focus on having comparable percentages of nodes in the training set when splitting the data?
- The results in Figure 4 are hard to correlate with the theoretical findings. The theorems show that, under certain assumptions, the gap in accuracy increases with the order of the polynom used by the spectral GNN. However, that is not the case in the experiments provided in Figures 3 and 4. The explanation is that the assumptions might not be fulfilled in some of the scenarios. Is it possible to check or impose that assumption such that the experiments can match the theoretical findings?
- While the paper provide interesting insights into what influence the generalisability of spectral GNNs, it does not provide any guidance towards how can we benefit from these findings.

**Questions:**

Please see the sections above

---

> ### Author Response · Authors · 2024-11-25
>
> > Q1: While the synthetic experiments clearly match the theoretical results, for the real-world datasets some discrepancies are present. In Figure 2, models with medium degree of heterophily has a similar gap of accuracy compared to strongly homophilic/heterophilic datasets. The authors motivate these results by the influence of training set cardinality, since the datasets not only differ in the homophily level, but also in size. Indeed, since the generalisation gap is influence by both number of training samples and homophily level, it is hard to disentangle the individual effect of each one of them in the presented real-world experiments. However, this makes it considerably harder to asses if the theoretical results correlates or not with the practical findings. Does repeating the experiments with a different train-val-split such that the number of training samples remains the same alleviates these discrepancy? Or, given the assumption that n→ infinity it is better to focus on having comparable percentages of nodes in the training set when splitting the data?
>
> A1: To address this concern, we conducted two additional experiments to clarify: (1) the relationship between the generalization error bound and the training sample size, and (2) the relationship between the generalization error bound and graph homophily $H_{edge}$ on real-world datasets.
>
> 1. **Effect of Training Sample Size ($m$):** We increased the training sample size from $m = 500$ to $m = 4,500$ in a step of 500 on a synthetic dataset with a fixed $H_{edge}$. The results show that both the accuracy gap and loss gap decrease as $m$ increases, which aligns with our analysis in Lemma 10 (see Fig.1(d-f)).
>
> 2. **Effect of Graph Homophily ($H_{edge}$) on real-world datasets:** We fixed the training sample size at $m = 100$ across all real-world datasets. The relationships between the accuracy gap, loss gap, and $H_{edge}$ closely resemble those observed in synthetic datasets, consistent with our theoretical analysis in Proposition 16 (see Fig.1(g-i) in the revised version).
>
> These experiments confirm that the theoretical results correlate well with practical findings when training sample size and graph homophily are controlled independently.
>
> Details are included in Section 6 of the revised manuscript
>
> > Q2: The results in Figure 4 are hard to correlate with the theoretical findings. The theorems show that, under certain assumptions, the gap in accuracy increases with the order of the polynomial used by the spectral GNN. However, that is not the case in the experiments provided in Figures 3 and 4. The explanation is that the assumptions might not be fulfilled in some of the scenarios. Is it possible to check or impose that assumption such that the experiments can match the theoretical findings?
>
> A2:  We conducted additional experiments to examine the relationship between the accuracy gap, loss gap, and the conditions specified in Proposition 16. Specifically, we analyzed the relationship between these gaps and the conditions $\rho_1 = \sum_{k=2}^K \theta_k \frac{(k-1)!}{2^{k-1}}$ and $\rho_2 = \sum_{k=2}^K \theta_k^2 \frac{(k-1)!}{2^k}$.
>
> The experiments used JacobiConv and BernNet on both a synthetic dataset with $H_{\text{edge}} = 0.2$ and the real-world Chameleon dataset with $H_{\text{edge}} = 0.24$. The results, presented in Fig. 2, show that when the slope of the line corresponding to $\rho_1$ is smaller than that of $\rho_2$, the accuracy gap and loss gap increase. This is consistent with Proposition 16, which states that the generalization error bound increases with the polynomial order $K$ when $\rho_1$ grows more slowly than $\rho_2$.
>
> These findings align with the theoretical results and confirm that the behavior observed in experiments can be explained when the specified assumptions are met. Details are provided in Section 6 in our revised version.

---

> ### Author Response · Authors · 2024-11-25
>
> > Q3: While the paper provide interesting insights into what influence the generalise ability of spectral GNNs, it does not provide any guidance towards how can we benefit from these findings.
>
> A3:
> We have added a new section (Section 5.3) to the revised manuscript that outlines the practical implications of our theoretical findings.
>
> In summary, our results suggest two key guidelines for improving model design:
>
> (1) Rewiring Graphs:
>    Our analysis demonstrates that graph homophily strongly influences generalization error. Specifically, graphs with higher homophily (or heterophily) tend to exhibit lower variance in $k$-length walks, revealing clearer structural patterns. This reduction in variance leads to a smaller gradient norm bound $\beta$ (Theorem 8, Theorem 13), improving the $\gamma$-uniform transductive stability (Theorem 6). As a result, the generalization error bound decreases (Theorem 9). Therefore, practitioners can enhance the performance of spectral GNNs by focusing on graph structures that maximize homophily or heterophily.
>
> (2) Constrained Graph Convolution:
> Our results show that constraining the convolution parameters $0 \leq \theta_k \leq 1$ helps prevent the generalization error bound from increasing with the polynomial order $K$. This is because the constraint ensures that the condition in Proposition 16, where $\sum_{k=2}^K \theta_k \frac{(k-1)!}{2^{k-1}}$increases slower than  $\sum_{k=2}^K \theta_k^2 \frac{(k-1)!}{2^k}$, is violated, as $\theta_k \geq \theta_k^2$. Previous work~\citep{bernnet} reports that constraining $\theta_k$ to non-negative values with Bernstein polynomial basis leads to valid polynomial filters. Our analysis further suggests adding the constraint $\theta_k \leq 1$ to maintain stable generalization error as $K$ increases.

---

> ### Author Response · Authors · 2024-11-29
>
> We would greatly appreciate your feedback on our rebuttal whenever you have a chance - thank you.

---

> ### Author Response · Authors · 2024-12-02
>
> The deadline for discussion is approaching.
>
> We would greatly appreciate your feedback on our rebuttal whenever you have a chance - thank you.

---

### Official Review · Reviewer_MSpN · 2024-11-04

**Soundness:** 3
**Presentation:** 2
**Contribution:** 2
**Rating:** 3
**Confidence:** 5

**Summary:**

The paper introduces a generalization study of the GNNs based on transductive learning and homophily/heterophily. The paper's results are relevant, and the topic is important. However, the paper's presentation can be improved, and a better discussion of the results should be added. The author's results are not novel per se, given that there were similar results before. Also, the link between homophily/heterophily and generalization is independent of GNNs and only depends on the walk on the graph, which provides little intuition on the implications of them. The numerical experiments are almost uninformative, and the results show a large amount of noise. I believe the topic paper is relevant, but I fail to understand what novelty this paper brings to the community.

**Strengths:**

1. The paper's topic is relevant.
2. To the best of my knowledge, the theoretical results of the paper are correct.

**Weaknesses:**

1. The paper's theory is separated - the generalization results are on one side, and the homophily/heterophily on the other one. This undermines the importance of the results. That is to say, the authors present two relevant results that are disjoint and use different theoretical assumptions. I believe this undermines the importance of the paper given that it should be two different papers, and the novelty and contributions of each should be evaluated separately.

2. Regarding the generalization results based on transductive learning, these results are not novel, and similar results exist in the literature:

 - Transductive Robust Learning Guarantees, Montasser et al.
 - On Inductive–Transductive Learning With Graph Neural Networks Ciano et al.
 - Towards Understanding Generalization of Graph Neural Networks Tang et al.

The authors should explain why their results are important, and how they fit in the crowded landscape.

3. The previous point is regarding novelty, but there is also something to be said about this transductive analysis in and of itself. The graph is completely and almost entirely ignored. Why is this analysis even relevant to the context of GNNs? Why is it valid to analyze GNN while ignoring the graph completely? Even by looking at the assumptions for Theorem 6, the is not a single mention of the graph. The graph could have no edges (be a set of samples) and the results would still hold. In particular, Theorem 6 is a known result.

4. The paper uses a very strong assumption regarding uniform transductive stability. In particular, Theorem 9 is valid because of the strong assumption \gamma-uniform transductive stability. When does this assumption hold? The authors should explain why using this assumption is valid.

4. The graph is introduced in the second part of the paper, and the only relevant features of it are community and label. This largely ignores the progress in graph theory, and graph models that have been used in the last 10+ years. The authors utilize an extremely simple model that does not appear in any real-world problem. What is even worse, in this extremely simple model, the results are impossible to digest; Theorem 13 is unintuitive, and the explanation given with the bullets after it is largely trivial and uninformative. I fail to understand the contribution of these theoretical results.

5. The numerical results are difficult to process and understand. Given that the authors provide a theoretical framework that is very difficult to follow, the numerical results shed no light on any real-world conclusion. Looking at Figure 2, there is little connection to the theory. Even more so, the theory is so detached from practical implementations, that new metrics have to be introduced (H_{edge}, order K for example). On top of that, the results are noisy and no statistical bands are being provided.

**Questions:**

See section weaknesses.

---

> ### Author Response · Authors · 2024-11-25
>
> > Q1: The paper's theory is separated - the generalization results are on one side, and the homophily/heterophily on the other one. This undermines the importance of the results. That is to say, the authors present two relevant results that are disjoint and use different theoretical assumptions. I believe this undermines the importance of the paper given that it should be two different papers, and the novelty and contributions of each should be evaluated separately.
>
> A1: We are unsure what the reviewer means for "the paper's theory is separated ... disjoint and use different theoretical assumption". We believe there is a misunderstanding regarding the connection between the theoretical components of our paper. To clarify:
>
> (1) **Generalization Framework:**
> Our paper proposes a single generalization bound (Theorem 9), where the stability parameter $\gamma$ plays a central role. The stability $\gamma = r\beta$, as outlined in Theorem 6, is composed of two terms: $r$ captures the Lipschitz continuity and smoothness of the loss function and the GNN model while $\beta$ represents the gradient norm bound. We further demonstrate in Theorem 8 that $\beta$ is influenced by the graph structure, node labels, and node features. Specifically:
>    - The graph structure is captured by $A^k$, governed by the parameter $Q$.
>    - Node features are controlled by the parameter $\Pi$, represented through the mean $\pi_{y_i}$ and variance $\Sigma_{y_i}$ of nodes belonging to class $y_i$.
>
> Thus, the relationship between generalization and homophily/heterophily is inherently embedded in our unified framework through the dependence of $\beta$ on graph parameters.
>
> (2) **Theorem 13 as an Specialization of Theorem 8:**
> Theorem 13 is a more explicit formulation of Theorem 8 in the binary case. It allows us to directly estimate the relationship between the edge homophilic ratio ($H_{edge}$) and the gradient norm bound, providing a clear connection between graph homophily and the generalization bound.
>
> In summary, our theoretical results are connected and explains how graph homophily, polynomial order in influencing generalization for spectral GNNs.
>
> > Q2: Regarding the generalization results based on transductive learning, these results are not novel, and similar results exist in the literature: (1)
> Transductive Robust Learning Guarantees, Montasser et al.
> (2) On Inductive–Transductive Learning With Graph Neural Networks Ciano et al.
> (3) Towards Understanding Generalization of Graph Neural Networks Tang et al.
> The authors should explain why their results are important, and how they fit in the crowded landscape.
>
>
> A2: We respectfully clarify that the referenced works do not provide results similar to ours. Here's how our work differs from each: Paper (1) uses VC bounds for generalization analysis, producing error bounds that depend on the graph size $n$. It does not consider other factors such as node label distribution or graph homophily. Paper (2) does not conduct generalization analysis. Instead, it proposes a new mixed inductive–transductive learning framework for graph-structured tasks. Paper (3) employs transductive Rademacher Complexity to study how graph matrix norms affect generalization. However, it does not explore the relationship between node label distribution and generalization, nor does it consider the impact of polynomial order in spectral GNNs.
>
> **Novelty of Our Work:**
> While prior studies have examined factors like graph size, training set size, graph matrix norms, and node features, they have largely overlooked:
> - **The role of graph homophily in generalization:** We explicitly analyze how homophily affects the generalization error bound through the gradient norm bound ($\beta$) in Theorem 8.
> - **The effect of increasing the polynomial order in spectral GNNs:** Our work uniquely gives condition that when generalization error increase with polynomial order $K$ in Theorem 14.
>
> To our knowledge, this is the first study to establish a theoretical link between graph homophily, spectral GNN architecture, and generalization. We have also added a detailed comparison of generalization bounds in Section 2 to highlight the distinct contributions and relevance of our work within the broader literature.

---

> ### Author Response · Authors · 2024-11-25
>
> > Q3: The previous point is regarding novelty, but there is also something to be said about this transductive analysis in and of itself. The graph is completely and almost entirely ignored. Why is this analysis even relevant to the context of GNNs? Why is it valid to analyze GNN while ignoring the graph completely? Even by looking at the assumptions for Theorem 6, the is not a single mention of the graph. The graph could have no edges (be a set of samples) and the results would still hold. In particular, Theorem 6 is a known result.
>
> A3:
> We would like to address the misunderstanding regarding the role of the graph in our transductive analysis:
>
> (1) **Graph Is Central to the Analysis:**
> The graph is not ignored in our transductive analysis. Theorem 6 decomposes the $\gamma$-uniform transductive stability into two components, $\gamma = r\beta$:
> - $r$ captures the Lipschitz continuity and smoothness of the loss function and the spectral GNN.
> - $\beta$ represents the gradient norm bound, which is influenced by graph parameters $(n, f, \Pi, Q)$, as shown in Theorem 8.
>
> Further details of how the graph parameters are incorporated include:
> - **Graph Structure $A^k$:** The term $A^k$ depends on the graph structure, governed by the parameter $Q$.
> - **Node Features $X$:** These are influenced by $\Pi$, represented by the mean $\pi_{y_i}$ and variance $\Sigma_{y_i}$ of nodes within each class $y_i$ in Theorem 8.
>
> These parameters directly affect $\beta$, which in turn influences the $\gamma$-uniform transductive stability. Hence, the graph structure and node features are integrated in our analysis.
>
> (2) **Framework of Uniform Transductive Stability:**
> Uniform transductive stability provides a robust framework for generalization analysis in transductive settings. Theorem 6 follows frameworks established in prior works such as [1,2]. While this framework is not limited to GNNs, our focus is specifically on spectral GNNs. In our analysis:
> - The **polynomial order $K$** of the spectral GNN architecture and the **graph adjacency matrix $A$** play a critical role, as demonstrated in Lemma 23 in appendix, which works as a stepping stone to prove Theorem 8, Theorem 13 and other related analysis.
>
> (3) **Distinct Contributions in Theorem 6:**
> Our approach introduces several distinctions from prior works:
> - We define stability based on the loss function rather than algorithm output in the transductive setting (see Definition 5).
> - We explicitly consider the Lipschitz and smoothness properties of both the loss function and the GNN model.
> - Unlike [3,4], which conduct gradient analysis in the inductive setting, we derive Theorem 6 specifically for the transductive setting.
> - In contrast to [2], which relies on algorithm stability and margin loss, we define stability directly on the cross-entropy loss (Definition 5).
>
> If Theorem 6 is considered a "known result," we kindly request a reference to the specific paper for further clarification and comparison.
>
> > Q4: The paper uses a very strong assumption regarding uniform transductive stability. In particular, Theorem 9 is valid because of the strong assumption $\gamma$-uniform transductive stability. When does this assumption hold? The authors should explain why using this assumption is valid.
>
> A4:
> We would like to clarify a potential misunderstanding regarding $\gamma$-uniform transductive stability:
>
> $\gamma$ is not an assumption but a parameter that quantifies stability, as defined in Definition 5 and in Theorem 6. It describes the sensitivity of the model’s predictions to changes in the training data. Our Definition 5 is based on the framework of uniform transductive stability introduced in [1], but with a key difference: we define stability in terms of sample loss, rather than algorithm output, as in [1]. The stability parameter $\gamma$ is explicitly derived in Theorem 6, following the frameworks in [1,2].

---

> ### Author Response · Authors · 2024-11-25
>
> > Q5: The graph is introduced in the second part of the paper, and the only relevant features of it are community and label. This largely ignores the progress in graph theory, and graph models that have been used in the last 10+ years. The authors utilize an extremely simple model that does not appear in any real-world problem. What is even worse, in this extremely simple model, the results are impossible to digest; Theorem 13 is unintuitive, and the explanation given with the bullets after it is largely trivial and uninformative. I fail to understand the contribution of these theoretical results.
>
> A5: We are not sure what 'the second part of the paper' mean.
>
> (1) We have added the motivation for using cSBM in Section 1. In summary, cSBM is useful for analyzing generalization due to the following reasons:
>
> 1. **Relevance to Real-World Datasets:** cSBM has been shown to effectively model real-world datasets such as Citeseer, Cora, and Polblogs, which are widely used in GNN research [5,6,7,8].
>
> 2. **Flexibility Graph Homophily:** cSBM can generate graphs with well-defined block structures. This allows us to model both homophilic graphs, where nodes within the same block are more likely to be connected, and heterophilic graphs, where connections are more likely between nodes in different blocks.
>
> 3. **Analytical Tractability:** cSBM provides a clear and systematic framework for theoretical analysis. By leveraging its properties, we can systematically vary graph homophily and study its impact on GNN generalization, which would be challenging with more complex or less structured graph models.
>
> These features make cSBM an ideal choice for exploring generalization properties in GNNs.
>
> (2) To derive the explicit relationship between generalization error bound and graph homophily (edge homophilic ratio $H_{edge}$), architecture of spectral GNNs (order $K$), we consider cSBM$(n, f, \mu, u , \lambda, d)$, a well-studied specialization of the general multi-class cSBM [5]. Theorem 13 highlights how parameters like $d$ and $\lambda$ influence the graph structure by affecting $c_{in} = d+ \lambda\sqrt{d},c_{out} = d- \lambda\sqrt{d}$ and $E[H_{edge}]=\frac{c_{in}}{c_{in}+c_{out}}$ in Proposition 12. We explain in detail in cases that $c_{in}\geq c_{out}$ and $c_{in}\leq c_{out}$, how the expectation $E[A^k_{ij}]$ will change and thus affect the gradient norm bound $\beta$ in Theorem 13 and then the uniform transductive stability in Theorem 6 as $\gamma=r\beta$.

---

> ### Author Response · Authors · 2024-11-25
>
> > Q6: The numerical results are difficult to process and understand. Given that the authors provide a theoretical framework that is very difficult to follow, the numerical results shed no light on any real-world conclusion. Looking at Figure 2, there is little connection to the theory. Even more so, the theory is so detached from practical implementations, that new metrics have to be introduced ($H_{edge}$, order $K$ for example). On top of that, the results are noisy and no statistical bands are being provided.
>
> A6:
> (1)  $H_{edge}$ is a parameter to quantify the graph homophily. $K$ is a parameter of architecture of spectral GNNs.
>
> (2) For the real-world datasets in Figure 2 (original version), we follow the standard data splitting protocol~\citep{bernnet, chebii, gprgnn}, where 60\% of nodes are used for training. As a result, the number of training samples ($m$) varies significantly across datasets, ranging from 183 samples in Texas to 19,717 samples in Pubmed. This variation in $m$ contributes to the differences observed in the gaps on real-world datasets.
>
> To further investigate this, we conducted two additional experiments to isolate the effects of $m$ and the homophilic ratio $H_{edge}$:
>
> 1. **Experiment 1:** We fixed the number of training samples to $m = 100$ across all real-world datasets, which exhibit varying homophilic ratios.
> 2. **Experiment 2:** We varied the number of training samples $m$ on synthetic datasets while keeping the homophilic ratio fixed.
>
> The results, now presented in Figure 1 of the revised manuscript, reveal the following:
> - When $m = 100$ is used for all real-world datasets, the behavior of spectral GNNs aligns more closely with that observed on synthetic datasets (Figures 1(a-c) and 1(g-i)).
> - On synthetic datasets, the accuracy and loss gaps consistently decrease as the number of training samples increases (Figures 1(d-f)).
>
> These additional experiments confirm that the variations in training sample size ($m$) play a critical role in the observed differences between synthetic and real-world datasets.
>
> (3) We have added a new section (Section 5.3) to the revised manuscript that outlines the practical implications of our theoretical findings.
>
> In summary, our results suggest two key guidelines for improving model design:
>
> - Rewiring Graphs:  Our analysis demonstrates that graph homophily strongly influences generalization error. Specifically, graphs with higher homophily (or heterophily) tend to exhibit lower variance in $k$-length walks, revealing clearer structural patterns. This reduction in variance leads to a smaller gradient norm bound $\beta$ (Theorem 8, Theorem 13), improving the $\gamma$-uniform transductive stability (Theorem 6). As a result, the generalization error bound decreases (Theorem 9). Therefore, practitioners can enhance the performance of spectral GNNs by focusing on graph structures that maximize homophily or heterophily.
>
> - Constrained Graph Convolution:
> Our results show that constraining the convolution parameters $0 \leq \theta_k \leq 1$ helps prevent the generalization error bound from increasing with the polynomial order $K$. This is because the constraint ensures that the condition in Proposition 16, where $\sum_{k=2}^K \theta_k \frac{(k-1)!}{2^{k-1}}$increases slower than  $\sum_{k=2}^K \theta_k^2 \frac{(k-1)!}{2^k}$, is violated, as $\theta_k \geq \theta_k^2$. Previous work~\citep{bernnet} reports that constraining $\theta_k$ to non-negative values with Bernstein polynomial basis leads to valid polynomial filters. Our analysis further suggests adding the constraint $\theta_k \leq 1$ to maintain stable generalization error as $K$ increases.
>
> (4) Standard deviations in experimental results are provided in Table 4, Table 5, Table 6, Table 7 in appendix.

---

> ### Author Response · Authors · 2024-11-25
>
> References:
>
> [1] El-Yaniv, Ran, and Dmitry Pechyony. "Stable transductive learning." Learning Theory: 19th Annual Conference on Learning Theory, COLT 2006, Pittsburgh, PA, USA, June 22-25, 2006. Proceedings 19. Springer Berlin Heidelberg, 2006.
>
>
> [2] Cong, Weilin, Morteza Ramezani, and Mehrdad Mahdavi. "On provable benefits of depth in training graph convolutional networks." Advances in Neural Information Processing Systems 34 (2021): 9936-9949.
>
>
> [3] Verma, Saurabh, and Zhi-Li Zhang. "Stability and generalization of graph convolutional neural networks." Proceedings of the 25th ACM SIGKDD International Conference on Knowledge Discovery & Data Mining. 2019.
>
>
> [4] Zhou, Xianchen, and Hongxia Wang. "The generalization error of graph convolutional networks may enlarge with more layers." Neurocomputing 424 (2021): 97-106.
>
>
>
> [5] Deshpande, Yash, et al. "Contextual stochastic block models." Advances in Neural Information Processing Systems 31 (2018).
>
>
> [6] Lei, Jing. "A goodness-of-fit test for stochastic block models." (2016): 401-424.
>
>
> [7] Dreveton, Maximilien, Felipe Fernandes, and Daniel Figueiredo. "Exact recovery and Bregman hard clustering of node-attributed Stochastic Block Model." Advances in Neural Information Processing Systems 36 (2024).
>
>
> [8] Zhang, Yingxue, et al. "Detection and defense of topological adversarial attacks on graphs." International Conference on Artificial Intelligence and Statistics. PMLR, 2021.

---

> ### Author Response · Authors · 2024-11-29
>
> We would greatly appreciate your feedback on our rebuttal whenever you have a chance - thank you.

---

> ### Author Response · Authors · 2024-12-02
>
> The deadline for discussion is approaching.
>
> We would greatly appreciate your feedback on our rebuttal whenever you have a chance - thank you.

---

### Official Review · Reviewer_AQtc · 2024-11-04

**Soundness:** 2
**Presentation:** 3
**Contribution:** 2
**Rating:** 3
**Confidence:** 3

**Summary:**

The paper provides generalisation bounds for a specific class of spectral GNNs and assuming a particular graph model (cSBM) derives the stability of the spectral GNN training in terms of generalisation for differrent homophily and heterophily values.

**Strengths:**

- Clear presentation
- Significant technical machinery is introduced (though unclear how interesting or relevant it is given the assumptions and results)
- The authors attempt to validate their high-level conclusion empirically, which is nice!

**Weaknesses:**

TLDR: The assumptions and restrictions of the theoretical analysis are high (full-batch gradient descent and a specific spectral GNN architecture). And it's unclear whether the derived bounds are tight enough to be interesting. The empirical validation is only at a very high-level. If the authors can demonstrate that the bounds are tight, I am willing to raise my score to acceptance.

- Please include proof sketches in the main paper and add links to where the full proof is in the Appendix.
- The work assumes that the GNN is trained with gradient descent and a fixed learning rate. When this is not true in practice, and it is generally known that some optimizers in fact work very poorly. This is fine given the theoretical nature of the paper, but this limitation should be more explicitly addresssed.
- It should be shown that the generalization bound is not vacuous, i.e. that for some realistic values of n and other parameters the the value out of the bound is precise enough to be interesting. In particular it seems to me that assuming a fixed ratio between m and n the generalisation bound quickly gets worse as it relies on the absolute difference of n-m and not the ratio. Even worse in practice the labelled samples m is often quite small compared to n, so much so that O(n-m) approx O(n), which makes for an uninteresting generalisation bound. (I might be missing something here.)
- For the experimental results: it only validates the very high-level conclusion, which is less interesting. Ideally, the generalisation bound and it's tightness would be empirically validated, i.e. plot the generalisation bounds on the graph as a reference else the theory is not very useful.
- Also the presentation of the results could be improved please label the points in Figure 2 for instance to show which datapoint represents which dataset.
- The architecture restriction is quite severe.

**Questions:**

- Theorem 6 is odd to me, it seems to not be specific to spectral GNNs none of the parameters of the graph (n,f, Pi, Q) seem to affect the transductive stability. In fact nothing except the size of the dataset m seems to affect the result. What am I missing?
- Theorem 6 What is beta_2?
- Have you validated how well you can fit the cSBM model to the real world datasets?

---

> ### Author Response · Authors · 2024-11-25
>
> > W1: Please include proof sketches in the main paper and add links to where the full proof is in the Appendix.
>
> A1: Thanks for the suggestion. We have included a proof sketch for the main theorems (Theorems 8 and 13) in the revised manuscript. Hyperlinks to the full proofs, located in the appendix, have also been added for easy reference.
>
> > W2: The work assumes that the GNN is trained with gradient descent and a fixed learning rate. When this is not true in practice, and it is generally known that some optimizers in fact work very poorly. This is fine given the theoretical nature of the paper, but this limitation should be more explicitly addressed.
>
> A2: Thanks for your comments. We acknowledge that in practice, optimizers like Adam, which are commonly used for training spectral GNNs, introduce a discrepancy between our theoretical analysis and real-world applications. In response, we’ve revised Section 7 to explicitly discuss this limitation.
>
> Regarding the fixed learning rate, this is a standard practice for training spectral GNNs [1,2,3]. This differs from training strategies in fields like computer vision (CV) or natural language processing (NLP), where decaying learning rates are more common.
>
> > W3: It should be shown that the generalization bound is not vacuous, i.e. that for some realistic values of n and other parameters the the value out of the bound is precise enough to be interesting. In particular it seems to me that assuming a fixed ratio between m and n the generalisation bound quickly gets worse as it relies on the absolute difference of n-m and not the ratio. Even worse in practice the labelled samples m is often quite small compared to n, so much so that O(n-m) approx O(n), which makes for an uninteresting generalisation bound. (I might be missing something here.)
>
>
> A3: Thanks for your comments. We would like to clarify that our generalization bound is not vacuous, as outlined below:
>
> 1. **Behavior of the Bound:**
>    In Theorem 6, the stability parameter $\gamma = O(1/m)$. When $n$ is sufficiently large, the term $1 / (n - m)$ becomes negligible, and $\Omega(m, n - m)$ grows as $O(m^{1/2})$. Consequently, the generalization error bound decreases at the rate $O(1/m) + O\big(\sqrt{2\log(1/\delta)/m}\big)$. This demonstrates that the bound remains meaningful even when $n$ is large and $m$ is small. Details can be found in remark of Theorem 9 and Lemma 10.
>
> 2. **Impact of Stability ($\gamma$):**
>    The stability parameter $\gamma$ plays a crucial role in the generalization error bound (Theorem 9). $\gamma$ is influenced by both the architecture of spectral GNNs and the graph’s node label distribution. Specifically:
>    $\gamma = r\beta$, where $r$ incorporates the Lipschitz continuity and smoothness of the loss function and GNNs, and $\beta$ represents the gradient norm bound.
>    The gradient norm bound $\beta$ is directly affected by the parameters of the cSBM model $(n, f, \Pi, Q)$, as shown in Theorem 8. These parameters encapsulate the graph structure, node features, and node labels, which implicitly influence the generalization error bound.
>
> By incorporating these graph-specific parameters, our generalization bound is not vacuous and is interesting.
>
> > W4: For the experimental results: it only validates the very high-level conclusion, which is less interesting. Ideally, the generalisation bound and it's tightness would be empirically validated, i.e. plot the generalisation bounds on the graph as a reference else the theory is not very useful.*
>
> A4:
> (1) Calculating the exact values of the constants ($Lip(\ell)$, $Lip(\Psi)$,$Smt(\ell)$, $Smt(\Psi)$) required for our generalization bound is infeasible because determining these values is NP-hard [4]. As a result, directly plotting the precise generalization bound in Theorem 9 is not practical. This challenge is not unique to our work and applies to all generalization bounds that rely on Lipschitz constants, including those based on Rademacher complexity [5,6]  or uniform stability [7,8].
>
> (2) To address this, we follow established practices by validating our bound empirically. Specifically, we correlate it with the accuracy gap (difference between training and testing accuracy) and the loss gap (difference between training and testing loss). We conduct experiments to explore the relationship between generalization error and graph homophily on real-world datasets. The results show that, for a fixed training sample size $m$, the generalization behavior on real-world datasets mirrors the trends observed on synthetic datasets. These findings are presented in Fig.1(a-c)(g-i) in the revised version.

---

> > ### Comment · Reviewer_AQtc · 2024-12-02
> >
> > Unfortunately, I am still convinced that the generalisation bound is not vacuous assuming some realistic values for Lipschitz, etc.
> >
> > Could the authors maybe plot a curve assuming some realistic values of the relevant parameters?

---

> > > ### Author Response · Authors · 2024-12-04
> > > **calculated bounds**
> > >
> > > > Q1: I am still convinced that the generalisation bound is not vacuous assuming some realistic values for Lipschitz, etc. Could the authors maybe plot a curve assuming some realistic values of the relevant parameters?
> > >
> > >
> > > A1:
> > > Thank you for the suggestion.
> > >
> > >  We want to take this last chance to conduct two additional experiments to verify that the calculated bounds align with our theoretical analysis, and also align with the empirical accuracy gaps and loss gaps shown in Figure 1 of our paper.
> > >
> > >
> > > 1. **Synthetic Dataset**:
> > >    - Setup: 1,000 nodes, feature dimension 400, average node degree 5, $ H_{\text{edge}} = 0.1 $.
> > >    - Training samples: $ m $ varies from 100 to 900 (step 100).
> > >    - **Result (Table 1)**: Bounds **decrease** with increasing $ m $ in most cases, consistent with **Lemma 10** and **(d)-(f) in Figure 1**.
> > >
> > >
> > > | GNNs   | 100       | 200       | 300       | 400       | 500       | 600       | 700       | 800       | 900       |
> > > |--------|-----------|-----------|-----------|-----------|-----------|-----------|-----------|-----------|-----------|
> > > | **Training Sample Number $ m $** |||||||
> > > | **cheb**   | 2026.65   | 1615.39   | 1571.08   | 990.77    | 791.97    | 806.82    | 645.26    | 551.23    | 446.33    |
> > > | **chebii** | 522.67    | 698.58    | 727.53    | 666.02    | 530.87    | 348.22    | 283.31    | 203.76    | 157.62    |
> > > | **jacobi** | 715.34    | 1070.06   | 1195.99   | 1217.54   | 1082.42   | 737.02    | 595.62    | 456.52    | 317.38    |
> > > | **gpr**    | 11002.44  | 8734.97   | 4993.39   | 5691.92   | 1761.19   | 1285.22   | 1057.41   | 976.36    | 690.74    |
> > > | **bern**   | 497.93    | 576.84    | 602.32    | 557.68    | 440.45    | 339.07    | 234.03    | 189.62    | 111.31    |
> > >
> > > **Table 1:** Generalization bounds on synthetic dataset when graph node $n=1000$ and training sample number $m\in[100,900]$ with step $100$.
> > >
> > > 2. **Real-World Datasets**:
> > >    - Setup: Fixed training samples $ m = 100 $, datasets ranked by $ H_{\text{edge}} $ (low to high).
> > >    - **Result (Table 2)**: Bounds are smaller for datasets with extreme $ H_{\text{edge}} $ values (e.g., Texas, Cora) and larger in the middle in most cases, consistent with **Proposition 16** and **(g)-(i) in Figure 1**.
> > >
> > > | GNNs     | Texas    | Wisconsin | Squirrel    | Chameleon   | Citeseer  | Cora       |
> > > |----------|----------|-----------|-------------|-------------|-----------|------------|
> > > | **$0.0 \leftarrow H_{edge} \rightarrow 1.0$** |||||||
> > > | **cheb**  | 5742.97  | 3900.31   | 114511.58   | 39292.71    | 2267.07   | 4703.77    |
> > > | **chebii** | 347.63   | 3147.73   | 66046.45    | 68065.1     | 566.94    | 996.15     |
> > > | **jacobi** | 690.23   | 657.62    | 26315.65    | 38245.87    | 4817.62   | 555153600.0|
> > > | **gpr**    | 1982.21  | 2787.31   | 219304.73   | 83610.28    | 12070.99  | 37278.78   |
> > > | **bern**   | 341.34   | 474.44    | 167122.14   | 128725.05   | 552.41    | 770.75     |
> > >
> > > **Table 2:** Generalization bounds on real-world dataset when graph node $n$ varies and training sample number $m=100$.
> > >
> > > **Note**:
> > > - We set all Lipschitz constants and smoothness values to 1, potentially overestimating the bounds.
> > > - Accurately computing these constants is NP-hard, so exact theoretical bounds are infeasible [1].
> > >
> > > **Conclusion**: These results support our theoretical analysis .
> > >
> > > References:
> > >
> > > [1] Virmaux, Aladin, and Kevin Scaman. "Lipschitz regularity of deep neural networks: analysis and efficient estimation." Advances in Neural Information Processing Systems 31 (2018).

---

> ### Author Response · Authors · 2024-11-25
>
> > W5: Also the presentation of the results could be improved please label the points in Figure 2 for instance to show which datapoint represents which dataset.
>
> A5:  Thank you for the suggestion. In the revised version, we have added dataset names and their corresponding edge homophily ratios in the legend of Fig.1.
>
> > W6: The architecture restriction is quite severe.
>
> A6: We respectfully disagree with this statement. Spatial GNNs and spectral GNNs are two main streams of GNNs, each with distinct mechanisms and application areas. Spatial GNNs employ a message-passing mechanism and are widely used for graph classification tasks. In contrast, spectral GNNs process data in the spectral domain and achieve strong performance in node classification. Both paradigms are essential and cater to different types of problems. The architecture discussed in our paper represents a significant branch of popular and widely used spectral GNNs that utilize polynomial graph filters, including ChebNet, GPRGNN, BernNet, etc.
>
> > Q1: Theorem 6 is odd to me, it seems to not be specific to spectral GNNs none of the parameters of the graph $(n,f,\Pi,Q)$ seem to affect the transductive stability. In fact nothing except the size of the dataset m seems to affect the result. What am I missing?
>
> A1:
> Theorem 6 decomposes the $\gamma$-uniform transductive stability into two terms, $\gamma = r\beta$, where $r$ captures the Lipschitz continuity and smoothness of the loss function and spectral GNN, and $\beta$ represents the gradient norm bound. The parameters of the graph $(n, f, \Pi, Q)$ influence the gradient norm bound $\beta$, as shown in Theorem 8. Further, the term $A^k$ depends on the graph structure, which is determined by the parameter $Q$. The node features $X$ are governed by $\Pi$, shown as the mean $\pi_{y_i}$ and variance $\Sigma_{y_i}$ of nodes in class $y_i$. Thus, through their impact on $\beta$, the graph parameters $(n, f, \Pi, Q)$ affect the $\gamma$-uniform transductive stability.
>
> > Q2: Theorem 6 What is $\beta_2$?
>
> A2: $\beta_2$ is defined in Assumption 2. It bounds the gradient of the model output with respect to the model parameters.
>
> > Q3: Have you validated how well you can fit the cSBM model to the real world datasets?
>
> A3: (1) Thank you for your comment. We have added a discussion in the introduction of the revised manuscript to address this point. The contextual Stochastic Block Model (cSBM) is a widely used generative model that effectively captures both homophilic and heterophilic graph structures in a controlled and analytically tractable manner. Previous studies have demonstrated that cSBM models real-world datasets such as Citeseer, Cora, and Polblogs, which are frequently used in GNN research [9,10,11,12].
>
> (2) Our experimental results also confirm that theoretical analysis built on cSBM can be applied to real-world datasets. Details are in Section 6.
>
> [1] Chien, Eli, et al. "Adaptive universal generalized pagerank graph neural network." arXiv preprint arXiv:2006.07988 (2020).
>
> [2] He, Mingguo, Zhewei Wei, and Hongteng Xu. "Bernnet: Learning arbitrary graph spectral filters via bernstein approximation." Advances in Neural Information Processing Systems 34 (2021): 14239-14251.
>
> [3] He, Mingguo, Zhewei Wei, and Ji-Rong Wen. "Convolutional neural networks on graphs with chebyshev approximation, revisited." Advances in neural information processing systems 35 (2022): 7264-7276.
>
> [4] Virmaux, Aladin, and Kevin Scaman. "Lipschitz regularity of deep neural networks: analysis and efficient estimation." Advances in Neural Information Processing Systems 31 (2018).
>
> [5] Esser, Pascal, Leena Chennuru Vankadara, and Debarghya Ghoshdastidar. "Learning theory can (sometimes) explain generalisation in graph neural networks." Advances in Neural Information Processing Systems 34 (2021): 27043-27056.
>
> [6] Tang, Huayi, and Yong Liu. "Towards understanding generalization of graph neural networks." International Conference on Machine Learning. PMLR, 2023.
>
> [7] Verma, Saurabh, and Zhi-Li Zhang. "Stability and generalization of graph convolutional neural networks." Proceedings of the 25th ACM SIGKDD International Conference on Knowledge Discovery & Data Mining. 2019.
>
> [8] Zhou, Xianchen, and Hongxia Wang. "The generalization error of graph convolutional networks may enlarge with more layers." Neurocomputing 424 (2021): 97-106.
>
> [9] Deshpande, Yash, et al. "Contextual stochastic block models." Advances in Neural Information Processing Systems 31 (2018).
>
> [10] Lei, Jing. "A goodness-of-fit test for stochastic block models." (2016): 401-424.
>
> [11] Dreveton, Maximilien, Felipe Fernandes, and Daniel Figueiredo. "Exact recovery and Bregman hard clustering of node-attributed Stochastic Block Model." Advances in Neural Information Processing Systems 36 (2024).
>
> [12] Zhang, Yingxue, et al. "Detection and defense of topological adversarial attacks on graphs." International Conference on Artificial Intelligence and Statistics. PMLR, 2021.

---

> ### Author Response · Authors · 2024-11-29
>
> We would greatly appreciate your feedback on our rebuttal whenever you have a chance - thank you.

---

> ### Author Response · Authors · 2024-12-02
>
> The deadline for discussion is approaching.
>
> We would greatly appreciate your feedback on our rebuttal whenever you have a chance - thank you.

---

### Official Review · Reviewer_8f65 · 2024-11-10

**Soundness:** 2
**Presentation:** 2
**Contribution:** 2
**Rating:** 6
**Confidence:** 3

**Summary:**

The paper explores the generalization capabilities of spectral GNNs, examining how graph homophily and architectural choices affect GNN generalization in node classification tasks. The paper leverages the notion of $\gamma$-uniform transductive stability to obtain stability guarantees and, consequently, generalization bounds for spectral GNNs. From the theoretical analysis, the paper claims that spectral GNNs perform well on graphs with strong homophily or heterophily. Experimental results on synthetic and real-world datasets aim to support the theoretical insights.

**Strengths:**

- Relevance: Since generalization is one of the most important theoretical aspects of ML, the paper studies a very relevant problem in graph representation learning.

- Novelty: As far as I know, this is the first paper to theoretically investigate connections between generalization and homophily.

- Another positive aspect is that the empirical assessment considers five popular spectral GNNs.

**Weaknesses:**

**Generality**: The proposed analysis is valid for SBM-based graphs, which limit the generality of the paper's claims. Overall, the paper fails to discuss the impact of this choice --- how limiting is that choice? How well does such a class capture the distribution of real-world graph data?

**Inconsistent results**: Based on Figures 1 and 2, there is a clear difference in behavior for the experiments using synthetic and real-world datasets. I found the justification based on the number of training samples quite hand-wavy. The authors could strengthen the discussion with additional empirical analysis to confirm the proposed explanation.

**Missing related works**: The paper should acknowledge other prior works on the stability/generalization properties of GNNs, e.g., "Stability properties of GNNs by F. Gama, J. Bruna, and A. Ribeiro".

**Tightness**. The paper lacks a discussion regarding the tightness of the obtained generalization bounds. Also, how well do the obtained bounds correlate with observed generalization gaps?

**Parallel with prior results**: What parallel can one establish between the paper's findings and existing results regarding the stability/generalization of (spectral) GNNs? While I understand that comparing generalization bounds is challenging (e.g., due to different assumptions), the authors could try to connect their findings with what is known about the generalization capabilities (and stability properties) of GNNs.

**Impact**: The paper does not discuss how the theoretical results could be used to inform practitioners in designing better models. Thus, I found the overall impact of this contribution in practice limited.

**Questions:**

See weaknesses.

---

> ### Author Response · Authors · 2024-11-25
>
> > Q1: Generality: The proposed analysis is valid for SBM-based graphs, which limit the generality of the paper's claims. Overall, the paper fails to discuss the impact of this choice --- how limiting is that choice? How well does such a class capture the distribution of real-world graph data?
>
> A1: Thank you for your comment. We have added the motivation for using cSBM in Section 1. In summary, cSBM is useful for analyzing generalization due to the following reasons:
>
> 1. **Relevance to Real-World Datasets:** cSBM has been shown to effectively model real-world datasets such as Citeseer, Cora, and Polblogs, which are widely used in GNN research [1,2,3,4].
>
> 2. **Flexibility Graph Homophily:** cSBM can generate graphs with well-defined block structures. This allows us to model both homophilic graphs, where nodes within the same block are more likely to be connected, and heterophilic graphs, where connections are more likely between nodes in different blocks.
>
> 3. **Analytical Tractability:** cSBM provides a clear and systematic framework for theoretical analysis. By leveraging its properties, we can systematically vary graph homophily and study its impact on GNN generalization, which would be challenging with more complex or less structured graph models.
>
> These features make cSBM an ideal choice for exploring generalization properties in GNNs.
>
> > Q2: Based on Figures 1 and 2, there is a clear difference in behavior for the experiments using synthetic and real-world datasets. The authors could strengthen the discussion with additional empirical analysis to confirm the proposed explanation.
>
> A2: Thank you for your comment. We appreciate the opportunity to clarify the observed differences in behavior between synthetic and real-world datasets and have added further empirical analysis to strengthen the discussion.
>
> For the real-world datasets in Figure 2 (original version), we follow the standard data splitting protocol [5,6,7], where 60\% of nodes are used for training. As a result, the number of training samples (\(m\)) varies significantly across datasets, ranging from 183 samples in Texas to 19,717 samples in Pubmed. This variation in \(m\) contributes to the differences observed in the gaps on real-world datasets.
>
> To further investigate this, we conducted two additional experiments to isolate the effects of \(m\) and the homophilic ratio $H_{edge}$:
>
> 1. **Experiment 1:** We fixed the number of training samples to \(m = 100\) across all real-world datasets, which exhibit varying homophilic ratios.
> 2. **Experiment 2:** We varied the number of training samples \(m\) on synthetic datasets while keeping the homophilic ratio fixed.
>
> The results, now presented in Figure 1 of the revised manuscript, reveal the following:
> - When \(m = 100\) is used for all real-world datasets, the behavior of spectral GNNs aligns more closely with that observed on synthetic datasets (Figures 1(a-c) and 1(g-i)).
> - On synthetic datasets, the accuracy and loss gaps consistently decrease as the number of training samples increases (Figures 1(d-f)).
>
> These additional experiments confirm that the variations in training sample size (\(m\)) play a critical role in the observed differences between synthetic and real-world datasets.
>
> **Q3:** *The paper should acknowledge other prior works on the stability/generalization properties of GNNs, e.g., "Stability properties of GNNs by F. Gama, J. Bruna, and A. Ribeiro"*
>
>
> **A3:** Thank you for the suggestion. The paper "Stability properties of GNNs" examines the stability of GNNs by analyzing the maximum output change caused by perturbations in graph structure. However, their notion of stability is distinct from the concept of "uniform stability" that we focus on in this paper. Specifically, uniform stability measures the maximum change in a model's output when it is trained on two datasets differing by a single training sample and connects this to the generalization error bound. Thus, this paper and our work address different notions of stability.

---

> ### Author Response · Authors · 2024-11-25
>
> > Q4-Q5: Parallel with prior results: What parallel can one establish between the paper's findings and existing results regarding the stability/generalization of (spectral) GNNs? While I understand that comparing generalization bounds is challenging (e.g., due to different assumptions), the authors could try to connect their findings with what is known about the generalization capabilities (and stability properties) of GNNs. The paper lacks a discussion regarding the tightness of the obtained generalization bounds. How well do the obtained bounds correlate with observed generalization gaps?
>
> A4-A5: Thank you for the suggestion.
>
> We have added a discussion and a summary table (Table 1) in Section 2 of the revised manuscript to compare our work with existing studies across three key aspects:
>
> 1. Analysis Settings:
>
> Our work focuses on the transductive setting for node classification, similar to [8, 9, 10,11]. This is in contrast to works like [12,13], which analyze generalization in inductive settings.
>
> 2. Analysis Frameworks:
>
> We employ uniform stability, which allows us to analyze the relationship between generalization and graph homophily, a key factor that depends on both graph structure and node labels. Prior works using Rademacher complexity (e.g., [8,9,10]) cannot account for homophily as node labels cannot be considered when using Rademacher complexity for analysis. While [11] uses uniform stability, their analysis is limited to the effect of model depth.
>
> 3. Key Factors in Generalization Bounds:
>
> Unlike previous works that primarily focus on training sample size, model depth, and graph matrix norms, our analysis introduces two novel factors, graph homophily and polynomial order, into the generalization analysis for spectral GNNs. Furthermore, our finer-grained approach considers the expectation of individual graph matrix elements rather than just matrix norms.
>
> Direct comparisons of tightness are challenging due to differences in settings, frameworks, and assumptions across studies. However, by incorporating graph homophily and polynomial order, our work provides new insights into spectral GNNs' generalization behavior, which has not been explored before.
>
> > Q6: How can the theoretical results inform practitioners in designing better models?
>
> A6:
> We have added a new section (Section 5.3) to the revised manuscript that outlines the practical implications of our theoretical findings.
>
> In summary, our results suggest two key guidelines for improving model design:
>
> (1) Rewiring Graphs:
>    Our analysis demonstrates that graph homophily strongly influences generalization error. Specifically, graphs with higher homophily (or heterophily) tend to exhibit lower variance in k-length walks, revealing clearer structural patterns. This reduction in variance leads to a smaller gradient norm bound \(\beta\) ( Theorem 8, Theorem 13 ), improving the \(\gamma\)-uniform transductive stability ( Theorem 6). As a result, the generalization error bound decreases (Theorem 9). Therefore, practitioners can enhance the performance of spectral GNNs by focusing on graph structures that maximize homophily or heterophily.
>
> (2) Constrained Graph Convolution:
> Our results show that constraining the convolution parameters $0 \leq \theta_k \leq 1$ helps prevent the generalization error bound from increasing with the polynomial order $K$. This is because the constraint ensures that the condition in Proposition 16, where $\sum_{k=2}^K \theta_k \frac{(k-1)!}{2^{k-1}}$increases slower than  $\sum_{k=2}^K \theta_k^2 \frac{(k-1)!}{2^k}$, is violated, as $\theta_k \geq \theta_k^2$. Previous work [5] reports that constraining $\theta_k$ to non-negative values with Bernstein polynomial basis leads to valid polynomial filters. Our analysis further suggests adding the constraint $\theta_k \leq 1$ to maintain stable generalization error as $K$ increases.

---

> ### Author Response · Authors · 2024-11-25
>
> References:
>
> [1] Deshpande, Yash, et al. "Contextual stochastic block models." Advances in Neural Information Processing Systems 31 (2018).
>
> [2] Lei, Jing. "A goodness-of-fit test for stochastic block models." (2016): 401-424.
>
>
> [3] Dreveton, Maximilien, Felipe Fernandes, and Daniel Figueiredo. "Exact recovery and Bregman hard clustering of node-attributed Stochastic Block Model." Advances in Neural Information Processing Systems 36 (2024).
>
>
> [4] Zhang, Yingxue, et al. "Detection and defense of topological adversarial attacks on graphs." International Conference on Artificial Intelligence and Statistics. PMLR, 2021.
>
>
> [5] He, Mingguo, Zhewei Wei, and Hongteng Xu. "Bernnet: Learning arbitrary graph spectral filters via bernstein approximation." Advances in Neural Information Processing Systems 34 (2021): 14239-14251.
>
>
> [6] He, Mingguo, Zhewei Wei, and Ji-Rong Wen. "Convolutional neural networks on graphs with chebyshev approximation, revisited." Advances in neural information processing systems 35 (2022): 7264-7276.
>
>
> [7] Chien, Eli, et al. "Adaptive universal generalized pagerank graph neural network." arXiv preprint arXiv:2006.07988 (2020).
>
>
>
> [8] Esser, Pascal, Leena Chennuru Vankadara, and Debarghya Ghoshdastidar. "Learning theory can (sometimes) explain generalisation in graph neural networks." Advances in Neural Information Processing Systems 34 (2021): 27043-27056.
>
> [9] Tang, Huayi, and Yong Liu. "Towards understanding generalization of graph neural networks." International Conference on Machine Learning. PMLR, 2023.
>
>
> [10] Oono, Kenta, and Taiji Suzuki. "Optimization and generalization analysis of transduction through gradient boosting and application to multi-scale graph neural networks." Advances in Neural Information Processing Systems 33 (2020): 18917-18930.
>
>
> [11] Cong, Weilin, Morteza Ramezani, and Mehrdad Mahdavi. "On provable benefits of depth in training graph convolutional networks." Advances in Neural Information Processing Systems 34 (2021): 9936-9949.
>
>
> [12] Verma, Saurabh, and Zhi-Li Zhang. "Stability and generalization of graph convolutional neural networks." Proceedings of the 25th ACM SIGKDD International Conference on Knowledge Discovery & Data Mining. 2019.
>
>
> [13] Zhou, Xianchen, and Hongxia Wang. "The generalization error of graph convolutional networks may enlarge with more layers." Neurocomputing 424 (2021): 97-106.

---

> ### Author Response · Authors · 2024-11-29
>
> We would greatly appreciate your feedback on our rebuttal whenever you have a chance - thank you.

---

> > ### Comment · Reviewer_8f65 · 2024-11-30
> >
> > I thank the authors for their reply. Overall, my concerns were partially addressed --- e.g., I am still not sure how vacuous the generalization bound is and how we can leverage it to perform graph rewiring in an end-to-end fashion. Nonetheless, since the authors have considered some suggestions, clarified some aspects of their contribution, and added new experiments, I am raising my score.

---

> > > ### Author Response · Authors · 2024-12-01
> > >
> > > Thank you for reviewing our response and for your support.
> > >
> > > We would like to provide one additional clarification: one of the practical guidelines presented in Section 5.3 involves rewiring graphs to maintain high homophily. Notably, a study that achieves this in an end-to-end manner is [1].
> > >
> > > [1] Li, Shouheng, Dongwoo Kim, and Qing Wang. "Restructuring graph for higher homophily via adaptive spectral clustering." Proceedings of the AAAI Conference on Artificial Intelligence. Vol. 37. No. 7. 2023.

---

### Author Response · Authors · 2024-11-25

Thank you all for the constructive feedback. We appreciate the recognition of our work’s originality in analyzing the relationship between the generalization of spectral graph neural networks (GNNs) and graph homophily, polynomial order. To clarify and improve the paper, we have made the following changes in the revised manuscript:

1. **Motivation for cSBM**: We added a discussion in Section 1 explaining the reasoning behind using the contextual stochastic block model (cSBM) in our theoretical analysis.

2. **Generalization Bound Comparison**: We added a table summarizing the comparison of our work with existing studies and expanded the related works discussion in Section 2.

3. **Proof Sketches**: We included proof sketches for Theorems 8 and 13 in the main content and provided hyperlinks to the full proofs in the appendix for easy access.

4. **Practical Guidelines**: We added a new section in Section 5.3 to discuss practical guidelines and insights derived from our theoretical analysis.

5. **Graph Homophily and Training Samples**: We conducted experiments to analyze the relationship between graph homophily and generalization. To isolate the effect of graph homophily, we used the same training sample size across real-world datasets and analyzed accuracy and loss gaps. We also examined the impact of increasing training samples on synthetic datasets. Results are presented in Figure 1 and discussed in detail in Section 6.

6. **Polynomial Order Condition**: We conducted experiments to illustrate the conditions under which the generalization error increases with polynomial order, as discussed in Theorem 14 and Proposition 16. Results are presented in Figure 2 and elaborated in Section 6.

7. **Large-Scale Dataset Validation**: We extended our experiments to large-scale datasets (Ogbn-Arxiv and Ogbn-Products) to validate the applicability of our theoretical analysis. Results are added to Figure 1 in Section 6 in our revised version.

We hope these revisions thoroughly address your concerns. Should you require further clarification, please let us know. We would greatly appreciate your consideration of a higher score if these changes meet your expectations. Thank you again for your valuable feedback.

---

### Author Response · Authors · 2024-11-27

We sincerely thank you for your thoughtful reviews. In response to your suggestions, we have reorganized the proofs in the appendix to make them more structured and concise. Further, we have included a detailed table of contents for the appendix, making it easier to locate specific proofs.

Should you require further clarification, please let us know. We would greatly appreciate your consideration of a higher score if these clarifications addressed your concerns. Thank you again for your valuable feedback.

---

### Meta-Review · Area_Chair_mTrg · 2024-12-19

**Metareview:**

**Summary**:
This work investigates the generalization of one-layer spectral graph neural networks (GNNs) on the task of node binary classification, assuming the graphs and node labels are generated using a contextual stochastic block model (cSBM).  Specifically, the authors provide generalization bounds via uniform transductive  stability. The impact of homophily (affinity with neighbors, based on label information) on the performance of these GNNs is established.  Empirical validation is provided on several synthetic and real-work benchmark datasets.

**Strengths**:
The reviewers acknowledged the key strengths of this work: (a) the novelty of this work in showing theoretical connections between homophily/heterophily  and generalization, and (b) experiments that seemed to substantiate the theory in at least some aspects on synthetic datasets. Many of them  also found the presentation to be well-structured and organised.


**Weaknesses**:
The reviewers also raised several concerns and suggestions for improving the work, including, missing related work, inconsistency of results between synthetic data and real-world data, possible vacuousness of the bounds, comparison with prior works, limited perceived impact of this work,  unconvincing assumptions (e.g., full-batch gradient descent,  restricted form of spectral GNN architecture, fixed learning rate, concerns about assumptions for uniform transductive stability being too strong), apparent disconnect between generalization and homophily results, and analysis being restricted to  single-layer spectral GNNs.

Further concerns were raised, including,  (a) limited technical novelty of the generalization bounds (given previous results on transductive learning), (b) results being hard to interpret, (c)  unclear effect of polynomial order on generalization, (d) small datasets, and (e) lack of additional metrics.

**Recommendation**:

During the rebuttal period, authors addressed many of the concerns of the reviewers but several remained unresolved as well.  I also noted some other issues/questions that need to be clarified/fixed; e.g.,

*Some prior works need to be properly credited*; e.g., Garg et al. (2020) proved the first data-dependent generalization bounds for GNNs, and their bounds already included depth as well as maximum node degree; a setting that was inherited by PAC-Bayes works such as Liao et al. (2021).

*How strong is the bounded gradients assumption?* The authors mention that the cross-entropy loss violates this assumption, however what effect does this assumption have in practice should be conveyed. For example, how does the performance compare when margin loss is used versus cross-entropy. Also, the need for convergence under optimization might be too strong as well. In particular, results from works on (non-spectral) GNNs such as Garg et. al do not make this assumption (albeit in the context of graph classification).

*The title is too generic*: it should be clearly mentioned that the result applies to only single layer spectral networks under a contextual stochastic block model.

*It is not clear whether/how the stochastic block model takes into account symmetries in a graph*; e.g., nodes with identical features and neighborhoods.

Given these concerns, despite its promise, I don’t think the paper in its current form meets the bar for ICLR.

**Additional Comments On Reviewer Discussion:**

The key concerns expressed by reviewers, and the authors' responses, are summarised below.

Concerns regarding missing related work*:  The authors convincingly explained during response how their work employs a different notion of stability than Gama et. al. Analyzing how the bounds differ in two cases would strengthen the work.

*Inconsistency of results between synthetic data and real-world data*: Some of the experimental results with real datasets deviated from the expected behaviour, in contrast to synthetic datasets.  The authors provided further empirical evidence during the response period to mitigate this concern.

*Tightness of the bounds*:  Specifically, whether the generalization bounds were vacuous from theoretical and empirical perspectives? I don’t think this question was satisfactorily addressed even after the authors’ response as two reviewers explicitly pointed out.

*Comparison with prior works*: The authors added a summary table differentiating their work from prior related works along three axes: transductive vs inductive,  specific notion of stability, and assumptions.  However, I believe they should make further effort to contrast the theoretical dependence on different hyperparameters in each case.

*Limited perceived impact of this work*: The authors response partially addressed this concern by suggesting employing spectral GNNs for settings with high homophily or heterophily possibly by appealing to graph rewiring,  and constraining graph convolution parameters.

*Unconvincing assumptions (e.g., full-batch gradient descent,  restricted form of spectral GNN architecture, fixed learning rate, concerns about assumptions for uniform transductive stability being too strong)*: Many of these concerns remained unresolved even after the response period.

*Apparent disconnect between generalization and homophily results*: The authors resolved this concern by arguing that the two were not independent.

*Single-layer Spectral GNNs*: Reviewers also raised concerns about the analysis being restricted to single-layer spectral GNNs. During the response, the authors emphasized that several of the common spectral GNNs such as ChebNet, BernNet, and JacobiConv employed only a single layer. I think this is a fair point, though, it would indeed be nice to have analysis of multilayer extensions.

However, the authors also made some remarks that reviewers objected to in this context. In particular, it is incorrect to say that the higher polynomial order serves an analogous purpose as having multiple layers. I also agree with the reviewer’s assertion that  the authors’ argument that graphs lack inherent hierarchy (unlike images) is misleading/incorrect.

Some other concerns were raised, including,  (a) limited technical novelty of the generalization bounds (given previous results on transductive learning), (b) results being hard to interpret, (c)  unclear effect of polynomial order on generalization, (d) small datasets, and (e) lack of additional metrics. The reviewers addressed (c), (d), and (e) but questions remain about (a) and (b).

Since some of the concerns that were not sufficiently addressed are rather significant, I am not able to recommend this paper for acceptance.

---

### Decision · Program_Chairs · 2025-01-22

Reject